# Adakites, High-Nb Basalts and Copper–Gold Deposits in Magmatic Arcs and Collisional Orogens: An Overview

**Pavel Kepezhinskas** [1,2,*] **, Nikolai Berdnikov** [1] **, Nikita Kepezhinskas** [3] **and Natalia Konovalova** [1]

1   Kosygin Institute of Tectonics and Geophysics, Russian Academy of Sciences, 680000 Khabarovsk, Russia; nick@itig.as.khb.ru (N.B.); turtle_83@mail.ru (N.K.)
2   PNK GeoScience, Tampa, FL 33647, USA
3   Department of Earth and Atmospheric Sciences, University of Alberta, Edmonton, AB T6G 2R3, Canada; kepezhin@ualberta.ca
*   Correspondence: pavel_k7@yahoo.com

**Abstract:** Adakites are Y- and Yb-depleted, $SiO_2$- and Sr-enriched rocks with elevated Sr/Y and La/Yb ratios originally thought to represent partial melts of subducted metabasalt, based on their association with the subduction of young (<25 Ma) and hot oceanic crust. Later, adakites were found in arc segments associated with oblique, slow and flat subduction, arc–transform intersections, collision zones and post-collisional extensional environments. New models of adakite petrogenesis include the melting of thickened and delaminated mafic lower crust, basalt underplating of the continental crust and high-pressure fractionation (amphibole ± garnet) of mantle-derived, hydrous mafic melts. In some cases, adakites are associated with Nb-enriched (10 ppm < Nb < 20 ppm) and high-Nb (Nb > 20 ppm) arc basalts in ancient and modern subduction zones (HNBs). Two types of HNBs are recognized on the basis of their geochemistry. Type I HNBs (Kamchatka, Honduras) share N-MORB-like isotopic and OIB-like trace element characteristics and most probably originate from adakite-contaminated mantle sources. Type II HNBs (Sulu arc, Jamaica) display high-field strength element enrichments in respect to island-arc basalts coupled with enriched, OIB-like isotopic signatures, suggesting derivation from asthenospheric mantle sources in arcs. Adakites and, to a lesser extent, HNBs are associated with Cu–Au porphyry and epithermal deposits in Cenozoic magmatic arcs (Kamchatka, Phlippines, Indonesia, Andean margin) and Paleozoic-Mesozoic (Central Asian and Tethyan) collisional orogens. This association is believed to be not just temporal and structural but also genetic due to the hydrous (common presence of amphibole and biotite), highly oxidized (>ΔFMQ > +2) and S-rich (anhydrite in modern Pinatubo and El Chichon adakite eruptions) nature of adakite magmas. Cretaceous adakites from the Stanovoy Suture Zone in Far East Russia contain Cu–Ag–Au and Cu–Zn–Mo–Ag alloys, native Au and Pt, cupriferous Ag in association witn barite and Ag-chloride. Stanovoy adakites also have systematically higher Au contents in comparison with volcanic arc magmas, suggesting that ore-forming hydrothermal fluids responsible for Cu–Au(Mo–Ag) porphyry and epithermal mineralization in upper crustal environments could have been exsolved from metal-saturated, $H_2O$–S–Cl-rich adakite magmas. The interaction between depleted mantle peridotites and metal-rich adakites appears to be capable of producing (under a certain set of conditions) fertile sources for HNB melts connected with some epithermal Au (Porgera) and porphyry Cu–Au–Mo (Tibet, Iran) mineralized systems in modern and ancient subduction zones.

**Keywords:** metal-rich adakite magmas; slab melting; lower crustal melting; high-Nb basalts; adakite–mantle interaction; Cu–Au deposits; Stanovoy suture; Cu–Ag–Au alloys; $H_2O$–S–Cl fluids

## 1. Introduction

The origin of intermediate to felsic magmas has been the subject of intense scientific debate for several decades. These studies were primarily triggered by two fundamental characteristics of andesitic magmatism, e.g., its general compositional similarity with

the average continental crust [1–3] and its close spatial and temporal association with copper–gold porphyry and epithermal gold mineralization in magmatic arcs and collisional orogens [4–7]. Early models for andesite genesis include (1) fractional crystallization of island-arc tholeiite or calc-alkaline basaltic magma [8–14], (2) hydrous melting of mantle peridotite [11,14–17], (3) mixing between mafic and felsic melts in crustal magmatic conduits [18–22], (4) crustal contamination [11,14,23–26], (5) anatexis of mafic lower crust via intrusion or underplating of hot basaltic magma [11,27,28] and (6) direct melting of subducted oceanic crust [29–32]. Numerous papers propose that multiple sources and processes combined to produce intermediate to felsic magmas in ancient and modern subduction zones [11,14,16,26] and references therein.

Several studies emphasized the importance of elevated geotherms in facilitating slab melting in subduction-like regimes associated with the hotter Precambrian Earth [29,32–34]. Although later these early ideas were met with vigorous criticism [35,36] and references therein, it was the recognition of the importance of elevated temperatures in subducting basaltic crust as precursor to generation of felsic magmas that lead to the conception and advancement of the "slab melting/adakite" hypothesis. Based on the available experimental work and close geographic association between high-Sr, low-Y siliceous magmas and the subduction of the young oceanic lithosphere, Defant and Drummond suggested that dehydration melting of garnet amphibolite in subduction zones occurs only when exceptionally young (typically < 25 Ma) basaltic crust is being subducted [33,37]. Almost at the same time, the senior author of the current study documented an association of sodic, hornblende-rich andesites with the subduction of the very young crust generated in the Komandorsky back-arc Basin (Bering Sea) beneath the northern Kamchatka arc [38]. Defant and Drummond named these geochemically unique andesitic and dacitic rocks "adakites" [37] following the historic occurrence of the slab-derived magnesian andesites on Adak Island, Alaska, documented earlier by Kay [31]. Adakites display peculiar geochemical characteristics such as high Sr/Y and La/Yb ratios (in comparison with "normal" calc-alkaline rocks) consistent with the model involving dehydration melting of hot basaltic crust during amphibolite to eclogite transition in the subducting slab with garnet- and amphibole-dominated solid residue controlling the chemistry of adakitic magmas [33,37,39].

These papers triggered a flurry of publications on slab melting and adakites over the next decade that examined tectonic setting and conditions for adakite generation in subduction zones ranging from the Aleutians, Kamchatka and Japan to the Philippines and the Aegean Islands on to Central and Southern America [40–55]. In addition, Defant and his co-authors documented an association of slab melts (adakites) in Central America with unusual sub-alkaline to alkaline basalts generally lacking high-field strength element (HFSE, such as Nb) depletions characteristic of "normal" calc-alkaline magmas, and consequently called them high-Nb basalts [56]. Kepezhinskas and co-authors investigated mantle xenoliths included in high-Nb basalts from the Northern Kamchatka arc and proposed their formation via slab melt–mantle wedge interaction and the subsequent low degree melting of hybridized HFSE-enriched sub-arc mantle source [57,58]. This was further corroborated by trace element and isotope geochemistry of high-Nb basalt magmas in Kamchatka, the Philippines and the southernmost Andes [51,52,59,60]. Ancient analogues of modern high-Nb basalts, frequently in association with adakites and MgO-rich andesites, were later documented in Precambrian greenstone belts [61–63] and Phanerozoic collisional orogens [64,65].

As adakites have been later found in post-subduction/collision settings or without customary association with the young oceanic crust, several alternatives to slab melting models have been put forward over the last 40 years that include (1) the melting of thickened continental lower crust [66–71]; (2) the re-melting of adakite-like rocks or adakite-hybridized garnet amphibolites tectonically incorporated into continental (cratonic) crust [72,73]; (3) the high- or low-pressure fractionation of mantle-derived magmas [74–78]; and (4) low degrees of melting of garnet- and amphibole-bearing metasomatized mantle peridotite [79–81]. Although various research has attempted to address potential composi-



tional differences (such as $SiO_2$, $Na_2O$, $K_2O$ and MgO contents, for example) between "true" slab melts such as adakites from North Kamchatka, Central America, some locations in the Philippines and the Aleutian arc as well as lower crustal ("continental", or C-adakites) adakites, it is quite difficult to establish clear compositional distinctions between adakite melts derived from different sources through diverse petrogenetic processes. As Defant and Kepezhinskas pointed out earlier [60], various geological criteria (presence of very hot and young oceanic crust, absence of thick, garnet-bearing continental crust, etc.) should be utilized to establish credible environments for adakite generation.

Adakites, in essence, are water-rich intermediates to felsic magmas, which are frequently associated with precious and base metal mineralization in ancient and modern subduction zones, collision and post-collision environments. Several pioneering papers documented the association of major epithermal gold and copper–gold porphyry mineralization in the Philippines with young adakitic magmatism [82,83]. Defant and Kepezhinskas have shown that copper–gold mineralization in volcanic arcs can be associated with the entire "adakite triad", e.g., adakites, mantle-contaminated high-Mg adakites (magnesian andesites) and high-Nb basalts (HNBs) formed during the low-degree melting of adakite-hybridized mantle sources [60]. Various researchers have recognized that giant Cenozoic copper deposits in Chile are hosted by adakitic intrusions, while smaller mineral showings tend to be associated with calc-alkaline granitoids [84,85]. Later, Chinese scientists proposed that ridge subduction may facilitate the formation of arc-related copper mineralization as the oceanic crust is endowed with copper contents several times higher than delaminated continental crust [86]. It was also suggested that prolonged magnetite crystallization accompanied by sulfate reduction triggers the partitioning of copper into ore-forming fluid during crustal fractionation of oxidized adakite magma [86,87]. This concept was challenged by several researchers, who argued that high Sr/Y geochemical signature in mantle-derived calc-alkaline magmas associated with Cu–Mo–Au mineralization can be created through the hydrous fractionation of amphibole, titanite and, possibly, garnet [7,88–90]. While the discussion on the exact nature of Sr/Y signatures in rocks associated with precious and base metal mineralization in subduction-related environments is still quite alive and open, adakites (and, to a lesser extent, high-Nb basalts) are being utilized by mining companies as useful exploration tools for the highly successful localization of individual copper–gold deposits as well as new mineralized trends, districts and entire metallogenic provinces (Tibet and the Central Asian Orogenic Belt can be mentioned among the latter). After all, some of the largest Cu–Au deposits on this planet are related to adakite magmatism.

We present in this paper an overview of main occurrences of Cu–Au mineralization (both porphyry and epithermal) associated with adakites and high-Nb basalts along with a discussion of existing ore-forming models in subduction zones with reference to slab and lower crustal melting, adakite production and adakite–mantle wedge hybridization.

## 2. Adakites

Adakites are commonly defined [37–39,49–53,60,80,81,91–115] as intermediate to felsic (54–70 wt.% $SiO_2$), Al-rich (typically > 15 wt.% $Al_2O_3$), predominantly sodic volcanic and plutonic rocks with abundant amphibole phenocrysts, pronounced Y and Yb depletions and characteristically high Sr/Y (>40) and La/Yb (>20) ratios (Table 1). Adakite magmas in most cases are associated with convergent plate boundaries, where they occur in oceanic and continental arc settings, as well as collision zones and accretionary belts (Table 2). Some adakite occurrences in post-collisional and continental rifting environments have also been reported (Table 2; [69,72,73]). Adakites typically form lava flows, scoria, tuffs and volcanic breccias, dikes and shallow extrusive plugs and domes, as well as deeper, frequently multiphase plutons within a variety of subduction-related environments [31,38,40,41,45,49,66]. Petrographically, adakites are characterized by abundant amphibole megacrysts, phenocrysts and microphenocrysts (several populations are usually present reflecting multiple stages of crystal fractionation under hydrous crustal conditions) along with clinopyrox-

ene (which frequently forms cores of zoned amphibole megacrysts and phenocrysts) and plagioclase [38,76,78,101]. It is important to emphasize that in some primitive (high-Mg) adakites, amphibole crystallizes during early stages of magmatic differentiation and is texturally and compositionally interpreted as a near-liquidus (at temperatures of 975–1050 °C and pressures up to 1.5 GPa) phase [38,52,76,81,99,100]. In some cases, adakites contain abundant crystal clots and cognate inclusions indicative of polybaric crustal fractionation of their parental melts [38,45,52,54,100–103].

**Table 1.** Petrologic and geochemical characteristics of adakites [37–39,49–53,60,80,81,91–115].

| Adakite Characteristics | Petrologic Interpretations |
|---|---|
| Abundant high- to low-pressure amphibole (multiple populations of megacrysts, phenocrysts and microphenocrysts) | High water content of pristine adakites or multi-stage fractionation under hydrous conditions |
| High $SiO_2$ content (typically > 55 wt.%) | High-P melting of basaltic source (eclogite or garnet amphibolite) |
| High $Al_2O_3$ ($\geq$15 wt.%) contents | High-P partial melting of garnet-bearing source (eclogite, garnet amphibolite) |
| High $Na_2O$ (>3 wt.%) and $Na_2O/K_2O$ > 1 | Sodic melts; melting of plagioclase or minor plagioclase in the source |
| High Sr (400–3000 ppm) and absence of negative Eu anomaly | Melting or lack of plagioclase in the residue; plagioclase accumulation during differentiation of primary adakite melts |
| Low MgO (<3 wt.%), low Cr and Ni | Melts from basaltic source; lack of interaction with peridotitic mantle wedge |
| Low Y ($\leq$18 ppm) and Yb ($\leq$1.8 ppm) contents, high Sr/Y (> 30) and La/Yb (>20) ratios | Garnet (also possibly amphibole and clinopyroxene) in the melting residue |
| Low high-field strength element (HFSE) contents (e.g., Nb and Ta) | Residual titanite, rutile or amphibole in the source |
| High Zr/Sm ratio (>100) | Amphibole fractionation or amphibole in the melting residue |
| Low K/La, Rb/La and Ba/La; low $^{87}Sr/^{86}Sr$, low $^{206}Pb/^{204}Pb$ and high $^{143}Nd/^{144}Nd$ isotope ratios | N-MORB-like basaltic source |
| Oxygen isotopes | Mixing of partial melts from different parts of the slab, or isotope exchange during slab melt–mantle wedge interaction |
| Low $\delta^7Li$ (+1.4 to +4.2) and low B/Be ratios (5.4 to 7.7) | Melting of a devolatilized MORB slab |
| Highly variable $^{187}Os/^{188}Os$ isotope ratios | Melting of mafic source or mantle peridotite followed by crustal assimilation |
| Uniform excess of $^{230}Th$ over $^{238}U$ combined with variable Th isotopes | Approximately 20% equilibrium melting due to amphibole decomposition in a heterogeneous subducting oceanic slab |

**Table 2.** Tectonic environments of adakite formation.

| Tectonic Environment | Geographic Examples | References |
|---|---|---|
| Subduction of young (<25 Ma) oceanic crust | Northern Kamchatka, Philippines, Costa Rica, Peru, Chile | [37,38,45,50–52,60,81,91] |
| Ridge (spreading center) subduction | Austral Andes, Panama, Columbia, Ecuador | [37,50,60,81,86] |
| Early stages of subduction/incipient subduction | Mindanao (Philippines), central Inner Mongolia (North China), SW Japan | [43,81,116,117] |

**Table 2.** *Cont.*

| Tectonic Environment | Geographic Examples | References |
|---|---|---|
| Flat slab subduction | Southern Ecuador, Peru, Chile, southern Alaska, Hailar Basin (NE China) | [105,118–120] |
| Oblique subduction | Central Aleutians (Adak), Panama. Solander Island (New Zealand), Wrangell Arc [1] (South Alaska) | [31,47,49,60,121,122] |
| Duplicate (overlapping) slabs | Central Japan | [123] |
| Transform faulting and slab tear | Central Kamchatka, Western Aleutians, Southern Tibet | [124–126] |
| Melting of mafic lower crust or underplated oceanic crust | Highlands of Papua New Guinea, Tibet | [39,67,68,70,72,73] |
| Post-subduction slab window, upwelling of asthenospheric mantle, slab break-off | Camaguin Island (Philippines), Baja California | [74,79,126,127] |
| Post-collisional and intracontinental (cratonic) extension | Tibet, North China craton, Dabie orogen, Dexing (South China) | [69,72,73,90,106,109] |

[1] Also a flat-slab region.

The origin of adakite magmas in various tectonic settings has been subject of the intense scientific debate over the last 30 years [28,33,37,39,60,67,78,80,81]. We believe that all adakite formation models can be grouped into three principal scenarios: (1) partial melting of subducted oceanic crust (young and hot crust, slab tear, ridge, incipient and flat slab subduction, etc.; Table 2) [31,33,37–44,48,50–53,98,124]; (2) partial melting of thickened, delaminated or underplated collisional or continental crust [28,67–73,107–110]; and (3) protracted (high- to moderate-pressure) fractionation of amphibole ($\pm$garnet) from hydrous, mafic mantle melts [40,74–78,100,102,114]. Variations of major elements suggest that some subtle, but detectable, differences exist between adakites formed by these diverse petrologic processes (Figure 1). Adakites associated with lower crustal melting appear to have systematically lower $Al_2O_3$ and higher $K_2O$ contents in comparison with slab melts and mafic differentiates, while MgO contents and Sr/Y ratios display almost complete overlap between the three types of adakitic compositions (Figure 1). This picture is further complicated by the effects of low-pressure fractionation on adakite magmas during their crustal emplacement [40,41,76,78,81,101] and intense (at least in some cases, e.g., Kamchatka, Philippines, Aleutians, Austral Andes) interactions between slab melts and peridotites from the overlying mantle wedge [38,49,51,54,57,58]. Adakites are distinguished from calc-alkaline and tholeiitic magmas in modern oceanic and continental arcs by clear depletions in yttrium and heavy rare-earth elements (HREE) (e.g., Yb; Figure 2). On the other hand, adakites share certain geochemical similarities with Archean tonalite-trondhjemite-granite (TTG) rocks, which, however, are clearly distinguished from adakites by more pronounced high-field strength element depletions and HREE fractionations (Figure 2; [32,33,35,50,80]). Based on this short summary of adakite geochemical characteristics, we are of the strong belief that adakite reconstructions should not be based exclusively on geochemical criteria but should necessarily include a wide range of geological, geochronological and structural information from specific adakite-bearing crustal terranes.

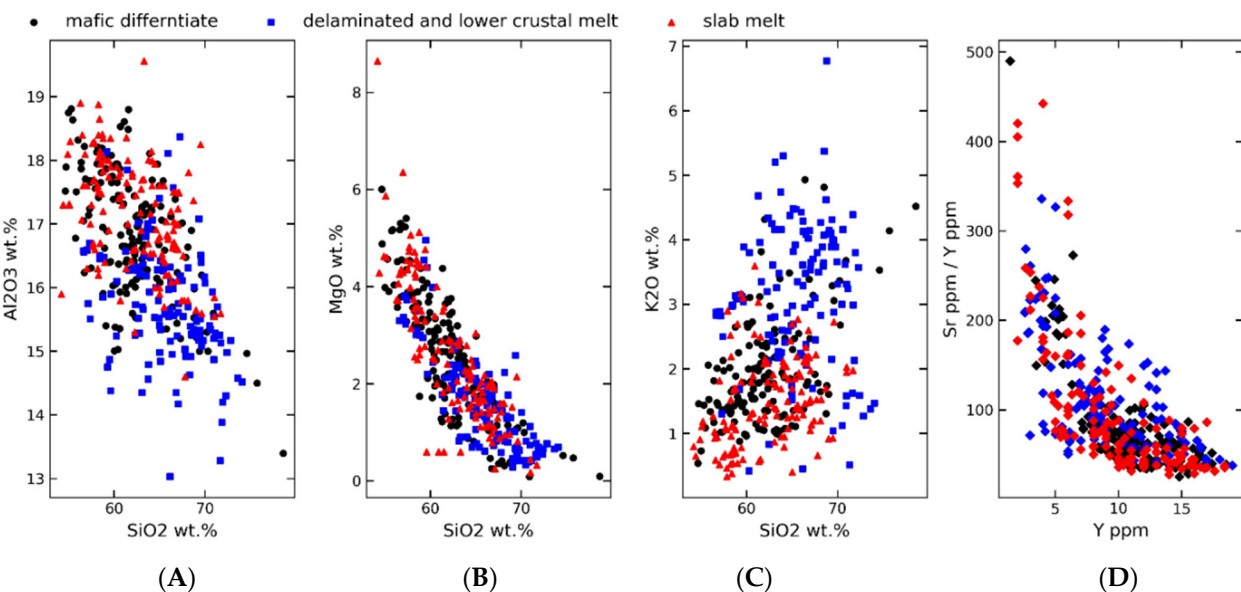

**Figure 1.** Variations of SiO$_2$ vs. MgO (**A**), SiO$_2$ vs. Al$_2$O$_3$ (**B**), SiO$_2$ vs. K$_2$O (**C**) and Sr/Y vs. Y (**D**) in adakites formed through slab melting (*n* = 141) [38,40,41,43,45,48,51,52,55,56,64,103,105], underplating and lower crustal melting (*n* = 160) [66,68,69,71–73,106–112] and high-pressure fractionation of mafic magmas (*n* = 156) [74–77,102,113–115].

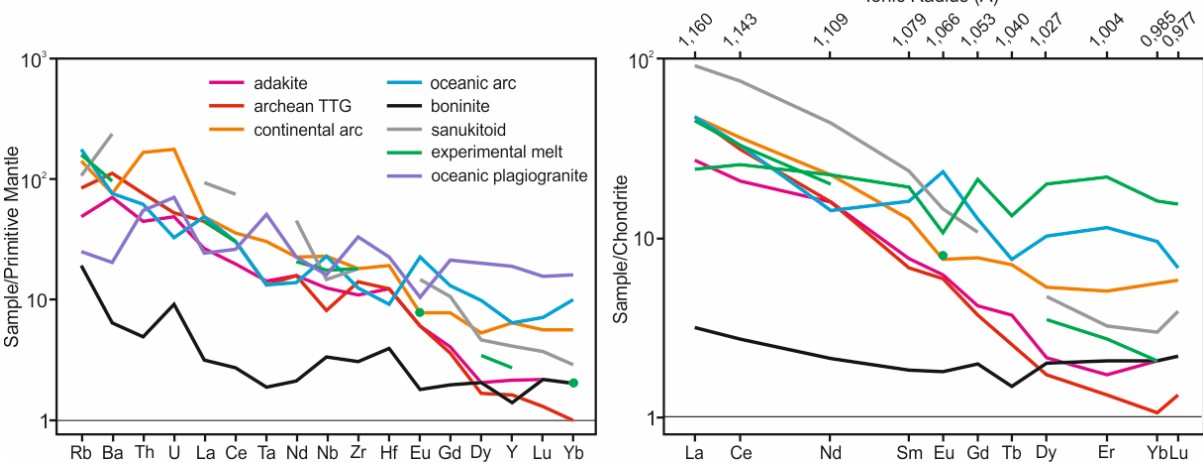

**Figure 2.** Geochemical characteristics of adakite compared to average Archean tonalite-trondhjemite-granodiorite (TTG) [50], andesite-dacite-rhyolite from continental and oceanic arcs [50], boninite [50], Archean sanukitoid [80], oceanic plagiogranite [50] and experimental melt from metabasaltic source [80]. Primitive mantle and chondrite normalizing values are from [104].

## 3. High-Nb Basalts

During their studies of adakite magmatism throughout the Central American volcanic arc, Defant and co-authors noticed that adakites in Panama and Costa Rica are frequently associated with basalts that are lacking negative Nb anomalies characteristic of subduction-zone magmas [56]. Similar mafic lavas (along with "normal" calc-alkaline basalts) were earlier described by Gill and Reagan at the Turialba volcano (Costa Rica) and dubbed "high-niobium basalts" to be distinguished from high-field strength element (HFSE)-depleted calc-alkaline volcanic series erupted within the same volcanic center [128]. Similar Nb-enriched basalts were later reported from Kamchatka [52,58], the Philippines [59] and Baja California [55], always in close spatial and temporal association with adakite magmas. High-niobium basalts (HNBs) are best defined (Table 3, Figure 3) as sub-alkaline to alkaline (Na$_2$O + K$_2$O > 3.5 wt.%).

**Table 3.** Representative compositions of high-Nb basalts (HNB) in volcanic arcs.

| Location | Kamchatka | Panama | Baja California | Philippines | Jamaica | Sulu | Utila, Honduras | Izu-Bonin |
|---|---|---|---|---|---|---|---|---|
| Age | 7 Ma | 2.35 Ma | 9.47 Ma | 1.98 Ma | ~52 Ma | Pliocene-Pleistocene | Late Pleistocene | 0.43 Ma |
| Sample# | VAL55 | P-26 | 99–131 | PH93-39 | AHWG08 | SBK13 | UTI1715 | FK-15 |
| Ref. | [52] | [56] | [55] | [59] | [129] | [130] | [131] | [132] |
| $SiO_2$ (wt.%) | 48.08 | 45.5 | 49.00 | 53.20 | 51.21 | 49.44 | 48.83 | 48.53 |
| $TiO_2$ | 2.03 | 2.27 | 1.67 | 1.73 | 3.05 | 2.07 | 1.60 | 1.69 |
| $Al_2O_3$ | 15.89 | 12.6 | 14.86 | 14.50 | 16.73 | 15.81 | 17.26 | 17.01 |
| $Fe_2O_3$ | 10.20 | 11.0 | 11.50 | 11.20 | 9.08 | 11.35 | 9.86 | 11.75 |
| MnO | 0.16 | 0.15 | 0.15 | 0.16 | 0.14 | 0.17 | 0.18 | 0.18 |
| MgO | 8.51 | 10.90 | 9.10 | 7.50 | 4.21 | 7.66 | 6.04 | 7.77 |
| CaO | 9.67 | 11.70 | 7.80 | 8.60 | 9.06 | 8.95 | 8.07 | 9.31 |
| $Na_2O$ | 3.01 | 2.65 | 3.60 | 2.89 | 4.64 | 2.97 | 3.61 | 3.01 |
| $K_2O$ | 1.96 | 1.10 | 1.15 | 0.54 | 0.62 | 1.20 | 2.34 | 1.11 |
| $P_2O_5$ | 0.50 | 1.00 | 0.35 | 0.24 | 0.77 | 0.35 | 0.56 | 0.31 |
| LOI | 0.00 | 0.40 | 0.63 | 0.18 | | 0.84 | 1.01 | 0.19 |
| Total | 100.04 | 99.27 | 99.86 | 100.50 | 99.51 | 99.97 | 99.36 | 100.67 |
| Cr (ppm) | 339 | 299 | | 274 | 62 | 231.2 | 170 | 144 |
| Ni | 136 | 218 | | 190 | 70 | 136.0 | 68 | 101 |
| V | 281 | | | 190 | 204 | 240.1 | 152 | 216 |
| Rb | 22 | 22 | 14.6 | 9.3 | 2.7 | 28.5 | 41.47 | 27.0 |
| Ba | 448 | 800 | 255 | 89 | 823 | 308 | 532 | 356 |
| Sr | 667 | 1200 | 495 | 298 | 489 | 388.4 | 528 | 566 |
| Zr | 162 | 160 | | 128 | 153.5 | 160.7 | 199 | 111 |
| Y | 22 | 22 | 18.5 | 23.4 | 22.2 | 24.9 | 27.2 | 19.7 |
| Nb | 32.1 | 51 | 26.5 | 25.8 | 47.21 | 35.2 | 71.19 | 24.5 |
| Ta | | | | | 2.73 | | 3.83 | 1.37 |
| Hf | 3.7 | | | | 3.39 | | 3.99 | 2.76 |
| Th | 3.0 | 5.8 | | | 3.37 | 2.8 | 5.03 | 2.42 |
| U | 0.9 | 1.6 | | | 0.88 | | 1.29 | 0.62 |
| La | 25.0 | 53.9 | 19.5 | 6.5 | 35.97 | 15.2 | 36.28 | 18.3 |
| Ce | 49.3 | 112.0 | 41 | 15.5 | 77.74 | 37.7 | 65.01 | 34.0 |
| Nd | 23.1 | 58.9 | 22 | 13.5 | 41.36 | 20.8 | 27.91 | 19.9 |
| Sm | 4.6 | 10.1 | 5.3 | | 8.32 | | 5.37 | 4.51 |
| Eu | 1.58 | 3.16 | 1.6 | 1.35 | 2.52 | | 1.75 | 1.52 |
| Gd | 4.6 | 8.6 | 4.6 | | 6.59 | | 5.29 | 4.64 |
| Tb | | 1.1 | | | 0.88 | | 0.82 | 0.72 |
| Dy | 3.9 | 5.3 | 3.7 | 4.4 | 4.37 | | 4.84 | 4.40 |
| Ho | | 0.96 | | | 0.70 | | 0.96 | 0.88 |
| Er | | 2.3 | 1.8 | 2 | 1.78 | | 2.76 | 2.50 |
| Tm | | 0.3 | | | 0.25 | | 0.401 | 0.35 |
| Yb | 2.0 | 2.0 | 1.42 | 1.7 | 1.44 | | 2.57 | 2.27 |
| Lu | 0.30 | 0.24 | | | 0.11 | | 0.39 | 0.33 |
| Nb/La | 1.28 | 0.95 | 1.36 | 3.97 | 1.31 | 2.32 | 1.96 | |
| Nb/U | 35.7 | 31.9 | | | 53.7 | | 55.2 | |
| Zr/Sm | 35.2 | 15.9 | | | 18.5 | | 37.1 | |
| $^{87}Sr/^{86}Sr$ | 0.703010 | 0.703555 | 0.703135 | 0.703360–0.703627 [1] | 0.70445 [2] | 0.704092 | 0.70284 | 0.7040.56 |
| $^{143}Nd/^{144}Nd$ | 0.513110 | 0.512994 | 0.512955 | 0.512942–0.512996 [1] | 0.51290 [2] | 0.512846 | 0.51305 | 0.512828 |
| $^{206}Pb/^{204}Pb$ | 18.4910 | | 18.702 | | | 18.528 | 18.741 | 18.1677 |
| $^{207}Pb/^{204}Pb$ | 15.6140 | | 15.570 | | | 15.566 | 15.541 | 15.5389 |
| $^{208}Pb/^{204}Pb$ | 38.6370 | | | | | 38.598 | 38.145 | 38.3893 |

[1] Based on four separate determinations of isotope ratios [101]. [2] Initial isotope values $(^{87}Sr/^{96}Sr)_I$ and $(^{143}Nd/^{144}Nd)_I$ calculated using the average $^{40}Ar/^{39}Ar$ plateau age of 52.74 Ma [133].

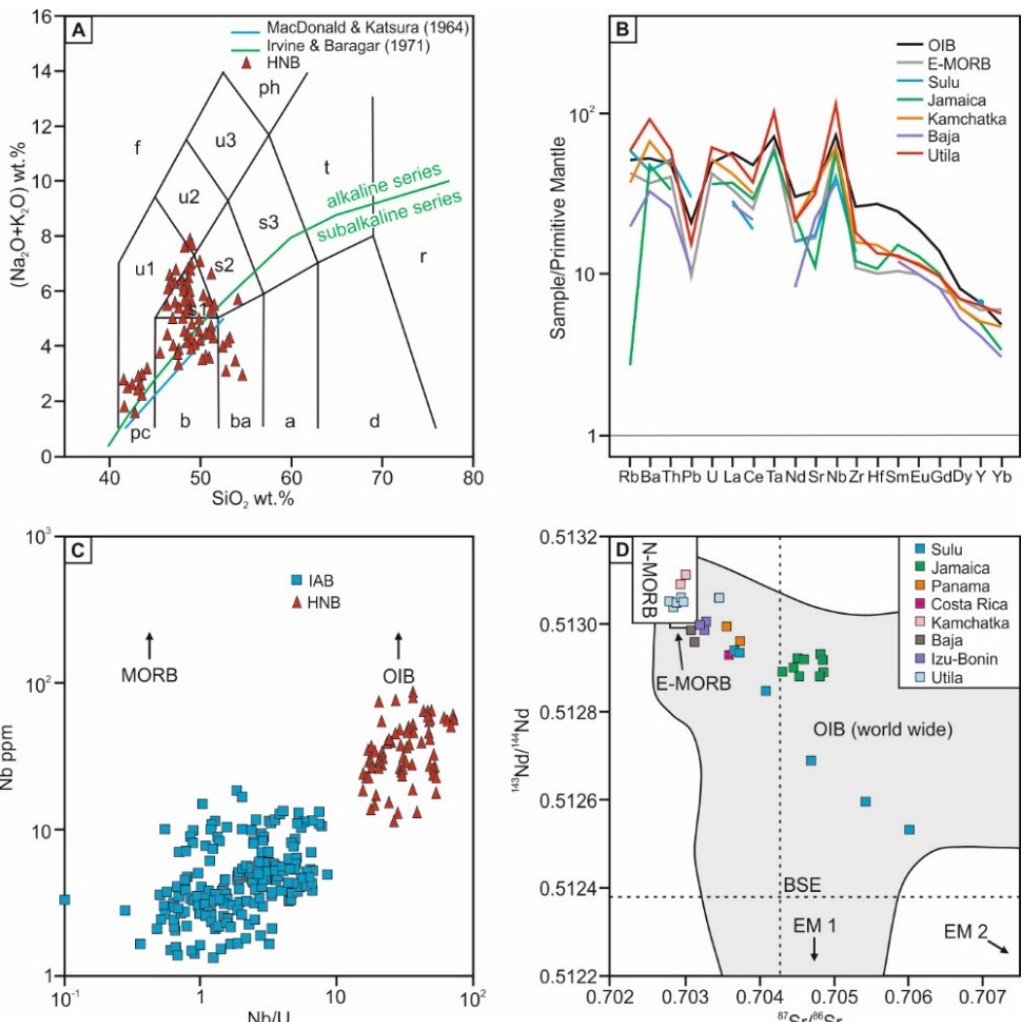

**Figure 3.** (**A**) Total alkalies-silica classification diagram [134] for high-Nb basalts [52,55,56,64,129–132,135–138]. Boundaries between the alkaline and the subalkaline volcanic series—MacDonald and Katsura (1964) and Irvine & Baragar (1971)—are, respectively, from [139,140]. Compositional fields: pc—picrobasalt, b—basalt, ba—basaltic andesite, a—andesite, d—dacite, r—rhyolite, s1—trachybasalt, s2—basaltic trachyandesite, s3—trachyandesite, t—trachyte, u1—basanite or tephrite, u2—phonotephrite, u3—tephriphonolite, ph—phonolite, f—foidite [134]. (**B**) Trace element compositions of high-Nb basalts from Honduras [131], Kamchatka [52] and Sulu arc [130] compared to low-degree enriched mid-ocean ridge basalt (E-MORB) melt [141] and average ocean island basalt (OIB) [142]. (**C**) Nb versus Nb/U diagram for basaltic rocks from volcanic arcs modified after [58]; data sources: high-Nb basalts (HNB) [52,55,56,64,129–132,135–138], arc basalts [52,136,143–150]. (**D**) Sr and Nd isotopes in high-Nb basalts from Sulu arc [130], Jamaica [129], Panama and Costa Rica [56], Kamchatka [52], Baja California [55], Izu-Bonin [132] and Utila, northern Honduras [131]. Average composition of low-degree E-MORB melt is from [141]. Fields of worldwide OIB and normal mid-ocean ridge basalts (N-MORB) are from [52,151], respectively.

However, rare HNBs with lower alkalinity also have been reported from the Izu-Bonin arc [132] mafic lavas with $TiO_2 > 1.5$ wt.%, $Nb > 20$ ppm, $Nb/U > 10$ and $Nb/La > 1$. High-Nb basalts can be aphyric or contain phenocrysts and micro-phenocrysts (Figure 4) of olivine, clinopyroxene (augite to Ti-augite), plagioclase, amphibole (pargasite or kaersutite), Ti-magnetite and, in some cases, ilmenite [52,58,59,131,132]. Major element variations identify HNBs as picrobasalts, basalts and trachybasalts that straddle the line between sub-alkaline and alkaline volcanic rocks (Figure 3A), which is also consistent with their elevated $TiO_2$ and variable MgO and $Al_2O_3$ contents in respect to typical arc

basalts [56,58,60,131]. Trace element compositions of HNBs, especially their enrichment in large-ion lithophile (LIL) and HFS elements, are broadly similar to OIB and E-MORB (Figure 3B). This general similarity has led several authors to suggest HNB derivation from enriched (plume-influenced) mantle sources [127,130,148,152]. However, OIB-type lavas tend to have, on average, higher concentrations of middle and heavy rare earth elements (MREE and HREE), as well as Zr and Hf (Figure 3B). Other distinct geochemical features of HNB include elevated Nb/U (>10), Nb/La (>1), Zr/Sm (>25) and low Ba/Nb and Ba/La ratios [56–60,153]. HNBs are clearly distinguished from volcanic arc basalts (both calc-alkaline and island-arc tholeiites) on the Nb–Nb/U graph (Figure 3C), where they also display compositional differences with mid-ocean ridge basalts (mean MORB with Nb = 5.24 ppm and Nb/U ratio of ~44 [154]) and OIB (similar Nb contents at broadly higher Nb/U ratios [56,58,60]). Recently, Kepezhinskas and Kepezhinskas [153] proposed the existence of two principal types of high-Nb basaltic magmas in volcanic arcs, which can be best distinguished by their Sr–Nd isotope variations (Figure 3D). Type 1 HNBs (exemplified by Kamchatka and Honduras; Table 3) are characterized by highest Nb contents (30–70 ppm) and Nb/LREE (e.g., Nb/La) and Nb/LILE (e.g., Nb/U) ratios coupled with the most depleted (broadly similar to MORB and frequently more depleted than spatially and temporally associated "normal" arc lavas [52]) Sr and Nd isotopic signature, similar to the Pacific MORB (Figure 3D). Pb isotopes in isotopically depleted type 1 HNBs also suggest derivation from N-MORB-type mantle sources, although a small (1–3%) involvement of subducted sediment (roughly comparable to EM-I and EM-II lead isotope signature) was identified in both Kamchatka and Honduras HNB suites [52,131]. Type 1 HNBs are typically found (commonly in close spatial and temporal association with adakites) in volcanic arcs associated with subduction of young oceanic crust or oceanic spreading center (ridge) [55–60,65,129,137]. Type 2 HNBs with an enriched Sr–Nd–Pb isotopic signature (Sulu arc, Baja California, Izu-Bonin arc, western Trans-Mexican Volcanic Belt) plot near the Sr–Nd isotope values typical of OIB (Figure 3D), suggesting involvement of enriched (plume-type) mantle source [127,130,152]. This is achieved either through influx of asthenospheric mantle through slab windows after cessation of active subduction (e.g., Baja California [127]) or mixing between enriched (plume-type) and depleted components within a compositionally heterogeneous mantle wedge (e.g., Sulu arc [130,155]). A variant of the latter model involves direct mixing between low-HFSE and high-HFSE melts, either in the mantle wedge or in the overlying arc crust [156]. Type 2 HNBs are frequently associated with extensional features (small-scale, intra-arc and back-arc rifts, grabens, faults dissecting volcanic arc fronts, etc. [127,130,135,152]), implying discontinuities in the subduction zone's geometry, discontinuity in the subducted oceanic slab or a general re-arrangement of subduction zone regimes (slab window formation, slab break-off, subduction pause or termination and other related geodynamic phenomena).

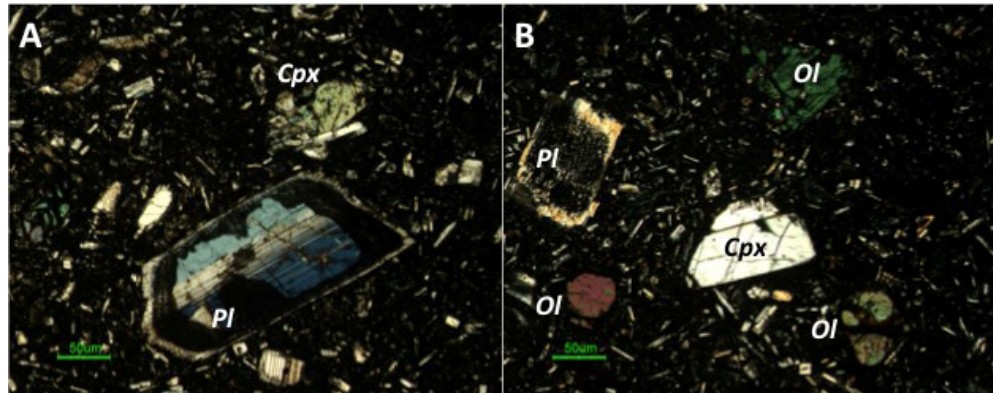

**Figure 4.** Petrographic features of porphyritic high-Nb basalts from the Utila Volcanic Center (northern Honduras). (**A**) clinopyroxene-plagioclase-phyric variety, (**B**) olivine-pyroxene-plagioclase-phyric variety. In both cases, groundmass is composed of plagicoclase microliths and partially altered glass. Ol—olivine, Cpx—clinopyroxene (Ti-augite), Pl—plagioclase. Detailed petrographic descriptions are available in [131].

The close association of type I HNB with adakites and young oceanic crust leads to an early suggestion that the interaction of the depleted mantle wedge with adakite melt may generate a hybrid mantle source enriched in HFSE and capable of producing high-Nb basalt magmas [57–60]. This process involves the dehydration melting of garnet amphibolite in hot subduction zones followed by the refertilization of the overlying mantle wedge with incompatible elements [51,54,57–60]. If slab melting occurs under water-deficient conditions (for example, during the amphibolite–eclogite transition in the slab), the presence of low-water fluid will promote titanite stability in the residual slab [157]. These residual titanites preferentially retain Ta, thus liberating Nb into a slab melt [158]. In addition, rutile may also be dissolved in felsic melt under these conditions, further enriching it in Ti, Nb and, to a certain extent, Ta [159]. Rutile appears to favor Ta over Nb with $D_{Nb}$ always lower than $D_{Ta}$ for each rutile/melt pair in experiments with total pressure >1.5 GP [159]. The residual titanite stability in the eclogitic slab is, therefore, the key to slab melt–peridotite interaction in hot subduction zones, as it fractionates Nb from Ta and results in adakitic melts that will be enriched in Nb. For example, an exhumed ultra-high pressure metamorphic terrane in North Qaidam (NW China) contains pods of eclogite juxtaposed against two types of leucosome, garnet-bearing and garnet-free, and a garnetite residue [158]. Garnetite residue contains rutile and titanite, which retains a substantial portion of the HFSE in the slab. Both garnet-free and garnet-bearing leucosomes contain two generations of rutile, being relict and neocryst (terminology after [158]). While relict rutile was inherited from an eclogitic source, newly crystallized rutile reflects the (limited) dissolution of rutile in a leucosome (essentially an adakitic melt) and strongly enhances its adakitic signatures. This felsic melt will also be enriched in Nb in comparison with "normal" adakites [158]. At Harts Range, Central Australia, a shallower slab (8–10 kbar) was melted to produce siliceous melt with elevated Nb/Ta ratios [160]. The dehydration melting of metabasalt generated garnet–clinopyroxene–titanite residue with titanite preferentially retaining Ta over Nb and suggesting that resulting partial melts may, indeed, carry a Nb-enriched adakitic signature.

In general, HFSE are not mobile in the presence of water-dominated volatiles (including slab-derived aqueous fluids), no matter how large the amount of volatiles [11,16] and references therein. This relative HFSE immobility in subduction zones controlled exclusively by fluid fluxes is caused by the insolubility of rutile, sphene and zircon in supercritical fluids within the range of pressures and temperatures typical of modern subduction zones [11,16,33]. Recently published experimental results, however, suggest that rutile's solubility can increase by several times in presence of Cl [161]. Some adakitic magmas have been reported to contain high concentrations of halogens, especially Cl [162]. Introduction of slab-derived silicic melt into the overlying mantle causes pervasive metasomatism and

formation of various metasomatic phases such as amphibole and phlogopite [57,163]. As the latter are Ti-bearing phases, they also contain elevated concentrations of Nb and Ta. Such HFSE-enriched amphiboles and phlogopites have been reported from lithospheric mantle xenoliths in some supra-subduction (Ichinomegata in Japan, Kamchatka and Papua New Guinea) and intraplate (Kerguelen, Eifel, Mongolia, Baikal rift zone) settings [58,164]. Experimental data also suggest that, along with amphibole and phlogopite, new metasomatic assemblages in slab melt-hybridized, sub-arc mantle sources include other Ti-bearing phases such as Ti-magnetite, titanite and rutile [57,58,60,163]. The presence of adakite melt in supra-subduction mantle xenoliths from Kamchatka, Austral Andes, Patagonia and SE Spain (Tallante) is well-documented by felsic interstitial glasses, veinlets and veins with distinctive adakitic geochemistry (high Sr/Y and La/Yb ratios) [58,165–167]. The melting of such adakite-hybridized, HFSE (Nb)-rich mantle wedge sources is capable of generating alkaline basaltic magmas with distinct Nb enrichments and variable or mixed (MORB to OIB) isotopic signatures [60]. At the same time, the assimilation of peridotitic mineral assemblages (pyroxene, olivine and spinel) by ascending silicic melt results in the formation of Mg-rich adakite (high-MgO andesite) from primary low-Mg slab melt (as suggested be typically higher MgO contents in erupted adakites in comparison with experimental slab melts [47,49–51,54,55,58,80,91]). It was also proposed [60] that adakite–mantle wedge interaction may result in a distinctive adakite–high-Mg adakite (magnesian andesite)–high-Nb basalt (Nb-rich shoshonite/lamprophyre) geochemical triad that can be used for the recognition of paleo-tectonic environments related to slab melting and subduction of young and hot oceanic crust.

## 4. Examples of Adakites and High-Nb Basalts Hosting Cu–Au Mineralization

*4.1. Russian Far East*

4.1.1. Cenozoic Kamchatka Arc

The Cenozoic Kamchatka arc consists of three sub-parallel volcanic belts (from west to east—Sredinny Range, Central Kamchatka Depression, Eastern Volcanic Front) composed of lavas and pyroclastic products of tholeiitic, calc-alkaline and alkaline (shoshonitic to ultra-potassic) volcanism intruded by differentiated ultramafic to felsic intrusions [52,168]. The southern segment of the Kamchatka arc (Figure 5) was formed in response to the subduction of the Cretaceous Pacific plate [52]. The northern segment (Figure 5) records the north-westward subduction of young (<15 Ma), hot oceanic crust formed in a back-arc spreading center of the Komandorsky Basin in the western Bering Sea [169,170]. Products of the magmatic response to young and hot subduction include calc-alkaline, high-K calc-alkaline and shoshonitic volcanic series along with adakites and high-Nb basalts [38,45,52,171]. Kamchatka adakites have been first described by the senior author of this study followed by several papers published by the informal consortium of Russian (Institute of Lithosphere headed by the Corresponding Member of Russian Academy of Sciences Professor Nikita Bogdanov), American (University of South Florida—Marc Defant and Alfred Hochstaedter; University of Alabama—Mark Drummond; Miami University, Ohio—Elisabeth Widom), French (Université Bretagne Occidentale—Rene Maury, Herve Bellon, Christian Honthaas) and British (Open University, Milton Keynes—Frank McDermott and Chris Hawkesworth) researchers [45,50,52,57,58,60,91,95,171–173]. Later, adakites and, in some cases, high-Nb basalts (e.g., Bakening volcano) were reported from the central [124,174–176] and southern [52,177] segments of the Kamchatka arc. Valovayam adakites are dated at 6 Ma (sample VAL4 in [52]), and HNB dikes in the vicinity of adakite VAL4 yielded K-Ar ages of ~7 Ma [52,171]. High-Nb basalts from the Valovayam volcanic field (northern segment) and Bakening volcano (southern segment) contain abundant mantle xenoliths showing extensive evidence (adakite veinlets, LREE-enriched metasomatic amphibole and clinopy-roxene, bulk enrichments in fluid-immobile trace elements, such as Ta and Hf, etc.) for hybridization by slab melts under the P-T conditions of sub-arc mantle wedge [57,58,60,173]. Some of the adakite–hybridized mantle nodules from Bakening and Valovayam exhibit

pronounced Au (as well as Pt and Pd) enrichments interpreted as signatures of advanced metasomatic reactions in the sub-arc mantle wedge beneath Kamchatka [60,95,173].

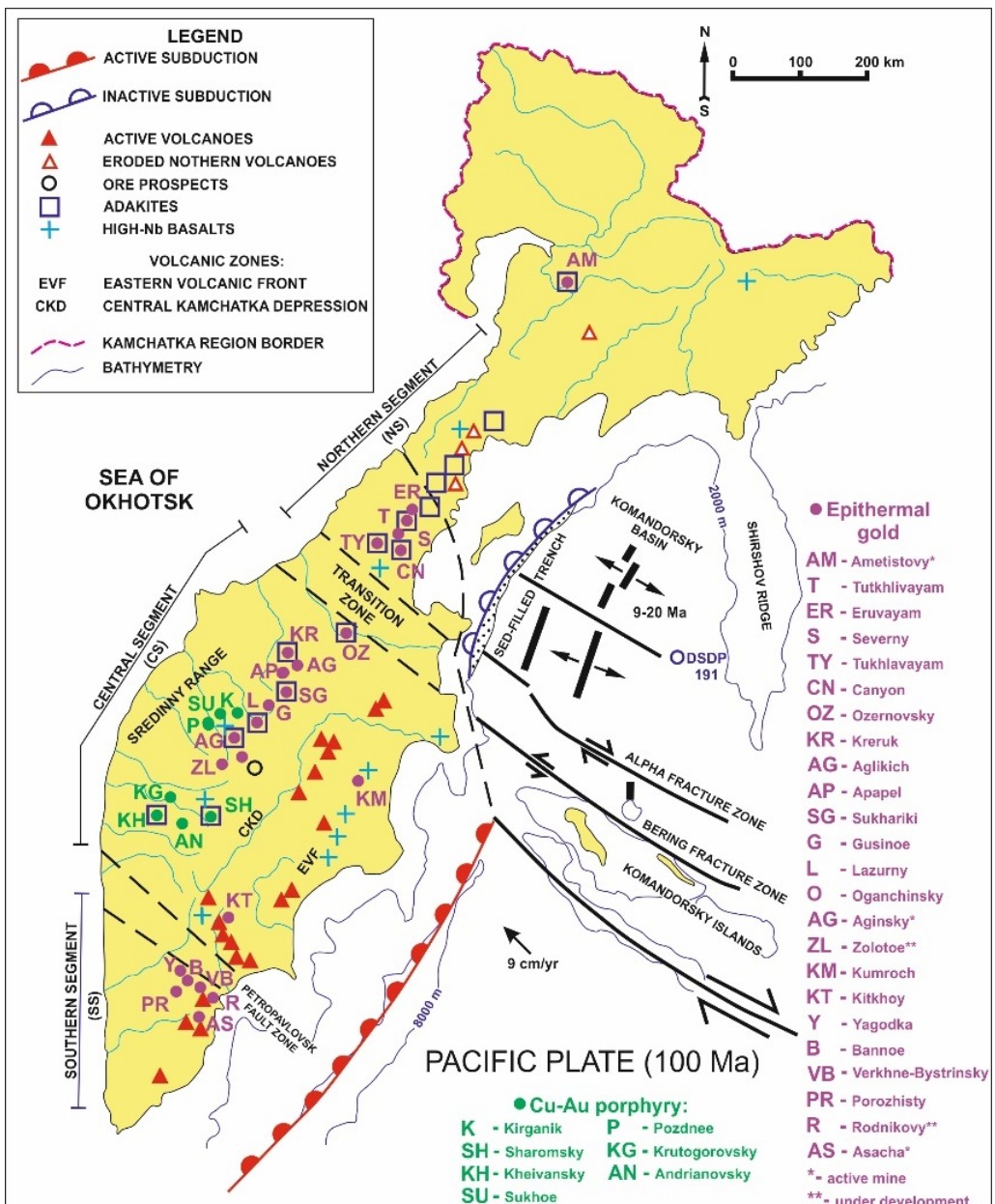

**Figure 5.** Tectonic setting of copper–gold porphyry and epithermal gold deposits in Kamchatka in reference to regional distribution of adakites and high-Nb basalts.

Kamchatka adakites are porphyritic rocks containing amphibole, andesine-labradorite-bytownite plagioclase and clinopyroxene as main megacrystic and phenocrystic phases [38,45,52]. Groundmass is composed of clinopyroxene, plagioclase and opaque minerals. Zoned amphibole phenocrysts are high-Al pargasites (Figure 6a; Table 4). Clinopyroxenes occasionally display high Cr contents (up to 1.5 wt.% $Cr_2O_3$) and are classified as diopsides (Figure 6b; Table 4). Plagioclase phenocrysts typically have Ca-rich (bytownite to labradorite) compositions (Figure 6c). The common presence of high-Cr (texturally corroded) diopside cores surrounded by high-Cr pargasitic rims and Cr-spinel inclusions in both clinopyroxene and pargasitic amphibole suggests either the derivation of Kamchatka adakites from a more mafic (high-Mg andesite) parental magma or extensive interaction with mantle wedge peridotites [45,52]. The presence of small (1–3 cm) ultramafic nodules in North Kamchatka

adakites (Valovayam volcanic field) appears to be supportive of the slab melt–peridotite interaction hypothesis [52,57,58]. Most rocks are fresh, although rims in some amphibole phenocrysts in adakites from North Kamchatka (Tymlat location) are replaced with chlorite. Accessory minerals include titano-magnetite, apatite, rare zircon and ilmenite, as well as isolated sulfides (pyrite and chalcopyrite).

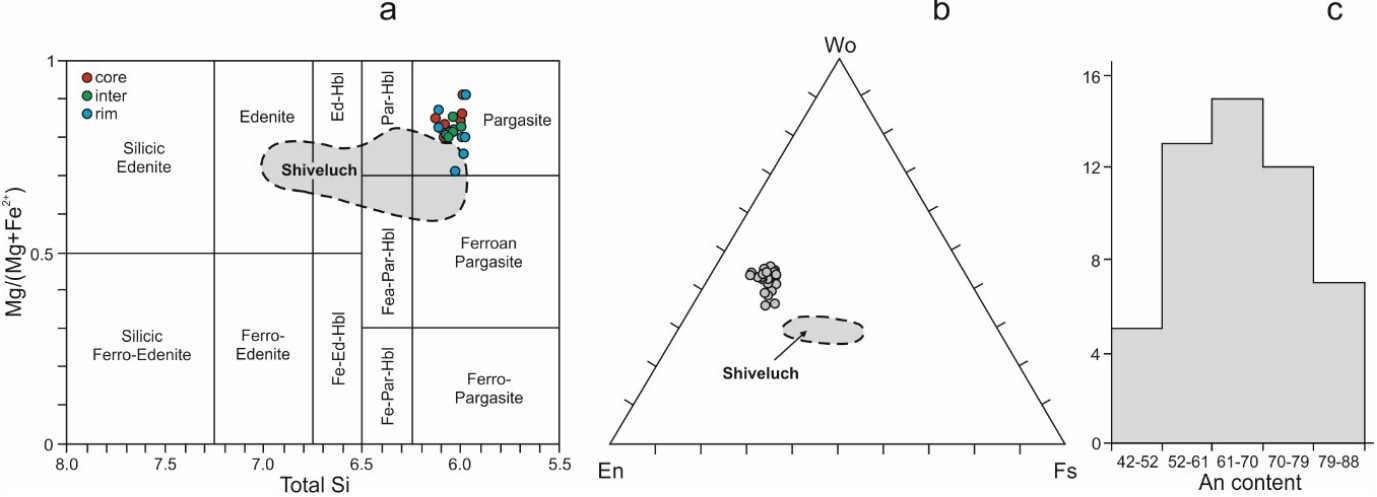

**Figure 6.** Chemical composition of phenocrysts in northern Kamchatka adakites: (**a**) amphibole, (**b**) clinopyroxene and (**c**) plagioclase. Field of mineral compositions from the Shiveluch adakites is after [175].

**Table 4.** Representative mineral compositions (wt.%) of Northern Kamchatka adakites.

| Sample | Val-4 | Val-4 | Val-4 | Val-4 | Val-29 | Val-29 | Val-29 | Val-29 |
|---|---|---|---|---|---|---|---|---|
| Mineral | Amp | Amp | Pl | Pl | Cpx | Cpx | Amp | Pl |
| $SiO_2$ | 42.37 | 41.30 | 47.84 | 50.87 | 47.34 | 52.56 | 40.89 | 46.36 |
| $TiO_2$ | 2.71 | 2.61 | ND | ND | 1.97 | 0.37 | 3.47 | ND |
| $Al_2O_3$ | 13.44 | 14.36 | 33.50 | 31.42 | 9.69 | 1.86 | 14.81 | 33.96 |
| FeO | 9.71 | 12.28 | 0.51 | 0.62 | 6.70 | 8.46 | 10.87 | 0.44 |
| MnO | 0.34 | 0.20 | ND | ND | 0.34 | 0.37 | 0.47 | ND |
| MgO | 15.03 | 13.39 | ND | ND | 13.52 | 15.83 | 13.73 | ND |
| CaO | 11.07 | 11.25 | 16.01 | 13.54 | 19.78 | 20.17 | 10.92 | 16.53 |
| $Na_2O$ | 2.14 | 2.15 | 2.41 | 3.80 | 0.83 | 0.27 | 2.61 | 1.98 |
| $K_2O$ | 0.20 | 0.21 | 0.06 | 0.06 | 0.00 | 0.00 | 0.42 | 0.01 |
| BaO | ND | ND | 0.17 | 0.03 | ND | ND | ND | 0.08 |
| Total | 97.01 | 97.75 | 100.35 | 100.34 | 100.17 | 99.89 | 98.19 | 99.36 |
| Mg-number | 75.4 | 67.7 | - | - | 78.3 | 77.0 | 71.2 | - |
| An content | - | - | 78.3 | 66.1 | - | - | - | 82.1 |

Mineral compositions determined by microprobe following methods described in [99]. Mg-number: $100 \times Mg/(Mg + Fe)$ (at.%). ND—not determined. Amp—amphibole, Cpx—clinopyroxene, Pl—plagioclase.

Kamchatka adakites range in composition from basaltic andesite to andesite [38,45,52]. Adakitic dacites so far had been detected only among the Miocene to Pleistocene eruptive products of the Bakening volcanic center in southern Kamchatka [52]. Adakites in the northern segment frequently display high MgO contents consistent with slab melt–peridotite hybridization model [45,52]. Adakites from the Kamchatka arc are characterized by a pronounced enrichment in Sr and a depletion in Y and HREE (Yb) typical of adakites worldwide (Figure 7). Their Sr–Nd–Pb isotopic characteristics are consistent with melting of the basaltic source isotopically analogous to the Pacific MORB with minor (1–3%) additions of subducted sediment or sediment-derived fluid [52].

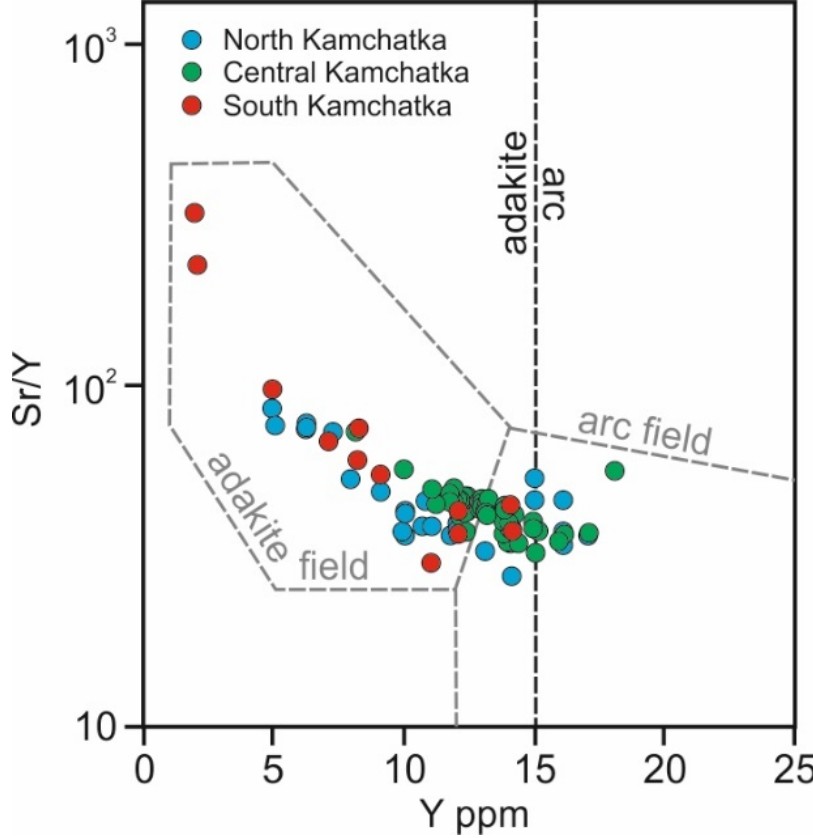

**Figure 7.** Variations of Sr/Y versus Y (ppm) in adakites and associated volcanic rocks from Kamchatka. Data sources: North Kamchatka [38,45,52], Central Kamchatka [174–176], South Kamchatka [52,177].

Adakites within the northern segment of the Kamchatka volcanic arc were generated via the dehydration melting of garnet amphibolite in the young (<15 Ma) subducted Komandorsky Basin oceanic crust [38,50,52]. Based on their high-Mg character and presence of adakitic veinlets (Sr/Y > 200) in ultramafic xenoliths from the Valovayam volcanic field [58], North Kamchatka adakites interacted extensively with mantle wedge peridotite upon their ascent to the surface [45,50,57,58]. Adakites in central Kamchatka appear to be products of the melting of the tectonically torn, old (Mesozoic) Pacific slab "warmed up" by the intense flow of hot mantle along its edge in a major transoceanic transform fault environment [124]. Mantle xenoliths brought up in modern explosive products of the Sheveluch volcano also record intense mantle–melt interactions [176] resembling those in northern Kamchatka xenoliths.

Kamchatka, like many Cenozoic magmatic arcs, is richly endowed with base and precious metal mineralization [178,179] and is home to numerous small to medium-size epithermal gold deposits and operating mines (Aginsky, Asachinsky, Ametistovy) with reserves and resources in excess of 1 million ounces of gold equivalent (Table 5). Adakite-hosted epithermal gold mineralization in the Sredinny Range of the Kamchatka volcanic province is paralleled (Figure 5) by Cu–Au(Mo) porphyry deposits and showings associated with volcanic (lavas, pyroclastics) and sub-volcanic/intrusive (dikes, sills, plugs and small plutons) shoshonites and high-Nb basalts [52,179,180]. Epithermal Au–Ag mineralization within Miocene-Pliocene adakitic volcanic edifices (including several notable nested caldera environments such as Ametistovy deposit in north Kamchatka and Verkhne-Sukhariki/Marina/Sukharikovsky Grebny cluster in Central Kamchatka; Figure 5; Table 5) is represented by quartz, quartz–carbonate and quartz–adularia veins, typically 1 to 2.5 m wide; argyllized shear zones (Tymlat deposit in Sredinny Range; Figure 5; Table 5); silicic to argyllic metasomatites; and sulfide-rich, stockwork-type zones (Sukhariki cluster in the Sredinny Range; Figure 5; Table 5). Both high- and low-sulfidation types

are well represented in the Kamchatkan epithermal heritage associated with Cenozoic adakitic magmatism. High-sulfidation systems are typically dominated by pyrite and chalcopyrite with minor sphalerite, galena, arsenopyrite, molybdenite and argentite in association with native gold and silver, electrum and various Au–Ag-sulfosalts (polybasite) and Ag-tellurides (hessite; Table 5). Both high- and low-sulfidation epithermal deposits and showings in Kamchatka are characterized by high Au grades (>5 g/t) and moderate (averaging in the 20–50 g/t Ag range) silver contents. Porphyry mineralization associated with shoshonites and high-Nb basalts is typically hosted in linear stockwork zones with moderate Cu (0.2–0.7%) and Au (0.1–0.5 g/t) grades supplemented occasionally with other metals such as silver, molybdenum, platinum and palladium (Table 5; [180]). In most cases, adakites and high-Nb basalts display clear structural and temporal association with Cu–Au porphyry and epithermal mineralization in Kamchatka, further strengthening earlier claims [60,82–84,87] about general genetic links between adakite magmas and precious and base metal deposits in subduction zone environments.

**Table 5.** Au–Cu (Ag, Mo) mineralization associated with adakite/HNB magmatism in Kamchatk.

| Deposits | Age (Ma) | Magmatism | Mineralization | Grades | Resources |
|---|---|---|---|---|---|
| Ametistovy | 38–41 Ma | Adakite nested caldera | 38 Qz veins (1–20 m) with Py, Mrc, Ccp, Sp, Gn, Tnt-Ttr, El. Argn. Pol | Au = 13.62 g/t; Ag = 84 g/t | Au—1.8 Moz; Ag—8 Moz |
| Ozernovsky | 7.6 Ma | Adakite nested caldera | Qz–Adl veins (1–11 m); silicic stockwork with | Au = 8.7–16.6 g/t | Au—2.7 Moz |
| Tutkhlyvayam | Miocene | Adakite, calc-alkaline andesite and dacite | Qz–Cb veins (22 in total) | Au = 9.6–11.2 g/t; Ag = 135–531 g/t | Au—1.6 Moz; Ag—36 Moz |
| Tymlat | Miocene | Adakite, granite porphyry | Argyllized shear zones (5–23 m) with Py+Apy; Py–Hem–Ccp silicic stockwork | Au~3 g/t; also Cu—0.36–0.5% | Au—0.7 Moz |
| Severny | Miocene | Adakitic diorite porphyry | Qz and Qz–Cb veins (1.5–4.3 m) and silicic (with Ccp, Py, Sp, Au, Hes, Tnt-Ttr) stockwork | Au = 23.2 g/t; Ag = 68.9 g/t | Au—0.41 Moz; Ag—1.2 Moz |
| Eruvayam | Miocene | Adakite porphyry and CA volcanics | 19 Qz–Cb (±Adl) veins (0.2–18 m) | Au = 8–10.6 g/t; Ag = 5–173.8 g/t | Au—1 Moz; Ag—31 Moz |
| Canyon | Miocene | Adakite porphyry | 30 Qz and Qz–Cb veins and stockwork zones (0.6–16 m) with Py, Gn, Ccp, Sp, Au, Hes, Argn, Tnt-Ttr, Pyr | Au = 2.2–281.6 g/t; Ag = 2.9–1450 g/t | Au—0.6 Moz; Ag—13.3 Moz |
| Kreruk | Pliocene | Adakite volcanics | Qz and Qz–Cb veins (0.5–4 m) with Au, Argn, Sp, Tnt-Ttr, Antm | Au = 10.9–93.3 g/t; Ag = 138.8–1109.1 g/t | Au—0.5 Moz; Ag—10 Moz |
| Apapel | Pliocene | Subvolcanic adakitic rhyolite | 11 Qz veins (0.2–8.8 m) with Au, Py, Mrc, Sp, Mol, Cin | Au = 12.6–102.1 g/t; Ag = 11.9–120.3 g/t | Au—0.5 Moz; Ag—3 Moz |
| Aglikich | Miocene- Pliocene | Adakite porphyry | 11 Qz–Cb–Adl–Zeo veins (0.3–7.5 m) with Au, Hes, Ccp, Tnt-Ttr, Mag, Mrc, Cv | Au = 7.1–63.8 g/t; Ag = 42.5–651.8 g/t | Au—0.5 Moz; Ag—3 Moz |
| Verkhne-Sukhariki | Miocene | Nested caldera with adakites | 18 Qz and Qz–Cb veins (0.1–5 m) within silicic to argyllic envelope with Au, Py, Mrc, Ccp, Tnt–Ttr | Au = 4.7–11.8 g/t; Ag = 3.9–11.4 g/t | Au—1.4 Moz |
| Marina | Miocene Pliocene | Nested caldera with adakites | Silicic metasomatites (4–25 m) with Qz, Cb, Adl, Adl–Cb–Qz and Qz–Cb veining (Au, Argn, Py, Ccp, Sp, Tnt–Ttr) | Au = 1.2–35 g/t; Ag = 1.6–115 g/t | Au—0.3 Moz; Ag—0.6 Moz |
| Sukharikovsky Grebny | Miocene | Nested caldera with adakites and ignimbrites | Multiple (13) Qz–Adl–Cb metasomatic zones (1–16 m) with Au, Ag, Argn, Ccp, Sp, Gn, Pol, Hes, Syl | Au = 7 g/t; Ag = 22.7 g/t | Au—0.9 Moz; Ag—5.2 Moz |
| Kirganik | 73.17 ± 0.54 Ma | Shoshonites with minor adakite and HNB | Metasomatic zones (20–40 m) with Ccp, Bn, Cct, Hem, Mag, Dg, Sp, Gn, Mer | Cu = 0.5–0.71%; Au = 0.5–0.75 g/t; Ag = 6–7 g/t; Pt+Pd = 1.8 g/t | Cu—905,000 t; Au—3.1 Moz; Ag—32.7 Moz; Pt + Pd—2.02 Moz |

**Table 5.** *Cont.*

| Deposits | Age (Ma) | Magmatism | Mineralization | Grades | Resources |
|---|---|---|---|---|---|
| Malachitovy (Lavlinsky ore field) | Miocene | Calc-alkaline andesite and dacite porphyry with minor adakite | Metasomatic and linear stockwork zones with Ccp, Bn, Cct, Dg, Hem, Mag | Cu = 0.17–0.47%; Au = 0.02 g/t; Ag~1.3 g/t; Mo~0.014% | Cu—3,539,000 t; Au—17.2 Moz |

Data sources: [178–180]; P.Kepezhinskas, unpublished data. Mineral abbreviations: Qz—quartz, Cb—carbonates, Adl—adularia, Zeo—zeolite, Hem—hematite, Py—pyrite, Mrc—marcasite, Mag—magnetite, Apy—arsenopyrite, Ccp—chalcopyrite, Cv—covellite, Cct—chalcocite, Dg—digenite, Tnt–Ttr—tennantite–tetrahedrrite, Sp—sphalerite, Gn—galena, Antm—antimonite, Mol—molybdenite, Cin—cinnabar, Argn—argentite, Pol—plybasite (Ag–Cu–Sb–As sulfosalt), Hes—hessite ($Ag_2Te$), Slv—sylvanite (Au, Ag)$Te_2$, Pyr—pyrargyrite ($Ag_3SbS_3$), Mer—merenskyte (Pt, Pd) (Te, $Bi)_2$, El—electrum, Au –native gold, Ag—native silver, S—native sulfur. Moz—million ounces.

### 4.1.2. Mesozoic Stanovoy Suture Zone

The Stanovoy Suture Zone (SSZ) marks the major tectonic boundary between the Aldan Shield (crystalline exposure of the Siberian Craton) and the Central Asian Orogenic Belt (CAOB) (Figure 8a). The SSZ records multiple episodes of paleo-ocean openings and closures in the Late Proterozoic and Paleozoic, culminating in the final collision episode associated with the Mesozoic (Jurassic to Early Cretaceous) closure of the Mongol-Okhotsk ocean [181]. Post-collision Cretaceous (114–117 Ma) adakites intrude PR to MZ accreted terranes of the central segment of the SSZ [181]. Adakites in the central SSZ form extensive lava sequences, pyroclastic deposits and dikes in differentiated ultramafic-mafic plutons (Figure 8b) and are spatially associated with lamprophyres resembling typical high-Nb basalt compositions [181].

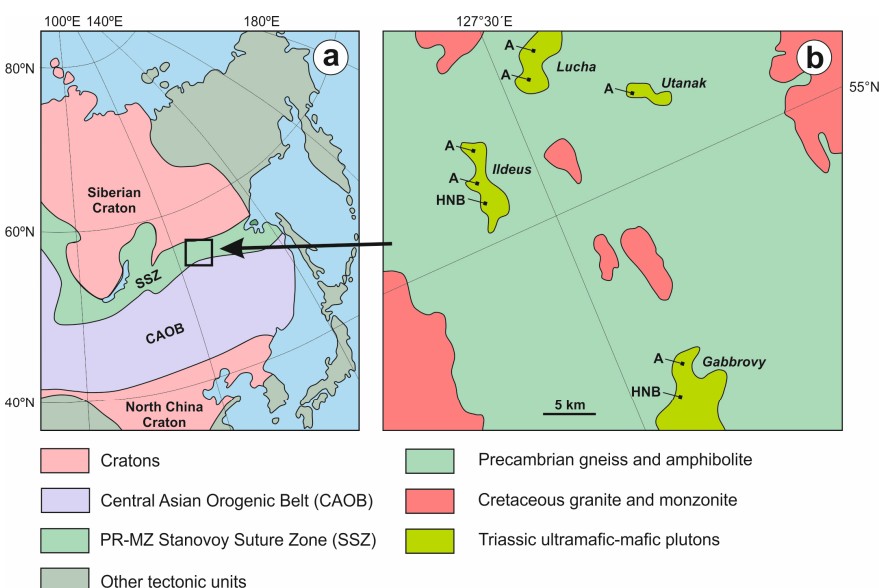

**Figure 8.** (**a**) Regional tectonic scheme showing position of Stanovoy Suture Zone (SSZ) in reference to Precambrian cratons, major orogenic belts (CAOB) and other tectonic units and (**b**) simplified geologic map of the central portion of the SSZ. Both maps are modified from [181]. A—adakite dikes, HNB—high-Nb basalt and lamprophyre dikes.

Adakite dikes and lavas in the central SSZ are aphyric to strongly porphyritic rocks (Figure 9). Phenocryst assemblages in adakites include hornblende, plagioclase, clinopyroxene and potassic feldspar of possible igneous origin (Table 6). Some adakitic dacite compositions (LU-9 in Table 7) contain quartz and biotite. Hornblende in strongly porphyritic SSZ adakites occurs as megacrysts, phenocrysts and micro-phenocrysts (Figure 9), which, together with strong optical and chemical zoning of larger hornblende crystals in respect to Al and Ti, suggests the polybaric fractionation of primary SSZ adakite melts in crustal magmatic conduits [181].

Stanovoy adakites display wide variations in some major oxides ($SiO_2$ = 57.89–72.16 wt.%; total $Fe_2O_3$ = 1.31–7.42 wt.%; $TiO_2$ = 0.13–0.83 wt.%; CaO = 1.36–8.55 wt.%) and trace elements (Ni = 12–237 ppm; Ba= 336–5777 ppm; Nb = 0.31–11.5 ppm; Th= 1.31–15.81 ppm; U = 0.05–3.54 ppm; Table 7), suggesting that the SSZ adakite series underwent variable degrees of fractionation of hornblende, plagioclase and magnetite [181]. All Stanovoy adakites display HFSE-depleted primitive mantle-normalized trace element patterns typical of subduction-zone magmas coupled with variable LREE/HREE enrichments (La/Yb = 12–134; Table 7) as well as high Sr and low Y contents (447–830 ppm and 1.91–14.6 ppm, respectively; Table 7). SSZ adakites have variable, but generally high, Sr/Y ratios (33.5–413.7; Table 7) and plot into (and above) the adakite field in Sr/Y versus Y discrimination diagram (Figure 10A). Adakites from different sites in the central segment of the SSZ (Figure 8b) exhibit subtle, but statistically valid variations in Sr/Y ratios (Figure 10B) and Y contents (Figure 10C). Adakite dikes and veins intruding the Ildeus massif, which carries sulfide-native metal-intermetallic alloy Ni–Co–Cu–Au mineralization, exhibit the highest Sr/Y ratios and the lowest Y contents among adakite sites sampled within the central segment of the SSZ (Figure 10B,C). Ildeus dikes are also characterized by the presence of the most siliceous (rhyolitic) compositions among the SSZ adakites, possibly reflecting advanced magmatic differentiation trends observed in some members of the Stanovoy adakite magma series [181].

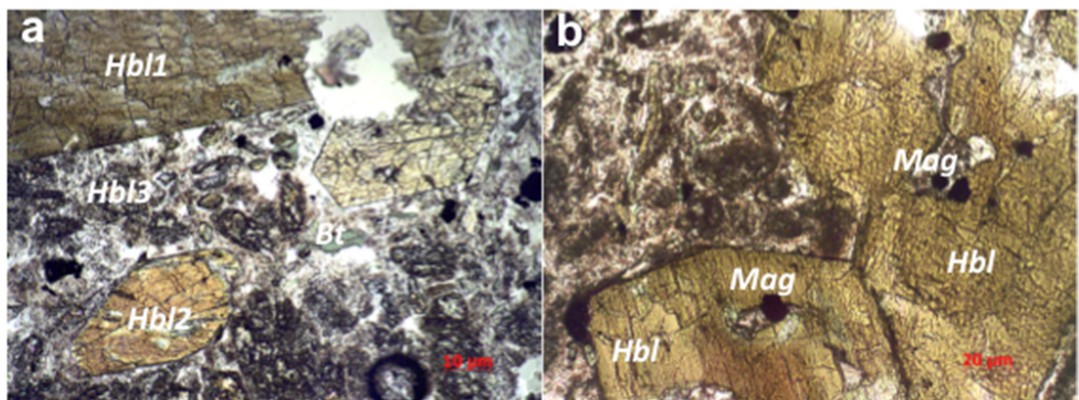

**Figure 9.** Petrographic features of Stanovoy adakites: (**a**) Porphyritic texture composed of three generations of hornblende (Hbl1—megacrysts, Hbl2—phenocrysts and Hbl3—groundmass) and biotite (Bt) phernocrysts; (**b**) zoned hornblende (Hbl) phenocrysts containing magnetite (Mag). Note that magnetite tends to occur in hornblende cores.

**Table 6.** Representative mineral compositions (wt.%) in the Stanovoy Suture Zone adakites.

| Sample | UTN-7 | UTN-7 | UTN-7 | UTN-7 | LU-9 | LU-9 | LU-9 | LU-9 |
|---|---|---|---|---|---|---|---|---|
| Mineral. | Amp | Amp | Amp | Pl | Amp | Amp | Bt | Pl |
| $SiO_2$ | 45.55 | 47.48 | 49.98 | 55.96 | 44.43 | 48.20 | 38.70 | 54.44 |
| $TiO_2$ | 1.51 | 0.80 | 0.58 | 0.00 | 2.53 | 1.45 | 3.17 | 0.13 |
| $Al_2O_3$ | 11.45 | 10.13 | 9.19 | 27.25 | 12.76 | 8.43 | 15.27 | 27.89 |
| $Cr_2O_3$ | 0.46 | 0.03 | 0.04 | 0.00 | 0.43 | 0.07 | 0.18 | 0.00 |
| FeO | 10.78 | 10.80 | 15.95 | 0.21 | 11.53 | 10.68 | 15.10 | 0.21 |
| MnO | 0.15 | 0.20 | 0.21 | 0.00 | 0.18 | 0.24 | 0.17 | 0.02 |
| MgO | 14.36 | 15.08 | 11.24 | 0.01 | 13.43 | 15.78 | 14.96 | 0.03 |
| CaO | 11.33 | 11.14 | 9.17 | 9.78 | 10.89 | 10.59 | 0.13 | 11.25 |
| $Na_2O$ | 1.98 | 1.94 | 1.47 | 5.49 | 2.31 | 1.67 | 0.79 | 5.09 |
| $K_2O$ | 0.41 | 0.23 | 0.26 | 0.15 | 0.31 | 0.21 | 7.97 | 0.12 |
| Total | 97.98 | 97.83 | 98.09 | 98.86 | 98.80 | 97.32 | 96.44 | 99.18 |
| Mg-number | 70.4 | 71.3 | 55.7 | - | 67.5 | 72.5 | 63.4 | - |
| An content | - | - | - | 49.1 | - | - | - | 54.7 |

Mineral compositions determined by microprobe following methods described in [99]. Mg-number: Mg/(Mg + Fe) (at.%). Amp—amphibole, Bt—biotite, Pl—plagioclase.

**Table 7.** Representative compositions of Cretaceous (117–114 Ma) adakites from the Stanovoy Suture Zone (SSZ).

| Location | Ildeus | Ildeus | Ildeus | Lucha | Gabbrovy | Gabbrovy | Utanak |
|---|---|---|---|---|---|---|---|
| Sample # | 042F | 016H | 047F | LU-9 | GAB-8 | GAB-8A | UTN-7 |
| $SiO_2$ (wt.%) | 62.40 | 72.16 | 71.48 | 68.85 | 62.01 | 60.73 | 57.89 |
| $TiO_2$ | 0.13 | 0.20 | 0.17 | 0.09 | 0.25 | 0.78 | 0.83 |
| $Al_2O_3$ | 12.40 | 13.35 | 13.27 | 15.42 | 15.23 | 15.78 | 14.80 |
| $Fe_2O_3$ | 4.17 | 2.05 | 1.78 | 1.31 | 5.16 | 5.24 | 7.42 |
| MnO | 0.08 | 0.03 | 0.03 | 0.04 | 0.09 | 0.09 | 0.13 |
| MgO | 1.82 | 1.34 | 1.77 | 1.06 | 3.33 | 3.48 | 4.16 |
| CaO | 8.55 | 2.20 | 2.51 | 1.36 | 4.59 | 4.50 | 6.12 |
| $Na_2O$ | 6.56 | 7.80 | 7.70 | 7.11 | 4.21 | 4.63 | 4.74 |
| $K_2O$ | 3.12 | 0.61 | 0.48 | 3.62 | 3.82 | 3.63 | 2.49 |
| $P_2O_5$ | 0.05 | 0.04 | 0.09 | 0.04 | 0.23 | 0.26 | 0.33 |
| LOI | 0.72 | 0.31 | 0.55 | 0.55 | 0.58 | 0.48 | 1.06 |
| Total | 100.00 | 99.92 | 99.85 | 99.45 | 100.00 | 99.99 | 99.97 |
| Cr (ppm) | 525 | 142 | 83.9 | 74.4 | 79.8 | 57.5 | 63.9 |
| Ni | 237 | 36.1 | 30.0 | 43.1 | 12.9 | 12.1 | 37.0 |
| Co | 28.3 | 16.6 | 13.4 | 10.9 | 17.5 | 17.1 | 21.4 |
| Cu | 21.0 | 32.5 | 28.3 | 42.9 | 18.7 | 20.2 | 28.1 |
| Zn | 155 | 31.6 | 23.7 | 131 | 72.8 | 70.9 | 86.0 |
| Sc | 5.67 | 2.14 | 0.82 | 1.92 | 7.98 | 7.37 | 10.9 |
| V | 23.6 | 36.2 | 15.9 | 8.12 | 96.6 | 83.2 | 105 |
| Cs | 0.16 | 0.11 | 0.09 | 0.25 | 1.78 | 1.41 | 0.62 |
| Li | 1.16 | 1.82 | 2.37 | 2.68 | 10.03 | 9.68 | 5.69 |
| Rb | 37.1 | 5.15 | 3.81 | 26.0 | 84.4 | 81.1 | 51.2 |
| Ba | 5777 | 598 | 336 | 2085 | 901 | 953 | 926 |
| Sr | 447 | 830 | 827 | 768 | 552 | 614 | 735 |
| Zr | 2.78 | 3.18 | 0.97 | 10.6 | 16.5 | 14.5 | 95.4 |
| Y | 1.91 | 3.88 | 2.00 | 2.40 | 14.6 | 13.9 | 12.9 |
| Nb | 0.31 | 2.25 | 0.54 | 1.26 | 11.5 | 10.4 | 8.84 |
| Ta | 0.20 | 0.40 | 0.11 | 0.40 | 0.79 | 0.42 | 0.60 |
| Hf | 0.10 | 0.15 | 0.034 | 0.03 | 0.92 | 0.76 | 2.56 |
| Th | 1.31 | 2.05 | 1.64 | 1.95 | 15.81 | 13.09 | 4.52 |
| U | 0.05 | 0.32 | 0.12 | 0.36 | 2.63 | 3.54 | 0.99 |
| La | 11.01 | 15.25 | 20.24 | 2.10 | 39.16 | 43.64 | 29.81 |
| Ce | 16.85 | 25.21 | 37.27 | 4.59 | 82.63 | 88.81 | 60.02 |
| Pr | 1.67 | 2.12 | 3.91 | 0.57 | 9.50 | 10.20 | 7.30 |
| Nd | 5.14 | 7.27 | 12.67 | 2.21 | 33.42 | 35.83 | 26.94 |
| Sm | 0.76 | 1.15 | 1.60 | 0.52 | 5.40 | 5.71 | 4.60 |
| Eu | 0.84 | 0.82 | 0.45 | 0.27 | 1.28 | 1.36 | 1.31 |
| Gd | 0.80 | 1.29 | 1.56 | 0.58 | 5.26 | 5.53 | 4.70 |
| Tb | 0.091 | 0.16 | 0.15 | 0.09 | 0.62 | 0.62 | 0.57 |
| Dy | 0.43 | 0.78 | 0.57 | 0.46 | 2.95 | 2.91 | 2.80 |
| Ho | 0.083 | 0.15 | 0.092 | 0.09 | 0.54 | 0.52 | 0.54 |
| Er | 0.23 | 0.40 | 0.25 | 0.21 | 1.56 | 1.47 | 1.52 |
| Tm | 0.033 | 0.053 | 0.025 | 0.03 | 0.21 | 0.18 | 0.20 |
| Yb | 0.21 | 0.31 | 0.15 | 0.17 | 1.31 | 1.12 | 1.28 |
| Lu | 0.034 | 0.045 | 0.022 | 0.02 | 0.19 | 0.16 | 0.18 |
| Au (ppb) | 9.01 | 75.38 | 5.53 | 93.72 | 100.65 | 97.27 | 134.85 |
| Sr/Y | 234.2 | 213.9 | 413.7 | 320 | 33.5 | 44.2 | 57.0 |
| La/Yb | 52.4 | 49.2 | 134.9 | 12.4 | 29.9 | 39.0 | 23.3 |

Major oxides by XRF, trace elements by ICP-MS. Details of analytical methods can be found in [182].

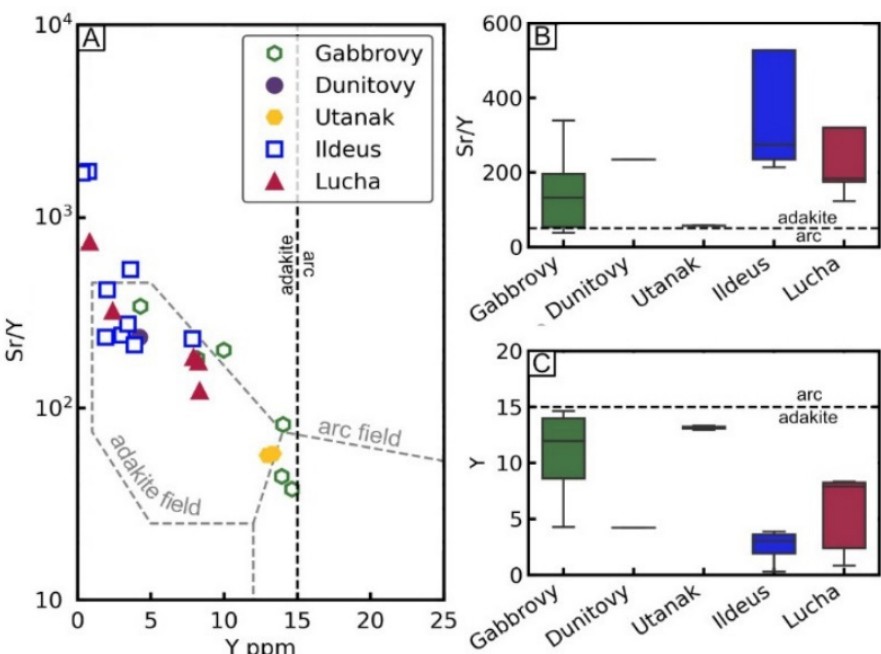

**Figure 10.** Sr/Y versus Y classification graph (**A**) and Sr/Y (**B**) and Y (**C**) variations in Stanovoy adakites. Compositional fields in (**A**) and boundaries in (**B**) and (**C**) are modified after [37,50]. The data in (**B**,**C**) are plotted showing the range and maximum/minimum of analytical values (excluding outliers), the 25th and 75th percentiles and the median (excluding outliers).

Stanovoy adakites contain diverse assemblage of metals and metal alloys included in amphibole, plagioclase and potassic feldspar phenocrysts. Our detailed Scanning Electron Microscope (SEM) study of Utanak adakite (sample UTN-7 in Table 7) identified the following metallic phases: Cu–Ag–Au alloys (Figure 11a), cupriferous silver (Figure 11b), native gold and platinum, Cu–Zn–Ni, Ag–Mo–Ni–Cu and Au–Ag–In–Zn alloys. Cu–Ag–Au alloys are compositionally similar to magmatic Au-rich intermetallic compounds in explosive volcanic rocks from the Lesser Khingan Range (Russian Far East) [182]. Native metals and alloys in the Utanak adakite are closely associated with non-stoichiometric silver chloride (Figure 11c), galena (Figure 11d), pyrite (Figure 11e), barite (Figure 11e,f), Pb–Fe and Pb–Fe–Sb sulfide phases, insizwaite (rare Pt-bismuthide—$PtBi_2$), cassiterite, zircon, allanite and F-apatite. The ubiquitous presence of barite and sulfide minerals attests to high sulfur activity during the magmatic differentiation of primary adakite melts [183]. Our new data on the distribution of native metals and intermetallic compounds in the SSZ adakite presented in this study, support earlier interpretations of adakites as metal-rich, S-saturated slab-derived or lower crustal melts [183,184]. The presence of Ag-chlorides as inclusions in the SSZ adakites suggests that Cl-rich fluids were also involved in adakite metallogenesis in the Stanovoy collision zone. This is consistent with experimental data on adakitic dacites from both Mount Hood and Mount St. Helens volcanoes in the Cascades, which suggest that both grade and tonnage of copper porphyry mineralization in volcanic arcs will be, to a certain extent, dependent on initial chlorine concentration in the adakite or calc-alkaline magma—" . . . increasing melt chlorine concentration from 500 to 3000 ppm imparts an almost eight-fold increase in copper extraction efficiency" [185]. The qbiquitous presence of barite ($BaSO_4$) and its frequent association with pyrite (Figure 11e,f) and other sulfides (Figure 11d) in the Utanak adakite suggests that high sulfur activity may effectively facilitate the transport of base (copper) and noble (gold, silver, platinum) metals by adakite magmas. The SSZ adakites display a wide range of bulk Au concentrations (5–134 ppb; Table 7), but most adakites from the Ildeus, Gabbrovy and Utanak sites have very high Au contents (>30 ppb) in comparison to modern volcanic arc magmas. Calc-alkaline basalts, andesites and dacites from the Kurile-Kamchatka volcanic province and Central American arc are characterized by gold concentrations of less than 10 ppb (independently of their

$SiO_2$ or MgO contents and other differentiation indices) with average Au content for arc magmas of around 3 ppm [186,187]. Gold values in the SSZ adakites partially overlap at the lower end with metal-rich adakitic dacites from the 1991 eruption of Pinatubo volcano in the Philippines (Au = 6–22 ppb; [183]) but in most parts display several fold enrichments of gold over the modern Philippine adakite magmas. The copper content of the SSZ adakites (18–42 ppm; Table 7) is similar to the lower end of the range of Cu values for Pinatubo adakite (26–77 ppm; [183]) and is statistically lower than the range of Cu concentrations in modern volcanic arc magmas (50–90 ppm; [188]). Our data appear to suggest that, other than their high water content, high oxidation state and $H_2O$–S–Cl composition of associated fluids [7,87–89], high initial Au concentrations in primary adakite magmas can play an important role in the transfer of precious metals to the upper crustal porphyry and epithermal environments.

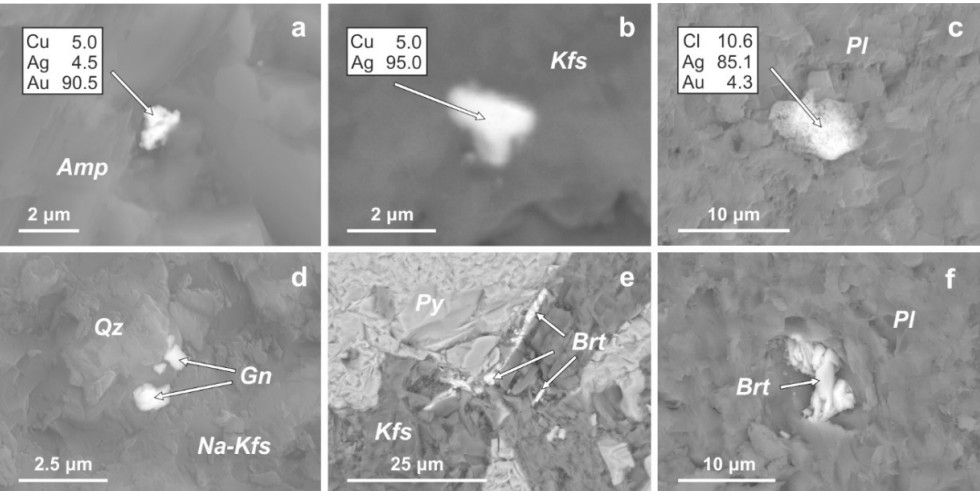

**Figure 11.** Cu–Ag–Au intermetallic compounds and associated sulfide and sulfate minerals in Stanovoy adakites: (**a**) Subhedral Cu- and Ag-bearing Au inclusion in amphibole (Amp); (**b**) euhedral Cu-bearing Au inclusion in K-feldspar (Kfs); (**c**) non-stoichiometric Au-bearing silver chloride inclusion in plagioclase (Pl); (**d**) euhderal galena (Gn) particles developed along the boundary between quartz (Qz) and Na–K-feldspar (Na-Kfs); (**e**) lamellar barite (Brt) crystals filling fractures between pyrite (Py) and K-feldspar (Kfs); (**f**) euhedral barite (Brt) inlcusions in plagioclase (Pl). All photos are SEM back-scatter electron (BSE) images.

### 4.2. Circum-Pacific Magmatic Arcs

Among the Circum-Pacific magmatic arcs, adakites, high-Nb basalts and associated Cu-Au porphyry and epithermal deposits are most common in the Philippines [189–198], Indonesia [189,199–209], the Papua New Guinea orogen and surrounding Melanysian volcanic arcs [210–216] as well as along the Andean magmatic arc [217–240] and the convergent margin of south-eastern Alaska [241–245].

### 4.2.1. Philippines

The Philippines island arc system records the complex tectonic interaction of three lithospheric plates: the Sundaland plate in the west, the Philippine Sea plate in the east and the Indo-Australian plate in the south (Figure 12; [189]). Arc volcanism in the Philippines is associated with the subduction of the Early Oligocene to Early Miocene South China Sea crust along the east-dipping Manila Trench, the Miocene Sulu basin along the NE-dipping Negros Trench and SE-dipping Sulu Trench and the Eocene Celebes Sea crust along the NE-dipping Cotabato Trench and Eocene West Philippine Sea lithosphere along the west-dipping East Luzon Trough and the Philippine Trench [103,189]. Adakites are common among the products of Cenozoic volcanism throughout the Phlippines Archipelago and within most volcanic complexes and are closely associated with calc-alkaline lavas and

volcanic rocks transitional between adakites and calc-alkalline series (termed earlier "transitional adakites" in [83]). Adakites from the Philippines (Batan, Mankayan, Baguio, Western and Eastern Volcanic Chains in Central Luzon, Negros, Acmigiun and Mindanao) display a wide range of $SiO_2$ contents (53–70 wt.%) coupled with high Sr/Y ratios (>30), low Y concentrations (<15 ppm) and extremely variable La/Yb ratios (10–150), somewhat similar to adakites from the Cretaceous Stanovoy Suture Zone (Table 7). Adakitic rocks from Luzon and Mindanao also exhibit quite variable MgO (0.6–4.0 wt.%), Cr (2–219 ppm) and Ni (7–49 ppm) contents reflecting not just the effects of protracted low-pressure magmatic differentiation but also various degrees of contamination of primary adakite magmas by ultramafic-mafic rocks of the pre-Cenozoic ophiolitic basement [103]. High-Nb basalts are spatially associated with adakites in the Sulu arc of southwestern Mindanao [59]. The melting of young oceanic crust has been considered as main mechanism of formation for the Philippines adakites [103], although other possible origins have been also proposed by different authors [74,75]. High-Nb basalts are believed to be products of partial melting of adakite-contaminated peridotite source [59], or mafic melts derived from mixing of HFSE-enriched asthenospheric and depleted lithospheric sources within the mantle wedge beneath the Philippines island arc system [130,155].

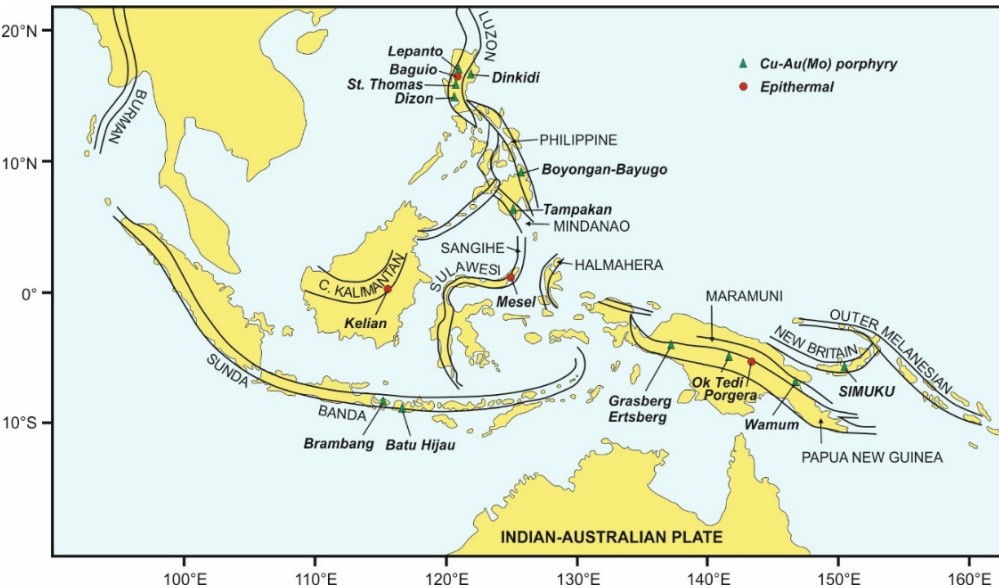

**Figure 12.** General tectonic setting and distribution of Cu–Au(Mo) porphyry and Au–Ag epithermal deposits associated with adakites and high-Nb basalts (modified after [189]).

As a fully-developed, tectonically complex island-arc system with a protracted magmatic history and igneous rocks ranging in composition from arc tholeiite to calc-alkaline and adakite (with high-Nb basalt) series, the Philippines host numerous copper–gold porphyry and epithermal gold deposits and mineral showings of Mesozoic to Late Cenozoic age [103,190–198]. Porphyry-type copper–gold and epithermal (both high- and low-sulfidation) mineralization in the Philippines can be grouped into several metallogenic provinces (Figure 12), including the West Luzon arc (Lepanto Far Southeast, Acupan-Itogon, Antamok, Santo Tomas II, Didipio, Minlawi, Batong Buhay, Nyak-Suyoc, Victoria, Marian, Runrun, Dinkidi, Santo Niño and Dizon deposits, East Rizal and Baguio mining districts), the Paracale mining district in southeastern Luzon (Santa Elena, Tabas, Longos, Paracale deposits), Southwestern Negros (Bulawan and Sipalay Cu–Au deposits), Cebu Island (Atlas mine consisting of Biga, Carmen and Lutopan deposits) and Mindanao (Figure 12—giant Tampakan deposit in the south, Canatuan Cu–Zn–Au–Ag system on Zamboanga Peninsula in the west, Boyongan porphyry deposit in the northeast and Kingking cluster including Batoto, Bukal, Maragusan, Masara, Amacan, Sumlog and Kingking ore bodies in the southeast). In addition to numerous showings and exploration prospects throughout the

Philippines, the archipelago includes several giant mineralized systems and world-class mines (Figure 12; Table 8). The Baguio mineral district (Figure 12) in northern Luzon is a world-class mineralized area with reserves of 12 Moz Au and 1.2 million tons of copper and resources of 40 Moz Au and 5 million tons of copper [197]. The Baguio district includes large porphyry copper systems such as Santo Tomas II (over 20 Moz of Au equivalent; Table 8) as well as Antamok (8.3 Moz Au) and Acupan (17 Moz Au) epithermal gold deposits. Both porphyry and epithermal mineralization at Baguio is hosted in Plio-Pleistocene magmatic rocks that show distinct adakitic geochemical characteristics (Figure 13). The formation of porphyry copper mineralization is related to the subduction of the Scarborough Ridge and South China Sea oceanic lithosphere along the Manila Trench [193,194]. The Lepanto mineral cluster (Figure 12; Table 8) is associated with Pleistocene dacitic volcanics, which host porphyry copper mineralization (the FSE and Guinaoang deposits, Bulalacao and Buaki-Palidan prospects), enargite-luzonite replacement veins (Lepanto copper–gold deposit) and quartz-gold-base metal sulfide veins (Suyoc and Victoria gold deposits, Nayak-Palidan prospects). The namesake Lepanto Cu–Au deposit (Figure 12) with a strike length of 2.5 km is composed of the main hydrothermal breccia and the footwall and hanging wall branching veins, which are oblique to the main ore body and most probably represent tensional fracture filling [197]. Alteration is predominantly silicic in the core of the porphyry system, which grades into intermediate to advanced argillic and outermost propylitic zone [197]. Ore minerals (Table 8) include enargite, luzonite ($Cu_3AsS_4$), tennantite–tetrahedrite, chalcopyrite, covellite, electrum and Au–Ag tellurides, while the principal gangue minerals are quartz, kaolinite, dickite, alunite, gypsum, anhydrite and barite.

**Table 8.** Main characteristics of selected Cu–Au deposits associated with adakites and high-Nb basalts in the Philippines, Indonesia and Papua New Guinea.

| Deposit/Host Rock | Age | Mineralization | Resources | Ref. |
|---|---|---|---|---|
| *Philippines* | | | | |
| Santo Tomas II adakite | 1.5 Ma | Au, Ag, El, Ccp, Cv, Cct, Bn, Dg, Sp, Gn, Hes, Pyr, Ptz, Alt, Mrn, Mlc, Ccl, Mol, Mag | 449 Mt @ 0.38% Cu, 0.0005% Mo, 0.7 g/t Au, 1.5 g/t Ag (1,700,000 t contained Cu) | [194] |
| Lepanto cluster adakite | 1.1–1.6 Ma | Au, Ag, El, Bn, Ccp, Hes, Ptz, Pyr, Slv, Eng, Tnt, Pol, Cal, Py, Qz, Anh, Mag, Py | *Lepanto:* 40.7 Mt @ 1.76–2.9% Cu, 2.4–3.4 g/t Au, 14 g/t Ag *Far South-East (FSE) Zone:* 657Mt @ 0.65% Cu and 0.94 g/t Au *Victoria and Teresa:* 11 Mt @ 7.3 g/t Au | [197] |
| Dizon adakite | 2.5–2.7 Ma | Au, El, Cu, Ccp, Cct, Ttr, Cin, Eng, Mlc, Gn, Sp, Antm, Mrc, Py, Mol, Mag, Anh | 187 Mt @ 0.36% Cu, 0.93 g/t Au, 2 g/t Ag, 0.003% Mo (670,000 t Cu) | [198] |
| Tampakan adakite | 4.24–4.26 Ma | Bn, Ccp, Cct, Cv, Dg, Eng, Tnt-Ttr, Py, Gn, Sp, Py, Mol, Mag, Anh, Col | 2940 Mt @ 0.52% Cu, 0.17 g/t Au, 0.006% Mo (15,000,000 t Cu) | [191] |
| *Indonesia* | | | | |
| Batu Hijau adakite | 3.67–3.76 Ma | Disseminated, veining, stockwork (Qz, Py, Ccp, Hem, Bn, Mol, Dg, Cct, Gn, Mag) | 1640 Mt @ 0.44% Cu, 0.35 g/t Au, 0.55 g/t Ag (7,200,000 t Cu) | [208] |
| Kelian adakite | 19.3–19.7 Ma | Propylitic, argillic, phyllic metasomatites, mineralized hydrothermal breccia (Qz, Cb, Adl, Py, Au, El, Sp, Gn, Ccp, Apy, Pyr, Pol) | 91.3 Mt @ 2.64 g/t Au and 4.85 g/t Ag | [207] |

**Table 8.** *Cont.*

| Deposit/Host Rock | Age | Mineralization | Resources | Ref. |
|---|---|---|---|---|
| Grasberg-Ertsberg adakite | 2.6–4.4 Ma | Stockwork veins, veinlets, disseminations (Anh, Ccp, Bn, Cct, Col, Dg, Cv, S, Eng, Mag, Hem, Mrc, Mol, Py, Sp, Gn, Au) | 11,100 Mt @ 0.6–0.7% Cu, 0.44–0.64 g/t Au, 2 g/t Ag (contained 32.6 Moz Au and 24,000,000 t Cu) | [215] |
| | | ***Papua New Guinea*** | | |
| Porgera HNB | 5.1–6.1 Ma | Mag–Cb-sulfide (Stage 1) and Qz–Py–Au (Stage 2) veins (Py, Ccp, Au, Mrc, Pyr, Cal, Ptz, Sp, Gn, Apy, Slv) | *Waruwari orebody*: 54 Mt @ 4.3 g/t Au; *Zone 7*: 5.9 Mt @ 27 g/t Au | [214] |

Mineral abbreviations: Qz—quartz, Cb—carbonates, Adl—adularia, Zeo—zeolite, Hem—hematite, Py—pyrite, Mrc—marcasite, Mag—magnetite, Anh—anhydrite, Apy—arsenopyrite, Ccp—chalcopyrite, Cv—covellite, Cct—chalcocite, Bn—bornite, Dg—digenite, Eng—enargite ($Cu_3AsS_4$), Ccl—chrysocolla, Mlc—malachite, Col—colusite ($Cu_{26}V_2(As, Sn, Sb)_6S_{32}$), Tnt–Ttr—tennantite–tetrahedrite, Sp—sphalerite, Gn—galena, Antm—antimonite, Mol—molybdenite, Cin—cinnabar, Argn—argentite, Pol—polybasite (Ag–Cu–Sb–As sulfosalt), Hes—hessite ($Ag_2Te$), Ptz—petzite ($Ag_3AuTe_2$), Slv—sylvanite (Au, Ag)$Te_2$, Pyr—pyrargyrite ($Ag_3SbS_3$), Alt—altaite (PbTe), Cal—calaverite ($AuTe_2$), Mrn—merenskyite (Pt, Pd) (Te, Bi)$_2$, El—electrum, Au—native gold, Ag—native silver, Cu—native copper, S—native sulfur. Moz—million ounces.

**Figure 13.** Variations of Sr/Y versus Y (ppm) in igneous rocks associated with porphyry Cu–Au–Mo (triangles) and epithermal Au–Ag (circles) deposits in the Philippines, Indonesia and Papua New Guinea (PNG). Data sources: Kelian [207]; NE Mindanao [103,192]; Brambang [209]; Cebu Island (Atlas, Lutopan) [195]; Baguio mineral district [103,193,194,197]; Dinkidi [190]; Lepanto cluster [103,194,197]; Dizon [198]; Tampakan [191,196]; Wamum [212].

The giant Atlas porphyry system (1420 Mt at 0.45% Cu, 0.018% Mo, 0.24 g/t Au and 1.8 g/t Ag) is associated with the 108–110 Ma old Lutopan quartz diorite stock emplaced into the Cretaceous volcanic-sedimentary Cansi Formation in the central Cebu Island. The hypogene mineralization at the Atlas deposit consists of the early-stage quartz–

magnetite–chalcopyrite–pyrite veins associated with potassic and propylitic alteration and late-stage anhydrite–pyrite–chalcopyrite–hematite veins associated with phyllic metasomatic assemblages [195]. Supergene mineralization is represented by abundant gypsum-dominated veins. The Lutopan stock displays high Sr/Y ratios (54–69) typical of adakites (Figure 13). The Lutopan adakite magma was apparently water-rich (amphibole and biotite phenocrysts), oxidized (average ΔFMQ of +2.7) and, based on enriched initial Sr and Nd isotope ratios, derived from the delaminated juvenile lower crust [195]. An alternative model invokes the partial melting of the amphibolite-facies, paleo-Pacific oceanic crust in the Early Cretaceous subduction zone beneath the Philippine archipelago [103]. The island of Mindanao (Figure 12) is richly endowed with Cu–Au porphyry mineralization, which, in many cases, is associated with Cenozoic adakitic magmatism [83,189]. The Surigao peninsula in NE Mindanao is composed of Cretaceous basement ophiolite and metasediment (Concepcion schist) overlain with Eocene to Pliocene volcanics, carbonate and turbidite deposits as well as Pleistocene andesite formations [192]. The copper–gold mineralization at the Boyongan and Bayugo deposits is primarily hosted in quartz-vein stockworks associated with Pleistocene (2.1–2.3 Ma) diorite porphyry intrusions [192]. The NE Mindanao deposits contained about 244 million tons of chalcopyrite–bornite–chalcocite–pyrite–magnetite–hematite ore grading 0.82% Cu and 0.83 g/t Au (2,000,000 tons of contained copper). Associated diorite porphyry intrusions have low Y and Yb contents combined with high Sr/Y ratios (>45; Figure 13) and are interpreted as typical adakites possibly formed via the partial melting of subducted oceanic crust beneath the NE Mindanao [103]. An alternative model based on Re–Os isotope geochemistry of some Mindanao adakites suggests a mantle origin for both ore metals and their host adakite magmas [96]. South Mindanao records a protracted history of terrane accretion and arc–arc collision [189] and is home to the super-giant Tampakan Cu–Au deposit (Figure 12), which contains around 15 Mt of copper in both reserve and resource categories (Table 8). The Tampakan deposit is essentially composed of the large-scale, high-sulfidation epithermal mineralization superimposed on a pre-existing low- to moderate-grade porphyry copper system [191]. The porphyry mineralization is hosted in Lower Pliocene pyroxene–hornblende–phyric andesite overlying Miocene volcanics and sediments and is intruded by hornblende diorite porphyry stocks and dikes [191]. High-grade Cu–Au mineralization is restricted to the advanced argillic metasomatic zone (also volumetrically most important), while silicic, phyllic and potassic alteration zones are either associated with lower copper and gold grades or are volumetrically minor and insignificant. Porphyry style mineralization (Table 8) is principally composed of disseminated pyrite and chalcopyrite with minor bornite. High-sulfidation epithermal mineral assemblages (Table 8) include pyrite, sulfosalts (enargite and luzonite), digenite, bornite with lesser covellite, chalcocite, chalcopyrite, tennantite–tetrahedrite, molybdenite and traces of colusite (Cu–Sn–V–As–Fe sulfide), galena and sphalerite [191]. Both andesite and porphyry intrusions at Tampakan predominantly display adakite-like Sr/Y ratios (Figure 13) interpreted as the result of "prolific crystallization of hornblende earlier in the crystallization sequence" from hydrous mantle-derived mafic melts [191]. In summary, independently of the petrologic mode of adakite generation (e.g., slab melting, lower crustal melting or fractionation of mantle melts), Cu–Au mineralization in the Philippines appears to be closely associated, both spatially and temporally, with Cretaceous (Atlas deposit on Cebu Island) and Late Cenozoic (Luzon and Mindanao) adakite magmatism.

### 4.2.2. Indonesia and Papua New Guinea

Indonesia and Papua New Guinea are two other island arc systems in the western Pacific where sporadic adakitic and HNB magmatism is associated with epithermal gold and copper–gold porphyry deposits (Figure 12). The Indonesian archipelago, which stretches over 5000 km from west to east, consists of more than 18,000 individual islands and is composed of several subduction systems [189], including (in the general west to east direction) the most prolific Sunda, Banda, Sulawesi, Sangihe and Halmahera volcanic arcs (Figure 12). The Sunda arc, which includes the islands of Sumatra, Java, Sumbawa,

Lombok and Flores, records the protracted (at least since Cretaceous) subduction of the Indo-Australian plate beneath Eurasia at a rate of ~6–7 cm yr$^{-1}$ along the Java Trench (Figure 12). This subduction is marked by prolific Late Cenozoic volcanism, which includes such famous volcanoes as the Toba caldera in West Sumatra (which caused severe human population bottleneck at ~74,000 B.P.), Krakatoa (famous 1883 explosive eruption offshore East Sumatra), Tambora (Sumbawa; 1815 eruption caused the "year without summer"), Merapi and many others. Although the age of oceanic crust at the Java Trench is 40 to 100 Ma, subduction is oblique and capable of generating additional stress-related heat flow and elevating geotherms in the downgoing Indo-Australian slab [199,200]. Sulawesi Island is located at the triple junction between the Eurasian, Australian and Phlippine plates associated with the subduction of the Australian and Philippine plates under the Eurasian plate and sinistral movement along the Australian–Philippine transform plate boundary (Figure 12; [189]). Sulawesi is bounded by several small trenches associated with subduction of relatively young (Late Miocene North Banda basin, Middle Eocene to Early Miocene Makassar Straight basin, Middle Eocene Celebes Sea) oceanic crust (Figure 12). Recent volcanic manifestations are restricted to the NE Sulawesi region, which is linked to the NW-dipping subduction of the Eocene-Oligocene oceanic lithosphere of the Molucca Sea representing southern termination of the Sangihe Arc [201]. The Molucca Sea plate represents the very rare case of divergent double subduction and is bound by two volcanic arcs—the Sangihe arc in the west and the Halmahera arc in the east (Figure 12). The Sangihe and Halmahera arcs are facing each other and currently colliding [202]. The Sangihe arc consists of 25 volcanic centers (8 currently active volcanoes are located within the southern segment of the arc) composed of pyroxene-dominated arc tholeiite and calc-alkaline series in the south and hornblende-dominated calc-alkaline volcanics in the north [202]. The last subduction beneath the Halmahera arc (along the Halmahera Thrust) started around 5 Ma and the onset of volcanism is dated at ~3 Ma, when the downgoing Molucca Sea slab reached the depths of ~100 km [202]. Active volcanic centers along the Halmahera volcanic front erupt typical differentiated (basalt through dacite) calc-alkaline magmas with strong sediment recycling signature, as suggested by their Pb-isotopic ratios [202].

Adakites in Indonesia are found in Sumatra [200], Java [199,203], Sulawesi [201,204,205] and Borneo [206]. Adakites in the Sunda arc (occurrences on Sumatra, Java and Sunbawa) are best interpreted as a consequence of the oblique subduction of Indo-Australian oceanic plate at the Java Trench (Figure 12). Some adakites in Western Sulawesi (such as ~5 Ma Palopo granodioritic intrusion) have formed through partial melting of delaminated lower crust that caused widespread Late Miocene to Pliocene rifting throughout the Western Sulawesi micro-continent [205]. This geodynamic setting is reminiscent of the post-collisional extension within the Stanovoy Suture Zone, which brought to the surface metal-rich adakite magmas sourced from the thickened lithosphere enriched in precious metals by asthenospheric mantle melts [181]; this study. Finally, adakites in Borneo (12–14 Ma Bau and 19–21 Ma Sintang felsic igneous suites of West Sarawak) are products of the remelting of hydrous arc basalt emplaced into the Borneo crust "tens or hundreds of millions of years previously" [206]. Although most Mesozoic to Cenozoic copper-gold deposits and showings within the Indonesian arcs (Sunda, Banda, Sangihe, Halmahera) are associated with typical normal and high-K calc-alkaline and shoshonitic magmas [189], some large, world-class porphyry and epithermal mineralized systems in Indonesia are clearly hosted in adakites and high-Nb basalts (Figure 12 and Table 8).

The Kelian epithermal gold deposit in East Kalimantan (Figure 12) is the largest gold producer in the country of Indonesia churning out between 400 and 500 Koz annually between 1992 and 2004 (Table 8). A total of 179 t (6 Moz) Au and 145 t Ag has been mined by Rio Tinto subsidiary in Indonesia during this period [207]. The deposit is a low-sulfidation epithermal system hosted in predominantly adakitic (hornblende-rich) andesite and dacite extrusives (Figure 13) with subordinate calc-alkaline basalts, basaltic andesites, dacites and rhyolites. U–Pb data on zircons from felsic intrusives and modern river sediments constrain the evolution of Kelian magmatic-hydrothermal system between

21.2 and 19.7 Ma, suggesting that the emplacement of adakite magmas and development of epithermal gold mineralization in the Kelian district took place within the period of 0.5 to 1 Ma [207]. Gold mineralization is represented by coarse native gold and electrum, gold intergrowths with sulfides, quartz, carbonates, Au–Ag sulfosalts and by solid solution Au impurities in pyrite [207]. Alteration styles range from proximal quartz–clay–pyrite to distal chlorite–carbonate–clay. Both alteration patterns and precious metal grades at the Kelian gold deposit resemble those typical of adakite-hosted Au deposits and showings in the Kamchatka volcanic province (compare Tables 5 and 8). The Batu Hijau deposit, located on the island of Sumbawa in the Sunda arc, represents the second largest (Table 8) open pit copper–gold operation in Indonesia after the giant Grasberg mine in West Papua (Figure 12). The Cu–Au sulfide (mostly bornite and chalcopyrite with minor native Au; Table 8) mineralization is hosted in the adakitic hornblende tonalite–diorite intrusive complex (Figure 13) and is associated with potassic alteration (biotite + quartz + magnetite) core grading outwards into pervasive propylitic (chlorite+epidote) alteration and finally into broad, fracture-controlled phyllic (serecite + pyrite), sodic (albite) and advanced argillic (sericite + alunite + kaolinite + pyrophyllite) alteration halos [208]. The highest copper grades are associated with potassic alteration core as is typical of many Circum-Pacific telescoped porphyry systems [4–6]. The Batu Hijau mine typically produces in excess of 300 million pounds of copper, 150 Koz of gold and almost a million ounces of silver on an annual basis. Brambang, a smaller Cu–Au porphyry deposit with characteristics similar to Batu Hijau, was recently discovered on the island of Lombok (Figure 12). Brambang mineralization (bornite, chalcopyrite, digenite, pyrite) is hosted in adakitic (Figure 13) diorite and tonalite, which appear to be "ore mineralization causative intrusions also recognized in Batu Hijau" [209]. Porphyry mineralization at Brambang is overprinted with high-sulfidation epithermal gold and silver and is enveloped in potassic, phyllic, propylitic, advanced argillic and argillic alterations. Adakitic intrusions exsolved metal-rich, saline (halite, sylvite, hematite, anhydrite) fluid, which precipitated Cu and noble metals at relatively high temperature of 450–600 °C and salinity of 60–70 wt.% NaCl eq. (corresponding pressure of ~300 bar and depth of ~3 km from the paleosurface) [209]. Some other Cu–Au porphyry and epithermal Au–Ag showings in Indonesia are also associated with adakite magmatism in one fashion or another. However, the majority of intrusive and volcanic rocks that host Cu–Au mineralization throughout Sunda, Banda, Sangihe and Halmahera arcs are dominated by calc-alkaline and shoshonitic intrusive and volcanic rocks and do not form part of the current review dedicated exclusively to adakites and high-Nb basalts.

Adakites occur on the island of New Guinea [210,211], which is also home to some world-class Cu–Au porphyry mineralization [189]. The island is principally composed of accreted terranes accumulated along the northern Australian continental margin during the Cenozoic [212]. Most autochtonous terranes (including supra-subduction zone ophiolites) are concentrated within the New Guinea Orogen (Central Range), which comprises the Papuan Fold and Thrust Belt along with uplifted tectonic blocks of the New Guinea Mobile Belt (Figure 12). Most of the Late Cenozoic magmatism in Papua New Guinea, including adakites, is localized within the Maramuni arc (Figure 12), which intrudes the New Guinea Orogen. There are two competing plate-tectonic models for the origin of the Maramuni arc magmatism [212]. The first model suggests that the collision of the Ontong Java Plateau with the Melanesian arc in the Early Miocene triggered south-dipping subduction at the Trobriand trough (Figure 12). Subsequent episodes of arc–continent collision starting at 12 Ma eventually led to the formation of the New Guinea Orogen [212]. The second hypothesis invokes north-dipping subduction system located to the south of New Guinea. This subduction resulted in collision of the Australian continent and an outer arc terrane, magmatism of the Maramuni arc and growth of the New Guinea Orogen from 12 Ma [212]. Young volcanism in the neighboring island of New Britain is associated with subduction of the 39–28 Ma-old oceanic crust of the Solomon Sea plate at the New Britain trench [212]. Adakites in the Arid Hills area of the Maramuni arc were derived from a garnet-bearing

source believed to be a subducted oceanic crust [37,210]. The nearby Mt. Lamington volcano in 1951 erupted andesitic lavas with adakite-like geochemical characteristics (Sr > 900 ppm; Y = 15–16 ppm; Yb < 1.5 ppm) [213]. Haschke and Ben-Avraham suggested that adakites (Quaternary hornblende-bearing dacites and trachytes) from the Lusancay Islands and Arid Hills (located 600 km apart) were either formed by melting of hot and young ("newborn") oceanic crust along the Trobriand Trough or derived from a deep mafic source (remnants of earlier subducted slab or lower crust) in the attenuated, collision-modified lithosphere [212]. The second scenario is similar to that invoked by J. Richards for the origin of alkaline mafic intrusions (as we will show further, essentially high-Nb basaltic rocks), which host the Porgera and Mout Kare gold deposits in the New Guinean Highlands [214].

The western (Indonesian) part of the island of New Guinea is home to the largest gold and the second-largest copper reserve in the world—the Grasberg deposit (Figure 12; Table 8). The giant Grasberg mine (located at the elevations of ~4200 m in the remote highlands of western New Guinea), which is currently owned by the Government of Indonesia and the Freeport-McMoran Corporation of the United States and mined by Freeport-McMoran, has around USD 40 billion in gold ore reserves and produced 528 billion ounces of copper and 53 million ounces of gold during the period from 1990 to 2019. Grasberg mine is a complex mining operation consisting of the main open pit, the Deep Ore Zone underground mine and the Big Gossan underground mine. Annual production from the entire Grabserg mining operation topped 1 million pounds of copper, 1 million ounces of gold and 3 million ounces of silver during recent years [215]. Along with the nearby Ertsberg mine (3.6 billion tonnes grading 0.60% Cu and 0.44 g/t Au), Grasberg forms one of the largest copper ore systems in the world, both by its lateral and vertical extent and overall ore tonnage. The economic hydrothermal magmatic and skarn mineralization at Grasberg and Ertsberg sites is vertically continuous over at least 1500 m and is hosted by two shallow potassic diorite intrusives with adakitic affinities (Sr/Y > 30; Y < 15 ppm) intruding Tertiary carbonates and Jurassic-Cretaceous siliciclastic rocks [215]. Porphyry-style alteration is well developed at both deposits following the typical pattern of a potassic core surrounded by external zones of phyllic and propylitic alteration. The potassic alteration core includes Cu-stockwork with chalcopyrite dominating the ore mineralogy and bornite content increasing with depth [215]. Covellite is the most abundant Cu mineral throughout the Grasberg phyllic zone followed by subordinate chalcopyrite and pyrite with native gold inclusions. Native gold is also included in chalcopyrite and bornite within the potassic alteration zone.

A minor adakite component is present at the Ok Tedi Cu–Au deposit in the Eastern New Guinea Orogen (Figure 12), which is the youngest (1.1—1.2 Ma) giant (854 Mt at 0.64 wt.% Cu and 0.78 g/t Au; 2.86 Mt Cu and 706 t Au produced until 2003) copper porphyry system in the world. Host rocks for chalcopyrite–chalcocite–bornite-bearing quartz veins and stockworks as well as massive magnetite and sulfide skarns are predominantly high-K calc-alkaline monzogabbros, monzodiorites and monzonites with subordinate adakitic tonalite [189]. Another adakite-hosted Cu–Au porphyry mineralization—the Wamum prospect—is known in the eastern New Guinea Orogen and is associated with the Miocene magmatism of the Maramuni arc (Figures 12 and 13). Potassic alteration at Wamum is centered on central tonalite stock intruding andesite volcanics with propylitic alteration. Pyrite and chalcopyrite mineralization at Wamum occurs as a fracture-controlled stockwork in Miocene (11.88–12.08 Ma) adakitic tonalite porphyry [212]. Another example of an adakite-hosted porphyry system is the Simuku Cu–Mo–Au deposit on the island of New Britain (Figure 12). At Simuku, Cu–Mo mineralization (mostly chalcopyrite with minor bornite and molybdenite) is hosted in adakite (Sr/Y > 30; Y < 14 ppm) porphyry intruding mafic to intermediate volcanic/volcaniclastic package of the Early Miocene age [216]. The Simuku Cu–Mo–Au resource stands at 373.6 million tonnes grading 0.31% Cu, 58.5 g/t Mo and 0.05 g/t Au [216]. The adakite component is also recognizable in intrusive suites hosting some other Cu–Au porphyry prospects (e.g., Mount Nakru, Uasilau Yau-Yau, Esis, Tripela as well as others) through out the New Britain magmatic arc.

Different types of gold mineralization in Papua New Guinea are associated with mafic alkaline intrusive magmatism. The Porgera epithermal gold deposit (Figure 12) is hosted in a high-level mafic intrusive complex and the surrounding Late Cretaceous sediments and is strongly controlled by such structural features as intrusive contacts, breccia zones and faults [214]. The Porgera gold resource (390 tonnes Au and 990 tonnes Ag; Table 8) is contained within two principal structures: (1) a main low-grade zone at Waruwari (78 million tonnes grading 3.7 g/t Au and 11.3 g/t Ag) and (2) a satellite high-grade zone located along the Roamane fault (4.5 million tonnes grading 21.9 g/t Au and 23.1 g/t Ag). The low-grade zone is mostly composed of As- and Au-bearing pyrite with minor sphalerite, galena, arsenopyrite, chalcopyrite, pyrargyrite, tetrahedrite, freirbergite, native gold and electrum associated with phyllic (serecite–carbonate) alteration [214]. Ore minerals typically fill open spaces and fractures within or near quartz–carbonate veins [214]. High-grade mineral associations, in addition to micro- and macroscopic gold and electrum, include Au–Ag-tellurides (krennerite, hessite and petzite) frequently mingled with pyrite, roscoelite and tetrahedrite [214]. Mafic rocks associated with both low- and high-grade Au mineralization at Porgera are characterized by predominant porphyritic textures with clinopyroxene, hornblende and plagioclase phenocrysts locally grading into coarse-grained ophitic intrusive rocks with large hornblende oikocrysts showing remarkable petrographic resemblance with high-Nb basalts from Kamchatka and northern Honduras [52,131]. Porgera basalts and basaltic andesites are sodic ($Na_2O/K_2O$ = 1.8–3.9; $Na_2O$ = 3.6–5.54 wt.%; Table 1 in [214]) alkaline rocks, which plot, as most high-Nb basalts worldwide do [131], into the alkali basalt–hawaiite–mugearite fields on the TAS diagram [214]. They display characteristic high Nb and variable $TiO_2$ concentrations (47–88 ppm and 0.79–1.40 wt.%, respectively; [214]) combined with the sharp Nb enrichments in primitive mantle-normalized trace element patterns (Figure 14B). Their overall HNB setting can be further emphasized by the Nb/U-Nb diagram, where they plot into the HNB field together with reference high-Nb basalts from Kamchatka and Central America volcanic arcs (Figure 14A). Previous studies viewed Porgera rocks as fractionated, non-subduction melts derived from metasomatized lithospheric mantle sources, which exploit deep faults related to continent-arc collision processes [214]. Based on geochemical comparisons presented in Figure 14, we believe that the Porgera sodic, high-Nb intrusions are typical HNB magmas sourced in the adakite-metasomatized mantle related to the subduction-driven development of the Miocene Maramuni arc.

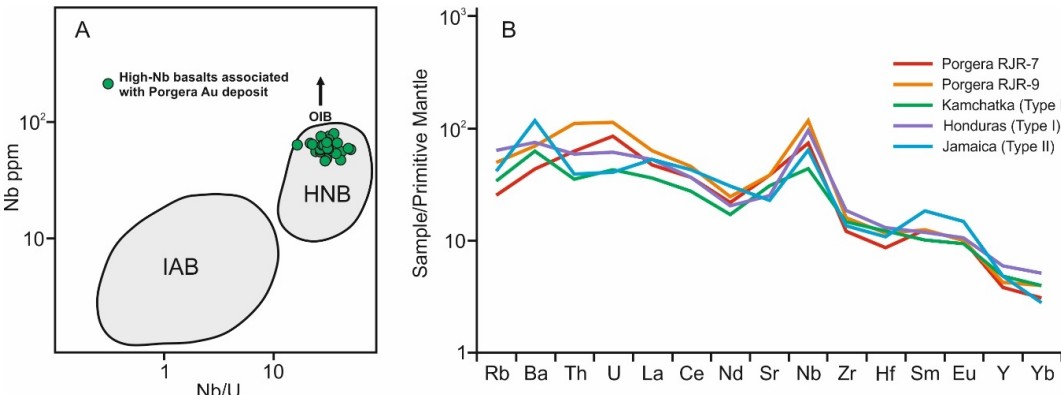

**Figure 14.** (**A**) Variations of Nb (ppm) versus Nb/U ratio in mafic alkaline rocks associated with Porgera Au deposit (PNG). Fields of island-arc basalts (IAB) and high-Nb basalts (HNB) are modified from [58,131]. (**B**) Primitive mantle-normalized trace element patterns of representative Porgera alkaline rocks (samples RJR-7 and RJR-9 from [214]) compared with Type I HNBs from Kamchatka [52] and Honduras [131] and Type II HNB from Jamaica [129]. Normalizing values are from [104]. Type I HNBs display N-MORB-like Sr–Nd–Pb isotopic signature, while Type II HNB was most probably derived from isotopically enriched, OIB-like mantle sources [153]. Please see text for further discussion of HNB origins.

### 4.2.3. Andean Magmatic Arc

The subduction of the Nazca plate beneath the South American plate resulted in magmatic and tectonic build-up and the formation of the Andean magmatic arc. The Nazca plate is a relatively young plate formed during the destruction of the Farallon plate approximately 23 million years ago [118]. As the Nazca plate carries several aseismic ridges (from north to south—Carnegie, Nazca, Iquique and Juan Fernandez) associated with thickened oceanic crust, the convergence of these ridges within the Andean margin (Figure 15) resulted in intermittent periods of flat slab subduction beneath Peru, central Chile and Argentina [118,120]. This episodic flat sudduction regime is well represented in the geologic record by tectonic uplift, arc migration (rollback), foreland deformation and the formation of pull-apart basins, as well as by prolific adakite-type magmatism throughout the Andean region [44,67,84,105,120]. Several authors suggested that flat slab subduction and adakite magmas are closely associated both spatially and genetically with the formation of Cu(Mo)–Au(Ag) deposits in the Peruvian and Chilean Andes, as well as the Patagonian metallogenic province [217–219]. Figure 15 illustrates close geographic link between Cu–Au mineralization and adakites throughout the Andean magmatic arc. It is important to emphasize for the purpose of this overview that Andean adakites (intrusive and volcanic facies) are associated, both temporally and structurally, with some of the largest Cu–Au porphyry systems in the world (Table 9). Based on the results from Stanovoy adakites, we believe that the link may also possibly be genetic, as metal-rich adakite magmas (similar to the Utanak hornblende adakites with Cu–Ag–Au alloys, Figure 11) can exsolve $H_2O$–S–Cl fluids, enormously enriched in ore metals, such as gold, copper, silver and others. Various tectonic, structural and physico-chemical factors in the Andean upper crust will facilitate the deposition of ore minerals and the formation of the unique metal inventory of such magmatic hydrothermal systems as Escondida, Chuquicamata and El Teniente. El Teniente, the world's largest copper deposit (Table 9) is hosted in the Late Miocene to Pliocene intrusive rocks that exhibit clear the adakite's geochemical signature (Figure 16). Mineralized breccia pipes with Cu contents in excess of 1 wt.% have a vertical extent of more than 1.5 km, and their roots are still not properly investigated. These high-grade pipes are associated with pervasive potassic alteration and are enveloped in a dense network of biotite-rich veins with chalcopyrite, bornite and pyrite [218]. Later-stage veins are composed of quartz, anhydrite, serecite, tourmaline, feldspars and various Cu–Mo minerals. Petrological data suggest that Cu-rich fluids were exsolved from a large (batholith-sized), long-lived, open-system magma chamber, which cooled and crystallized at depths over 4 km below the paleo-surface. According to some estimates, "to produce the ~$100 \times 10^6$ tonnes of Cu in the deposit requires a batholith-size (>600 km$^3$) amount of magma with ~100 ppm Cu" [218]. Metals, apparently, were transported from the differentiating magma body via hot (>550 °C), oxidized ($f$O$_2 \geq$ NNO + 1.3), SO$_2$-rich fluids exsolved from the roof of this giant magmatic plumbing system to form high-grade mineralized breccia pipes and associated veins and linear stockwork zone [236]. Some studies suggest that the El Teniente magma chamber was additionally fluxed with "exceptionally Cu, Mo, Li, and, probably, also S-rich volatile phase" based on the correlation between the inception of Cu precipitation and 4- to 10-fold increases in the Cu, Mo, Li and Fe concentrations in the "main" fluid input [237]. This possible duplication of fluid fluxes from the mantle and the evolving magma chamber may potentially explain dramatic vertical extent of copper mineralization at El Teniente as indicated by drilling and geophysical data.

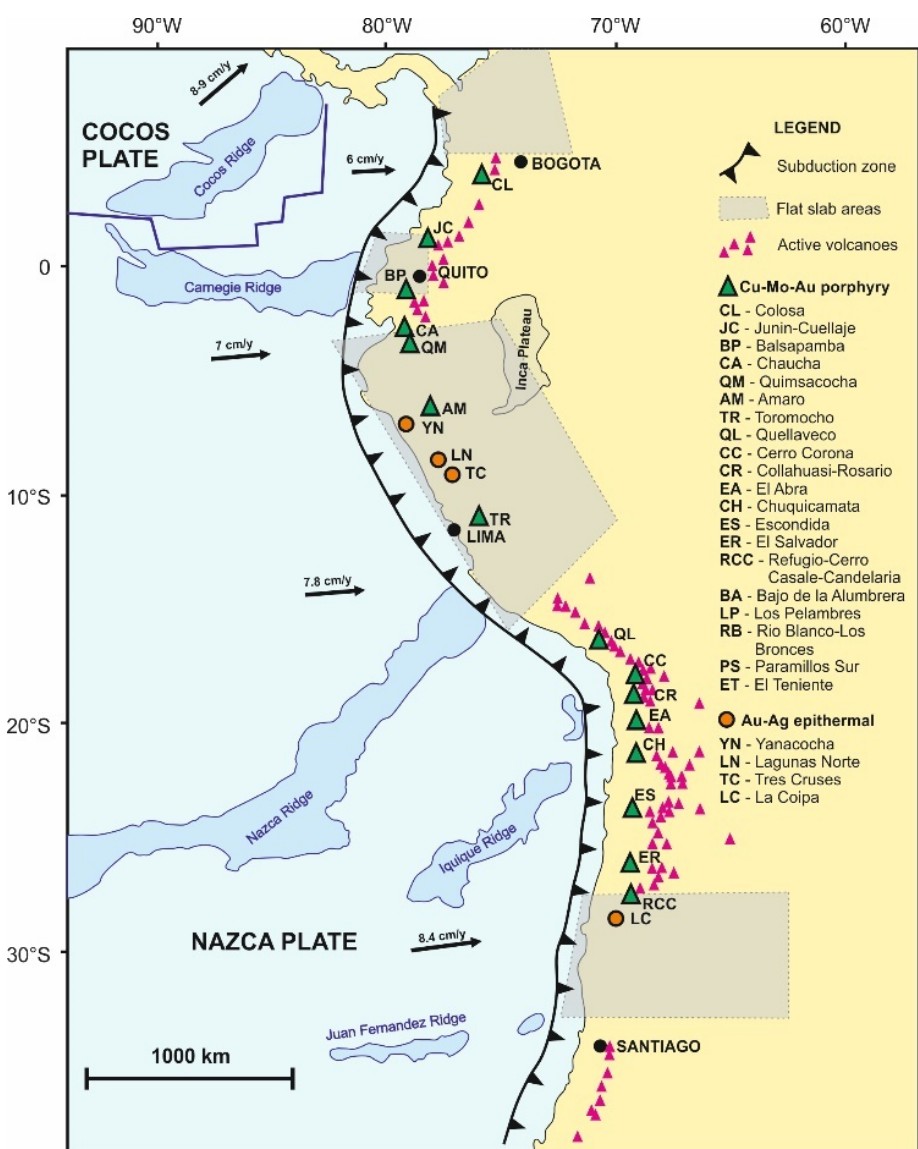

**Figure 15.** Distribution of Cu–Au porphyry and Au–Ag epithermal deposits associated with adakites and flat slab regions in the Andean magmatic arc. Tectonic framework of the Andean volcanic chain is modified from [217]. Location of adakite-hosted Cu–Au deposits is based on the geochemistry of magmatic rocks associated with porphyry and epithermal mineralization taken from the following sources [218–240] and shown separately in Figure 16.

**Table 9.** Main characteristics of selected adakite-hosted Cu–Au deposits in the Andes.

| Deposit/mine | Age | Geology | Resources | Refs. |
|---|---|---|---|---|
| El Teniente | 4.4–7.1 Ma | Breccia pipe | 20,731 Mt @ 0.62% Cu, 0.019% Mo, 0.005 g/t Au, 0.52 g/t Ag (8 ppb Pt, 32 ppb Pd in concentrate) | [218,236,237] |
| Escondida | 34–38 Ma | Breccia+veins; extensive supergene enrichment zone (65% of all resource) | Reserves: 7.688 Gt @ 0.61% Cu; Resources: 18.818 Gt @ 0.77% Cu | [224,238] |
| Los Pelambres | 9–10 Ma | Dacite porphyry breccia | 7458 Mt @ 0.617% Cu, 0.015% Mo, 0.028 g/t Au, 1.26 g/t Ag | [85] |

**Table 9.** *Cont.*

| Deposit/mine | Age | Geology | Resources | Refs. |
|---|---|---|---|---|
| Bajo de la Alumbrera | 6.64–8.02 Ma | Porphyry stocks and dikes | 806 Mt @ 0.53% Cu, 0.64 g/t Au, 2.5 g/t Ag (8 ppb Pt, 35 ppb Pd in concentrate) | [227] |
| Chuquicamata | 31.1–34.6 Ma | Multistage porphyry with major supergene enrichment zone | 11,450 Mt @ 0.76% Cu, 0.04% Mo, 0.013 g/t Au, 5 g/t Ag | [235,239] |
| El Salvador | 41.6–43.6 Ma | Multistage porphyry with dikes | 3836 Mt @ 0.447% Cu, 0.022% Mo, 0.1 g/t Au, 1.5 g/t Ag | [222] |
| Yanacocha | 8.5–11.5 Ma | Andesite flows, silicic diatreme, breccia and pyroclastic ignimbrite | 1142 Mt @ 0.9 g/t Au (contained 32.6 Moz Au) | [225] |
| Lagunas Norte | 16.8–17.3 Ma | Andesite lava, polymictic breccia, silicic diatreme | 123.5 Mt @ 1.83 g/t Au (contained 7.3 Moz) | [226,240] |

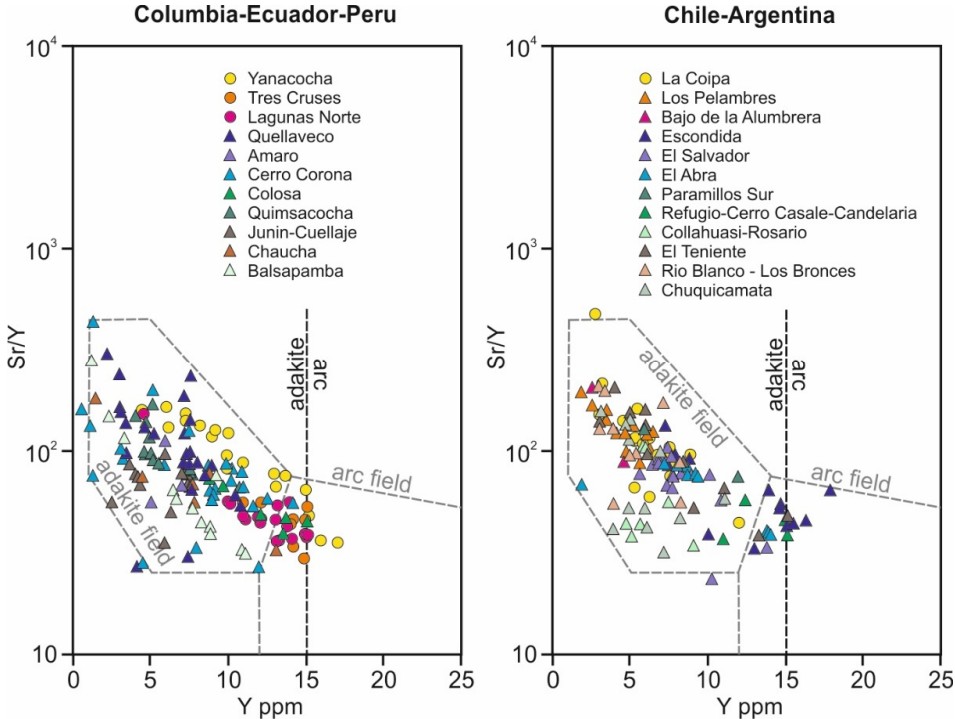

**Figure 16.** Variations of Sr/Y versus Y (ppm) in igneous rocks associated with porphyry Cu–Au–Mo (triangles) and epithermal Au–Ag (circles) deposits in the Andean magmatic arc. Data sources: Yanacocha [225]; Tres Cruses [219]; Lagunas Norte [226,240]; Quellaveco [228]; Amaro [229]; Cerro Corona [232]; Colosa [233]; Quimsacocha [234]; Junin-Cuellaje [234]; Chaucha [234]; Balsapamba [234]; La Coipa [230]; Los Pelambres [85]; Bajo de la Alumbrera [227]; Escondida [224,238]; El Salvador [222]; El Abra [220]; Paramillos Su [219]; Refugio-Candelaria [223]; Collahuasi-Rosario [231]; El Teniente [218,236,237]; Rio Blanco-Los Bronces [221]; Chuquicamata [235,239].

The giant Escondida porphyry system (Figure 15) consists of the Escondida (including Escondida Este), Escondida Norte-Zaldivar, Pampa Escondida and the smaller Baker and Pinta Verde deposits, which together combine for a resource of 25 Gt grading around 0.5–0.6 wt.% Cu (Table 9). The Escondida mining cluster produced 144 million metric tons of copper up to date [238]. The Escondida cluster is part of the Mid-Eocene to Early Oligocene copper porphyry belt bounded by the trench-parallel transpressional Domeyko Faul System [224,238]. As in the case of the El Teniente system, copper sulfide mineral-

ization is hosted in magmatic-hydrothermal breccias, which are closely associated with multiphase granodiorite–tonalite porphyry stocks with distinctive adakite geochemistry (Figure 16; [224]). The Escondida system is vertically zoned in respect to alteration and mineralization. The deeper (structurally lower) potassic core is overprinted with phyllic (mostly serecite) alteration and contains mostly chalcopyrite and bornite with minor hypogene digenite, chalcocite, covellite and molybdenite [238]. This deep core is overlain by a higher-level propylitic (pyrite+chlorite+serecite) and advanced argillic alteration zone associated with enargite-rich, massive sulfide veins [238]. All deposits in the Escondida cluster are covered by supergene caps of various depth (40 to 400 m) composed of hematite, kaolinite, alunite, montmorillonite, nontronite, vermiculite, chrysocolla, malachite, atacamite, cuprite, dioptase, antlerite, brochantite and copper-bearing clay [238]. The Sr–Nd–Pb–Hf–O isotopic data indicate that the parental Cu-bearing magma was derived from a fertile mantle source and assimilated a minor (~10%) amount of the underlying Paleozoic lower crust on its way to the upper crust [224]. Mass balance calculations suggest that the Escondida magma chamber had a mass of $10^{12}$ tonnes and volume of more than 400 km$^3$, which is consistent with the area covered by adakitic and calc-alkaline intrusions [224]. Early sulfide saturation sequestered most of the highly siderophile elements, such as Au and PGE, so the ore fluid leaving the batholitic magma chamber is only slightly enriched in Au and Pd. This may explain why the Escondida ore has only minor palladium and gold credits [224].

    Another giant copper porphyry cluster—Chuquicamata—is located in the Atacama Desert of northern Chile (Figure 15) and includes, in addition to the Chuquicamata deposit itself, the Radomiro Tomic and Mina Sur deposits, the Minstro Hales mine and the prospects of the Toki group [239]. The combined resources of the Chuquicamata district are estimated at 113.4 Mt of contained copper, with the namesake Chuquicamata Cu–Mo deposit contributing 63.7 Mt. The base metal mineralization in the Chuquicamata district is associated with the same regional Domeyko lineament, which controls the Escondida porphyry system (see above). The host hornblende–biotite granodiorite Chuquicamata complex is sub-divided into three main intrusive facies. Two of them, the East Banco and the West Porphyries, are characterized by variable, but distinct, adakitic geochemical signals (Figure 16; [235]). The main hypogene mineralization assemblage of veined and disseminated bornite, digenite, chalcopyrite and covellite is clearly associated with intense potassic alteration. The potassic episode was followed by phyllic (quartz-serecite with minor kaolinite and illite) alteration carrying most pyrite mineralization supplemented with varying amounts of digenite, covellite, enargite, chalcopyrite, tennantite and bornite [239]. A final hydrothermal episode at Chuquicamata produced a supergene oxidation zone dominated by alunite, chrysocolla, atacamite, brochantite and copper wad [239]. Adakitic tonalite intrusions in the Chuquicamata district appear not only to be the source of metals for Cu–Mo mineralization but also the heat and fluid driver for the entire trend of large porphyry systems on a regional (>1000 km) scale in the Atacama Desert of the High Andes [235].

    In addition to Cu–Au(Mo) mineralization, adakites are found among rocks hosting epithermal gold mineralization in the Ecuadorian and Peruvian Andes (Figure 16). The Yanacocha epithermal district in the Cajamarca region of northern Peru (Figure 15) is the most productive group of epithermal gold deposits in the world. Yanacocha also appears to belong to the "acid-sulfate" epithermal type with close links to deeper levels of porphyry mineralization. The Lepanto cluster in the Philippines (see above) is one the best examples of these complex, telescopic epithermal-porphyry environments [189,197]. Geochronological data suggest that the Yanacocha district-scale mineralization developed in five stages between 13.6 Ma and 8.2 Ma [225]. The district's geology is dominated by folded and thrust-faulted Cretaceous sandstone and limestone intruded by lower (19.5–13.3 Ma) and upper (12.3–11.2 Ma) "andesite" [225]. The term "andesite" at Yanacocha is quite loose, as it includes, for example, ignimbritic lapilli tuff confined to the western part of the Yanacocha district. The volcanic sequence is further injected with various hornblende diorite–andesite–dacite–rhyolite porphyry dikes, plugs and stocks with distinctive

adakite geochemical features (Figure 16). Alteration at Yanacocha exhibits clear zoning with massive silica core surrounded with silica-alunite (quartz + alunite ± pyrophyllite), silica-pyrophyllite (quartz + pyrophyrllite ± alunite ± diaspore ± zunyite), argillic (iilite + smectite ± montmorillonite), propylitic (chlorite ± calcite ± illite) and muscovite-sericite (muscovite + sericite ± dickite ± topaz ± anhydrite ± K-feldspar) outer zones [225]. Principal ore minerals are enargite, pyrite (with inclusions of chalcopyrite, bornite and pyrrhotite), covellite, chalcopyrite, galena, native gold and electrum. The vertical zoning of the Yanacocha system indicates that diaspora-rich roots may track the rise of magmatic ore fluids exsolved from metal-rich adakite magma, while the muscovite-sericite alteration associated with the highest gold grades (>1 ppm) probably represent the area of metal precipitation from evolved hydrothermal brines [225]. The adakite component can be also detected in the world-class Laguna Norte Au–Ag deposit in northern Peru (Figure 16; Table 9). The 13.1 Moz Au high-sulfidation epithermal deposit is hosted in Upper Jurassic to Lower Cretaceous weakly metamorphosed quartzites and in overlying Miocene dacitic to rhyolitic volcanics impregnated by two volcaniclastic diatremes [240]. The volcanic sequence exhibits typical alteration zoning with central vuggy quartz surrounded by quartz–alunite and quartz–alunite–kaolinite outer zones. Mineralization is represented by auriferous pyrite, digenite, chalcopyrite, enargite, stibnite and arsenopyrite [240]. The D–O–S isotopic systematics of alunite, pyrite and enargite suggest that gold precipitated from acidic, $H_2S$-rich fluids of largely magmatic origin within the temperature range of 190 to 280 °C [240]. The adakitic geochemical signal (Sr/Y > 30, Y < 16 ppm, Yb < 1.5 ppm) is present in felsic volcanic clusts in the ore-forming diatremes and in dacitic domes surrounding the Lagunas Norte deposit.

### 4.2.4. Southeastern Alaska

Southeastern Alaska (Figure 17) consists of a collage of oceanic, island-arc and back-arc terranes, fragments of sedimentary basins, metamorphic core complexes and magmatic belts amalgamated along the northern Pacific margin during the last 200 Ma [241]. The modern convergent margin of southern Alaska is characterized by a complex lateral change of geodynamic environments along the axis of the Aleutian-Alaska Trench (Figure 17). Northward movement of the Pacific plate changes from oblique subduction beneath the central-eastern Aleutians to a simple transform boundary beneath the western Aleutian-Komandorsky region [31,47,49,53,124]. The west to east transition from "normal" to flat slab subduction to strike-slip tectonics along the southern Alaskan margin is intimately connected with subduction of the Yakutat microplate (Figure 17). Subduction of the Yakutat microplate, an anomalously thick and buoyant oceanic plateau, beneath Alaskan margin is believed to be responsible for Oligocene-Holocene volcanism and deformation beneath eastern Alaska [122,241,242]. Late Oligocene to Early Miocene (ca. 30 Ma to ca. 19 Ma) volcanic rocks in the Sonya Creek Volcanic Field (SCVF), the oldest volcanic unit in the Late Cenozoic Wrangell arc (Figure 17), mostly display typical arc-related, calc-alkaline differentiation trends [122]. Some eruptive units in the SCVF (Rocker Creek and Ptarmigan Creek Volcanics; Figure 17) contain biotite- and amphibole-bearing andesites, dacites and rhyolites with adakitic geochemical features (Sr/Y > 35, Y < 15 ppm). These rocks have distinctly "adakitic" Sr/Y ratios but low La/Yb ratios "inconsistent with the formal usage of the term "adakite" as defined by [37]" [122]. Berkelhammer et al. [122] suggested that these Y-depleted lavas represent mixing between adakite-like (slab, or lower crustal) melt and mantle wedge-derived magma, which is consistent with geophysical data suggesting that the Wrangell arc is located above the Yakutat slab edge. The slab-edge melting scenario similar to the original concept introduced in [124] may successfully account for the occurrence of adakitic signature in both older (Oligocene-Miocene) and younger (Pliocene-Pleistocene) volcanic rocks within the Wrangell arc. Young (<5 Ma) volcanism related to recent flat slab subduction beneath the Wrangell arc includes several large composite shield volcanoes (Blackburn, Jarvis, Sanford, Tanada Peak, Capital Mountain), some stratovolcanoes (Drum and Churchill) along with numerous cinder cones in the

interior of the Wrangell arc [242]. Adakite magmas occur along the frontal part of the Wrangell arc, and their origins are attributed to the partial melting of the Yakutat oceanic slab [122,242]. Again, frequent co-existence of adakite and "normal" calc-alkaline rocks within the single volcanic center along the Wrangell arc suggests widespread mixing between the slab-edge and mantle wedge derived magmas through out the 30 Ma evolution of the southeastern Alaska continental margin.

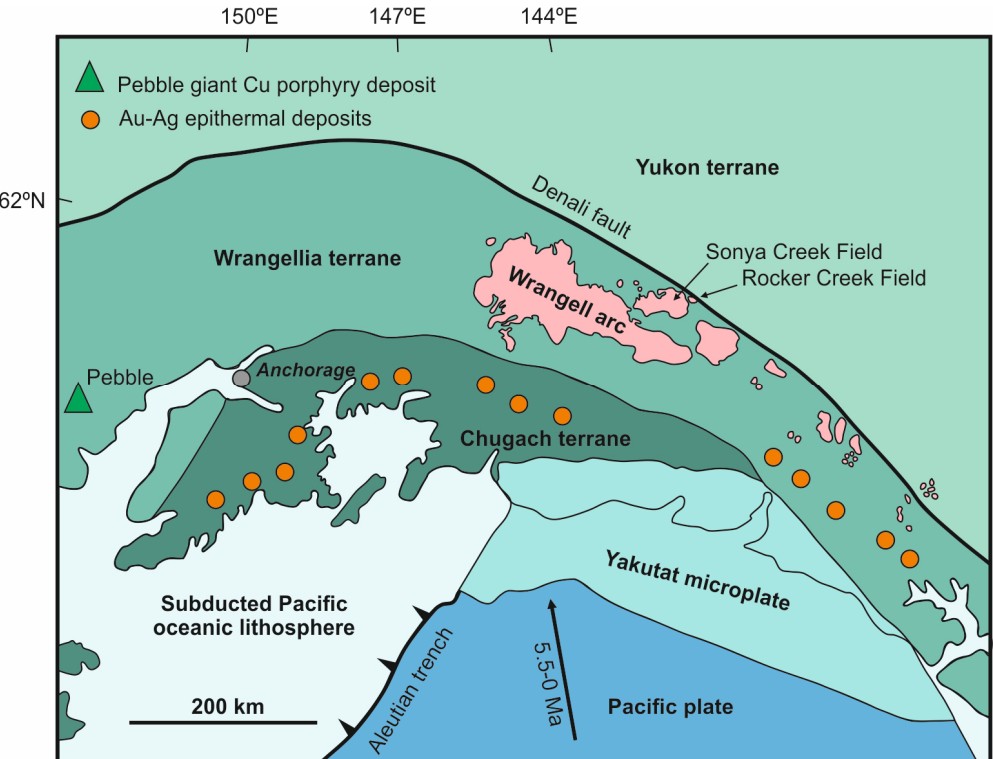

**Figure 17.** Simplified tectonic map of Southeastern Alaska showing location of principal adakite volcanic centers (23.3–18.6 Ma Sonya Creek and 26.3 Ma Rocker Creek Fields) and Cu porphyry (green triangle) and Au–Ag epithermal (orange circles) deposits. Location of main accreted terranes Yukon, Wrangellia and Chugach), subduction zones (Aleutian trench) and subducted plates (Pacific and Yakutat) are after [241].

Continental margin of southern Alaska, especially its Mesozoic to Early Tertiary accretionary prism, contains multiple lode-type gold deposits and showings (Figure 17; [243]). Gold-bearing quartz veins were emplaced in large-scale brittle structures after the regional deformation and metamorphism of clastic sediments in the Valdez and Orca Groups, the McHugh Complex and correlative units along the southern Alaska continental margin [243]. Regional deformation and metamorphism were accompanied by the emplacement of granitic magmas with distinctive adakitic signature (Sr = 367–692 ppm; Y = 3.53–4.27 ppm; Yb = 0.32–0.38 ppm; Sr/Y = 86–178; Table 3 in [244]). Although most of these occurrences are rather small (under 50,000 oz Au in individual veins and mineralized shear zones), the Chichagof district near the Border Ranges fault system produced over 800,000 ounces (25 tons) of gold [243]. As fore-arc areas are usually associated with low heat flow, a model invoking the spreading ridge subduction and opening of a slab window was put forward to explain localized high-*T*, low-*P* metamorphism and intense circulation of mesothermal gold-bearing fluids [243]. Slab melting of a spreading ridge crust (similar to Southwest Japan and NW Andes of Colombia), in our opinion, was the driving petrologic process behind this anomalous heat and fluid circulation within the south Alaskan accretionary margin. Gold was scavenged from various sources in the upper crust and re-deposited in fault-related, syn- and post-adakitic brittle structures in Mesozoic-Early Tertiary metasediments of the Chugach Terrane (Figure 17). This process (ridge subduction, slab window

and adakite production as defined in [126]) seems to be also operational in Mesoarchean paleo-subduction environments of the North Atlantic Craton [244].

Adakites in Alaska are also associated with some of the largest undeveloped resources of base and precious metals in the world. The Pebble porphyry Cu–Au–Mo deposit in southwestern Alaska (Peninsular terrane of the Jurassic-Pennsylvanian Wrangellia Composite Super-Terrane; Figure 17) contains 10.9 billion tonnes of ore grading 0.40% Cu, 0.34 g/t Au, 1.7 g/t Ag and 240 ppm Mo with substantial rhenium and palladium credits (36.9 Mt copper, 2.53 Mt molybdenum, 3054 t Au, 13,488 t Ag). The porphyry and skarn mineralization in the Pebble district is hosted in Cretaceous (91–89 Ma) Kaskanak batholith, which intrudes Early Cretaceous Kahiltna flysch and Jurassic arc-related volcanic units of the Talkeetna Formation [245]. The Kaskanak batholith is predominantly composed of equigranular granodiorite intersected by later-stage andesitic to dacitic (granitic) dikes and plugs "genetically linked to multiple centers of porphyry copper and skarn mineralization" [245]. The Kaskanak granodiorite is characterized by the porphyritic textures formed by plagioclase and amphibole with minor K-feldspar, quartz, biotite and magnetite. Mineralized porphyry dikes are crystal-rich and include plagioclase and amphibole as major phenocrystic phases. Most of the main phase granodiorite belongs to calc-alkaline intrusive series and exhibit pronounced HFSE depletions typical of island-arc magmas [245]. However, many granodiorites from the main phase and dacitic/granitic porphyry dikes exhibit elevated Sr/Y ratios (20–65) coupled with moderate LREE enrichments (La/Yb = 4–16) characteristic of normal arc compositions [245]. It is important to emphasize here that the highly evolved porphyry intrusions associated with Cu–Au–Mo mineralization display highest Sr/Y and La/Yb ratios and plot into adakite field in the Sr/Y–Y classification diagram (Figure 7A in [245]). Interestingly enough, many Kaskanak granodiorites (total of 42 analyses in [245]) plot below the adakite field in the Sr/Y–Y graph (Sr/Y < 30) but display characteristically low Y values (<15 ppm), indicating possible Sr loss during intense secondary alteration (mostly saussiritization of plagioclase) of Kaskanak granodiorites. In our opinion, the Kaskanak batholith provides another example of mixing between adakite (slab or lower crustal melts) and mantle-derived calc-alkaline magma that we already observed on the continent scale in the Andes of South America (Figure 15). It is quite revealing that ther mixed adakite–calc–alkaline Kaskanak batholith hosts a world class Cu–Au–Mo porphyry system in southern Alaska in the same manner as adakite-bearing igneous suites are associated with giant Cu–Au–Mo deposits throughout the Andean magmatic arc.

### 4.3. Central Asian Orogenic Belt

The Central Asian Orogenic Belt (CAOB) is the largest accumulation of accreted terranes (island arcs, back-arc basins, oceanic plateaus and islands, micro-continents, etc.) on our planet, stretching over 4000 km from the Ural Mountains to the Pacific Ocean and separating East European, Siberian, Tarim and Sino-Korean cratonic cores of the Eurasian super-continent [181,182]. The CAOB is richly endowed with Cu–Au–Mo porphyry and epithermal Au deposits of various size and Paleozoic to Mesozoic age [246,247], many of which are associated with adakite magmatism (Figure 18).

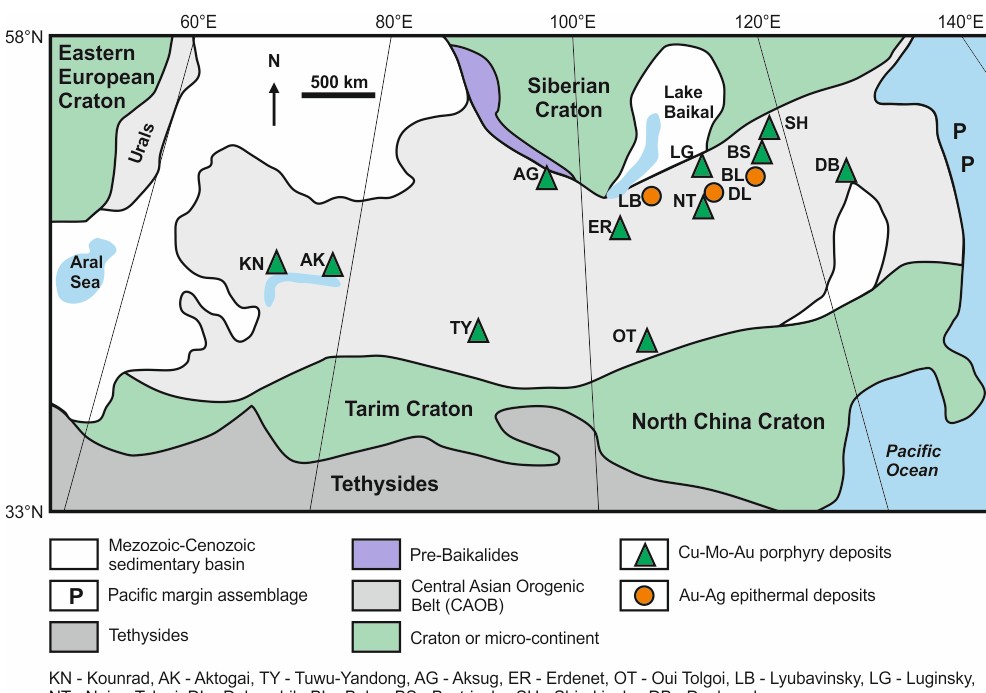

KN - Kounrad, AK - Aktogai, TY - Tuwu-Yandong, AG - Aksug, ER - Erdenet, OT - Oui Tolgoi, LB - Lyubavinsky, LG - Luginsky, NT - Noion-Tolgoi, DL - Delmachik, BL - Baley, BS - Bystrinsky, SH - Shirokinsky, DB - Duobaoshan

**Figure 18.** Distribution of major adakite-hosted Cu–Au porphyry (triangles) and A-uAg epithermal (circles) deposits within the Central Asian Orogenic Belt (CAOB). General tectonic setting of the CAOB and surrounding cratons is modified from [246].

### 4.3.1. Kazakhstan

The CAOB includes two principal structural oroclines, the Kazakhstan orocline and the Mongol-Okhotsk belt, both of which are characterized by adakite magmatism [246–258] (Figure 18). The Kazakhstan orocline is composed of Precambrian crustal fragments, Early (Cambrian to Early Silurian) and Late (Early Devonian to Carboniferous) Paleozoic magmatic arcs and accretionary wedges. Current geodynamic models suggest that the amalgamation of continental blocks and major magmatic arcs was completed by the Late Silurian and that a new Andean-type margin was formed in the Early Devonian as a result of the subduction beneath the edge of the older Kazakhstan orogen [248]. This Devonian-Carboniferous Andean-type margin includes a world-famous metallogenic province with several large (Bozhakol, Samarsk, Borly, Koksai) to giant (defined as having >2 Mt Cu, e.g., Aktogai and Kounrad) Cu–Au–Mo porphyry systems (Table 10) as well as numerous smaller porphyry and epithermal Cu–Au deposits and showings [246]. The giant Aktogai (12.5 Mt Cu; Table 10) deposit is centered over the Koldar granitoid pluton, which intrudes the Early Carboniferous volcanics. The Aktogai stockwork is associated with potassic (quartz, K-feldspar, biotite), phyllic (quartz, serecite, carbonate), boron-alumosilicate (tourmaline) and propylitic (chlorite, carbonate, zeolite, prehnite) alteration and has an elliptical shape measuring 2500 by 830 m [248]. The disseminated mineralization at Aktogai also includes subordinate quartz–K–feldspar–sulfide veins, which are thicker and more abundant in the potassic core and narrower within the outer phyllic and propylitic zones. Bornite occurs only within the potassic alteration core, while magnetite and pyrite dominate the outside alteration envelope [248]. Phyllic alteration also carries disseminated pyrite with minor molybdenite, sphalerite and galena. Anhydrite is common in all alteration zones. The tourmaline zone contains mineralized breccias with elevated copper grades [248]. The granodiorite porphyries hosting Aktogai Cu–Au mineralization display low Y (7.09–13.2 ppm) and Yb (0.71–1.22 ppm) concentrations combined with high Sr (521–923 ppm) contents and Sr/Y ratios over 40 (Figure 19).

**Table 10.** Main characteristics of selected adakite-hosted Cu–Au deposits in the CAOB.

| Deposit/Mine | Age | Mineralization | Resources | Refs. |
|---|---|---|---|---|
| | | *Kazakhstan* | | |
| Aktogai | 328–331 Ma | Potassic, phyllic and propylitic alteration associated with disseminated or vein mineralization | 3.25 Gt @ 0.4% Cu and 0.007–0.4 g/t Au (contained 12.5 Mt Cu and 2.7 Moz Au) | [248] |
| Kounrad | 325–327.3 Ma | Hypogene disseminated and stockwork (Py, Ccp, Bn, Cct, Tnt–Ttr, Eng, Mol, Sp, Gn, Apy); supergene (52 m; Cct-rich) | 800 Mt @ 0.61% Cu, 0.1–0.76 g/t Au, 0.0053% Mo | [249] |
| | | *Trans-Baikal* | | |
| Bystrinsky | 159–163 Ma | Lenticular and blanket-like skarns (Mag, Ccp, Py, Au, Po, Sch, Bn, Fe–Ni–Co arsenides, Bi–Te–S compounds), porphyry Qz–sulfide veins and disseminated (Ccp, Py, Sch, Mol, Au, Sp, Tnt–Ttr, Apy) | 300.9 Mt @ 0.3–16% Cu, 42–49% Fe, 0.054–0.23% Mo, 0.12–0.42% $WO_3$, 0.1–36 g/t Au, 5 g/t Ag | [250] |
| Novo-Shirokinsky | 140–156 Ma | Linear stockwork, veins, disseminated (Gn, Sp, Py, Tnt–Ttr, Ccp, Mol, Au, Apy, Bln, Jam, Brn) | 21.8 Mt@ 0.3% Cu, 3.5 g/t Au, 86.5 g/t Ag, 3.7% Pb (2.6 Moz of contained Au) | [251] |
| Baley (Taseevskoe) | Early Cretaceous | Low-sulfidation Qz–Cb–Adl veins (Au, El, Py, Mrc, Apy, Pyr, Myr, Brn, Ttr, Gn, Sp, Antm, Ste, Cal) | 31.1 Mt@ 5.1 g/t Au (5 Moz of contained Au) | [252] |
| | | *Tuva* | | |
| Aksug | 509–518 Ma | Multi-stage stockwork in phyllic (Qz–Ser) alteration (Bn, Ccp, Cct, Cu, Eng, Gn, Hem, Mag, Anh, Mrn, Mol, Py, Sp, Tnt–Ttr) | Indicated resource: 236 Mt @ 0.67% Cu, 0.019% Mo, 0.18 g/t Au, 0.99 g/t Ag and 0.29 g/t Re; Inferred resource: 486 Mt @ 0.37% Cu, 0.07 g/t Au, 0.008% Mo and 0.16 g/t Re | [247] |
| | | *Mongolia* | | |
| Erdenet | 220–242 Ma porphyry; 240–241 Re-Os molybdenite | Stockwork - 2 by 1 km surface; 560 m vertical including 100 to 300 m supergene zone (Ccp, Py, Cv, Cct, Bn, Mol, Tnt–Ttr, Gn, Sp, Anh, Mag) | 1490 Mt @ 0.509% Cu and 0.015% Mo (contained 7.65 Mt copper and 216,600 t Mo) | [246,247,255] |
| Oyu Tolgoi | 372–374 Ma | Qz vein stockwork (Ccp, Bn, Cct, Cv, Eng, Cu, Ccl, Mlc, Hem, Mag, Py, Pyr, Ser, Sp, Tnt, Tnr) in potassic and argillic alteration | 3755 Mt @ 0.98% Cu, 0.01% Mo, 0.38 g/t Au (37 Mt of contained copper) | [256] |
| | | *China* | | |
| Tuwu-Yandong (East Tianshan, NW China) | 332–341 Ma | Ductile veins and disseminated (Ccp, Py, Bn, Cct, Cv, Mag, Hem, Mol, Mlc, Atc) | 674 Mt @ 0.61% Cu, 0.01% Mo, 0.1 g/t Au, 1.28 g/t Ag (4.1 Mt contained copper) | [246,257] |
| Duobaoshan cluster (Greater Xin'gan, NE China) | 478–485 Ma and 223–244 Ma | E-w-trending vein clusters, 100–1400 m long, 2–390 m wide (Py, Ccp, Bn, Mol, Po, Gn, Sp, Ttr, Au, Dg, Mlc) | Contained 2,44 Mt Cu (av. 0.52% Cu), 110,00 tons Mo (av. 0.02% Mo), 2.4 Moz Au (av. 0.16–0.35 g/t Au), 34.8 Moz Ag | [258] |

Mineral abbreviations: Qz—quartz, Cb—carbonates, Adl—adularia, Zeo—zeolite, Hem—hematite, Py—pyrite, Mrc—marcasite, Mag—magnetite, Po—pyrrhotite, Anh—anhydrite, Ser—sericite, Apy—arsenopyrite, Ccp—chalcopyrite, Cv—covellite, Cct—chalcocite, Bn—bornite, Dg—digenite, Tnr—tenorite, Eng—enargite ($Cu_3AsS_4$) Ccl—chrysocolla, Mlc—malachite, Tnt–Ttr—tennantite–tetrahedrite, Sp—sphalerite, Gn—galena₃, Antm—antimonite, Mol—molybdenite, Cin—cinnabar, Argn—argentite, Sch—scheelite, Pol—polybasite (Ag–Cu–Sb–As sulfosalt), Hes—hessite ($Ag_2Te$), Ptz—petzite ($Ag_3AuTe_2$), Slv—sylvanite (Au, Ag)$Te_2$, Pyr—pyrargyrite ($Ag_3SbS_3$), Myr—miargyrite ($AgSbS_2$), Alt—altaite ($PbTe$), Cal—calaverite ($AuTe_2$), Mrn—merenskyite (Pt, Pd) (Te, Bi)$_2$, Atc—atacamite ($Cu_2Cl(OH)_3$), Bln—boulangerite ($Pb_5Sb_4S_{11}$), Jam—jamesonite ($Pb_4FeSb_6S_{14}$), Brn—bournonite ($PbCuSbS_3$), Ste—stephanite ($Ag_5SbS_4$), El—electrum, Au—native gold, Ag—native silver, Cu—native copper. Moz—million ounces.

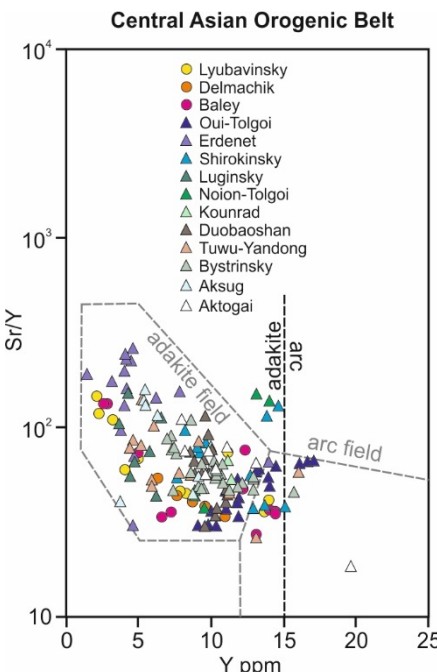

**Figure 19.** Variations of Sr/Y versus Y (ppm) in igneous rocks associated with porphyry Cu–Au–Mo (triangles) and epithermal Au–Ag (circles) deposits in the Central Asian Orogenic Belt. Data sources: Lyubavinsky [253]; Delmachik [254]; Baley (Taseevsky) [252]; Oui-Tolgoi [256]; Erdenet [246,255]; Shirokinsky [251]; Luginsky [252]; Noion-Tolgoi [252]; Kounrad [249]; Aktogai [246,248]; Duobaoshan [258]; Tuwu-Yandong [246,257]; Bystrinsky [250]; Aksug [247].

These classic adakite geochemical features of Aktogai granitoids are further emphasized by pronounced negative Eu anomalies, indicating a garnet-dominant mafic source for mineralized Aktogai magmas [248]. Petrogenetic modeling and Sr–Nd–Pb–Hf isotope considerations suggest the derivation of Aktogai adakites from the juvenile lower crust of the Carboniferous Andean-type magmatic arc [248]. Another world-class Kounrad Cu–Au–Mo system within the Andean-type paleo-margin of Central Kazakhstan is hosted in Carboniferous adakitic granodiorite porphyries (Figure 19) associated with inner silicic and outer propylitic alteration [249]. This metasomatic zonation is overprinted with argyllic and sericitic (phyllic) metasomatic assemblages [249]. The Kounrad orebbody is elongated in the northeastern direction and has the surface exposure measuring 1000 by 700 to 800 m. The disseminated and stockwork hypogene mineralization at Kounrad is composed of pyrite, chalcopyrite, bornite, mlybdenite, enargite, tetrahedrite and chalcocite with minor sphalerite, galena, arsenopyrite and tennantite (Table 10). Hypogene ore is overlain by a 52 m thick, chalcocite-dominated supergene enrichment blanket [249]. The Kounrad Cu–Au–Mo system displays clear mineral zonation with bornite and chalcopyrite dominating its central part and pyrite–chalcopyrite–galena and sphalerite mineralization occurring within its peripheral parts [249]. Many other Cu–Au–Mo deposits within the Kazakhstan orocline, independently of their size, grade and metal endowment, are associated with felsic magmatic rocks with pronounced adakitic geochemical signature (Figure 19; [246,247,249]).

### 4.3.2. Trans-Baikal Region

The Trans-Baikal segment of the CAOB (the Mongol-Okhotsk orogenic belt) extends from Lake Baikal towards the borders of Mongolia and North China (Figure 18). The Trans-Baikal region records a protracted history of ocean basin openings and closures, terrane accretions and major collision episodes associated with intense regional deformation, metamorphism and plutonism [251]. For example, the final episodes of the subduction of the Paleo-Asian ocean floor produced the 339–285 Ma granitic Angara-Vitim batholith, while the Triassic-Early Jurassic subduction of the Mongol-Okhotsk Ocean resulted in

widespread granitoid magmatism through the entire Trans-Baikal region [247]. The Late Jurassic-Early Cretaceous post-collision extension that followed the final closure of the Mongol-Okhotsk Ocean is marked with prolific calc-alkaline to shoshonitic and alkali basaltic volcanism as well as ultramafic/mafic to felsic plutonism associated with numerous Cu–Au–Ag–Mo–polymetallic porphyry/skarn/epithermal deposits and mineral showings [247,250,251]. Adakites are quite common through out the Trans-Baikal region, where they host several large Cu–Au–Mo porphyry, skarn and epithermal deposits (Figures 18 and 19; Table 10; [247,250–253]. One of the largest mineralized systems closely associated with adakite magmatism in the Trans-Baikal region, is the giant Bystrinsky Cu–Fe–Mo–Au deposit located in the eastern part of this metal-rich region (Figure 18; Table 10; [251]). Both skarn and porphyry-style mineralization are centered on the Bystrinsky polyphase granitoid pluton. Most of the Bystrinsky metal resource is hosted in lenticular and blanket-like ore bodies several tens of meters thick and up to 700–800 m long. The porphyry mineralization is associated with garnet–diopside exoskarns developed along the contact between Bystrinsky granitoids and carbonate country rock [251]. Massive, veinlet-disseminated and nested mineralization is composed mostly of magnetite, chalcopyrite, pyrite, pyrrhotite, scheelite and native gold with minor molybdenite, wolframite, sphalerite, arsenopyrite, bornite, Fe–Ni–Co arsenides and sulfoarsenides, hematite, valleriite and Bi–Te–S compounds [251]. Porphyry low-grade (0.20–0.33 wt.% Cu and 0.4–0.7 g/t Au) quartz-sulfide (chalcopyrite, pyrite, scheelite, molybdenite, with minor tennantite–tetrahedrite, sphalerite and native gold) veins and disseminated sulfide zones locally display cross-cutting relationships with the higher-grade skarns and are spatially associated with the late-stage felsic dikes and stocks [251]. Both the main phase monzodiorites and monzonites and the late-stage granodiorites display classic adakite geochemical characteristics, such as high Sr (340–900 ppm) and low Y (7.2–15.6 ppm) contents coupled with high Sr/Y ratios (44–111; Figure 16). The petrogenetic model for Bystrinsky adakite involves early amphibole fractionation of mantle-derived, hydrous mafic magma followed by limited interaction with lower crust and mixing with previously evolved melt batches in common lower crustal magma chamber [250].

Some adakite-hosted deposits in the Trans-Baikal region represent high-level Au–Ag or vertically zoned, telescoped porphyry/high-sulfidation Au-dominated polymetallic environments [252–254]. One of the best-known examples of this type is the Shirokinsky cluster (Figure 18), which includes the Novo-Shirokinsky gold-polymetallic deposit, currently a 0.8 million tonnes per annum (mtpa) operation by Highland Gold (Table 10). Novo-Shirokinsky is essentially a polymetallic deposit with commercial quantities of gold, silver, lead and zinc. Mineralization is hosted in Mid- to Late Jurassic (156–175 Ma) volcanic-sedimentary sequence intruded by granodiorite and quartz diorite porphyry dikes and stocks [251]. The deposit consists of 17 en-echelon ore bodies, with the 3 largest ore bodies constituting 91% of the resource. Gold-polymetallic mineralization is accompanied by four main types of alteration (from the earliest to the latest): (1) propylitic, (2) chlorite–dolomite, (3) serecite–dolomite and 4) quartz–dolomite. Based on mineralogy and geologic relationships, the Novo-Shirokinsky mineralzation was assembled in four principal stages: (1) tourmaline–pyrite (pyrite, chalcopyrite, molybdenite, melnicovite–colloform pyrite); (2) copper–sulfosalt–pyrite (pyrite, chalcopyrite, sphalerite, galena, native gold, tennantite–tetrahedrite, bournonite, jamesonite, bismuthite); (3) quartz-polymetallic (pyrite, sphalerite, galena, chalcopyrite, boulangerite, magnetite, native gold, hematite, bournonite, arsenopyrite) and (4) carbonate-polymetallic (sphalerite, galena, pyrite, native gold, marcasite, arsenopyrite, boulangerite) [251]. Gold grades are highest (up to 186 g/t Au) in the copper–sulfosalt–pyrite ores and lowest (0.1–16 g/t Au, average of 0.8–1.4 g/t Au) in the latest, carbonate–polymetallic mineralization. Gold (n × 0.001—n × 0.1 mm in size) is usually included in pyrite, chalcopyrite, tennantite–tetrahedrite and galena [251]. Quartz diorite and granodiotite porphyry intrusions at the Novo-Shirokinsky deposit display typical adakitic signatures (high Sr, low Y and Yb, high Sr/Y and La/Yb; Figure 19) and are interpreted as fractionated mantle-derived melts, or partial melts from either subducted slab, or

delaminated continental crust [251]. Several epithermal gold deposits in the Trans-Baikal region are also associated with adakite magmas. The Lyubavinsky low-sulfidation gold deposit (Figure 18; Table 10) is hosted in Jurassic (143–180 Ma) gradnodiorite and quartz porphyry dikes and stocks intruded into Devonian-Triassic clastic metasediments [253]. The mineralization is represented by high-grade (0.5–600 g/t Au, average of 45.1 g/t Au) quartz, quartz–micirocline and quartz–adularia veins containing minor pyrite, arsenopyrite, chalcopyrite and molybdenite [253]. Shallow porphyry intrusives are classified as adakites (Y = 3.6–13.7 ppm, Yb = 0.3–1.3 ppm; Sr/Y > 40; Figure 19), and their origin is assigned to the partial melting of the subducted oceanic lithosphere in a Jurassic subduction zone environment [253]. Another low-sulfidation Delmachik epithermal system in the Eastern Trans-Baikal region (Figure 18) is associated with Late Jurassic granodiorite and granite porphyry dikes developed within the eroded Middle to Late Jurassic nested caldera complex [254]. Gold (averaging around 3.6 g/t) is localized in quartz-sulfide (<5% sulfide by volume) veins (up to 30 cm wide), diatreme-hosted, stockwork-type metasomatic breccias (45 by 55 m in size) and shear zones (up to 70 m wide) in association with pyrite, chalcopyrite, stibnite, arsenopyrite, molybdenite, galena, sphalerite and Bi-sulfosalts [254]. Host adakite dikes (Figure 19) are characterized by low Y (6.6–14.8 ppm) and Yb (0.74–1.5 ppm) contents and were probably formed through the mixing of evolved mantle-derived and crustal melts [254].

### 4.3.3. Tuva and Mongolia

The eastern segment of the CAOB (Figure 18) includes numerous Cu–Mo–Au deposits, prospects and showings that can be grouped on the basis of the emplacement ages of porphyry intrusions (Table 10) into three ore belts (terminology after [247]): (1) the Early to Middle Paleozoic Altai-Sayan belt (including Kuznetsk Alatau and Tuva) west of Lake Baikal; (2) the Late Paleozoic to Early Mesozoic Mongolia; and (3) the Mesozoic Trans-Baikal region (see above). Large deposits in the Altai-Sayan region are related to Cambrian-Early Devonian intrusive magmatism and include the operating Sora Mo–Cu mine (remaining reserves after decades of mining activities stand at 111,000 tons of molybdenum at 0.06% Mo plus Cu and Ag credits) in the Kuznetsk Alatau Range (not shown in Figure 18) and two large Cu–Mo porphyry systems of Aksug (Figure 18) and Kyzyk-Chadr in the Autonomous Republic of Tuva [247]. The Aksug porphyry system is localized within the package of the Vendian-Cambrian accreted oceanic, back-arc and island-arc volcanic-sedimentary accreted terranes, as well as Neoproterozoic and Early Paleozoic ophiolites [247]. The Aksug deposit was discovered in 1952 and is one of the largest porphyry resources in Russia. The stockwork mineralization is centered on the Early Cambrian (Table 10) Aksug quartz diorite/tonalite pluton, while disseminated copper–molybdenum mineralization is associated with later-stage porphyry dikes and stocks, which can be further sub-divided into two groups [247]. Group I quartz–diorite–grandiorite porphyries are associated with low-grade (Cu = 0.1–0.2% with ~0.003% Mo) molybdenite–pyrite–chalcopyrite mineralization, while Group II granodiorite–tonalite porphyries host high-grade (Cu = 0.3–1.0%, Mo = 0.01–0.02%) chalcopyrite–bornite–molybdenite mineralization apparently superimposed onto the low-grade ore [247]. The main ore mineral assemblages in the Aksug deposit [247] are (from the earliest to the latest): (1) quartz–pyrite with hematite; (2) quartz–molybdenite with pyrite and chalcopyrite; (3) quartz–chalcopyrite with bornite, pyrite and molybdenite; and (4) sulfides and sulfosalts with tennantite–tetrahedrite, enargite, galena and sphalerite. Both the Aksug pluton granitoids and the later-stage granodiorite/tonalite prophyries exhibit low Y (3.9–8.8 ppm) and Yb (0.40–0.88 ppm) contents coupled with high Sr/Y (40–151) ratios (Figure 16; Table 1 in [247]) and represent adakite melts generated via partial melting of the basaltic lower crust in the Lower Cambrian subduction environment [247].

Two world-class adakite-related Cu–Au porphyry systems in Mongolia include Erdenet in the northern part of the country and Oyu Tolgoi in the Gobi Desert of southern Mongolia (Figure 18; Table 10). The Early to Middle Triassic Erdenet deposit is hosted in amphibole–biotite–plagioclase–phyric granodiorite porphyry intruded by multiple, later-

stage microgranodiorite dikes [247,255]. Most felsic rocks at Erdenet are Y (3.2–13.9 ppm) and Yb (0.4–1.4 ppm) depleted and characterized by generally very high Sr/Y ratios (>100; Figure 19), typical of fertile porphyry magmas [84–90]. The alteration zonation consists (from core to rim of the system) of early potassic (quartz–muscovite; most important), intermediate argyllic (illite ± kaolinite-smectite + anhydrite) and propylitic (chlorite–epidote) metasomatic mineral assemblages [255]. The Erdenet porphyry system is centered on a composite granodiorite stock approximately 900 m in diameter exposed in the southeast part of the current open pit. Hypogene mineralization includes pyrite–chalcopyrite–bornite, quartz–molybdenite and pyrite–chalcopyrite veins. The oxide cap (malachite, azurite, turquoise, powellite; now completely removed by mining) and underlying bornite–chalcocite supergene enrichment zone are characterized by the highest average copper grades of ~0.76% Cu (locally exceeding 5% Cu) and constitute economically the most important part of the Erdenet deposit [255]. Another super-giant Oyu Tolgoi porphyry Cu–Au cluster (Figure 18) forms a >6.5 km long mineralized trend related to ~373 Ma tectono-magmatic event associated with the Devonian to Carboniferous (390–320 Ma) Gurvansayhan magmatic arc in southern Mongolia [256]. The Oyu Tolgoi trend includes several Cu–Au porphyries and high-sulfidation deposits, such as South Oyu, West Oyu, Central Oyu and the Hugo Dummett zone (formerly Far North Oyu). Both the porphyry and the high-sulfidation mineralization are hosted in the Late Devonian volcano-sedimentary sequence of basalt–andesite–dacite lavas and ash flow tuffs intruded by high-K quartz monzonite (~372 Ma) and granodiorite (~366 Ma) [256]. The high-grade core of Southwest Oyu (250 m in diameter with 800 m vertical extent) is centered on syn- to late mineral porphyritic quartz monzodiorite associated with intense albite-biotite alteration overprinted by phyllic (quartz + serecite ± fluorite ± tourmaline) metasomatic assemblages [256]. Mineralization is composed of fine gold (1 to 120 microns) frequently intergrown with chalcopyrite filling fractures in pyrite or occurring as inclusions within or along the boundaries of chalcopyrite and bornite grains. High-sulfidation systems occur above and are partly telescoped onto the underlying porphyry at Central Oyu Tolgoi and Hugo Dummett deposits [256]. A supergene chalcocite-rich blanket overlies high-sulfidation (covellite, pyrite) mineralization at the Central Oyu system. Porphyry-style bornite–chalcopyrite–chalcocite-rich quartz veins dominate the mineralization at South Hugo and extend vertically downwards into a biotite and chlorite-altered porphyry sulfide (bornite, chalcopyrite, chalcocite) core with local grades of 10% Cu over 2 m sampling intervals [256]. North Hugo includes a high-grade (Cu > 1%), bornite-dominated core associated with quartz monzonite stocks. Porphyry intrusions throughout the Oyu Tolgoi mineral trend display high Sr/Y ratios (Figure 19) coupled with quite variable Yb contents and La/Yb ratios and "generally fit the major element definition for adakites" [256]. Late Devonian, syn-mineral adakite magmas in the Oyu Tolgoi cluster most probably represent melting of young and hot subducted oceanic crust shortly after the inception of the Gurvansayhan magmatic arc at ~390 Ma [256]. The presence of high-Nb basaltic lavas ($TiO_2$ = 1.79 wt.%; $Na_2O+K_2O$ = 6.71 wt.%; Nb = 47.4 ppm; Table 6.2 in [256]) in volcanic sequence hosting the Oyu Tolgoi mineralization suggests that these metal-rich slab melts interacted with mantle wedge peridotite to form hybrid, HFSE-enriched mantle sources beneath southern Mongolia similar to modern HNB provinces of Kamchatka, Philippines and Central America [52,56,58,59,131].

### 4.3.4. Northern China

Adakites are widespread in Northern China, where their prolific crustal emplacement contributed to the destruction of the North China cratonic lithosphere through rapid delamination, thermal/chemical erosion and hydration by slab-related fluids [73,106,257–259]. Adakites are spatially and temporally associated with the Cu–Au deposits in the eastern segment of the CAOB (Figure 18) including the eastern Tianshan and other structural units of NW China [64,86,107,111,116,246,257]. For example, adakites host large Cu–Mo–Au porphyry systems (Daheishan, Yili, Duobaoshan) in NE China (Figure 18), where they are commonly linked to the flat-slab subduction of the Paleo-Pacific Plate [119,258,260–262].

Mineralized adakites are closely associated with high-Nb basalts at several locations in both NW and NE China [64,260]. The Tuwu-Yandong Cu–Au porphyry cluster (Figure 18) is located in the eastern Tianshan, Xinjiang, China [246,257], and contains over 4 million tons of copper averaging 0.61% Cu (Table 10). The copper–gold mineralization is associated with Early Carboniferous tonalite porphyry, which intrudes Lower Carboniferous basaltic and andesitic volcanics [246]. Both tonalite porphyry and surrounding mafic to intermediate extrusive rocks underwent intense potassic, intermediate argillic and phyllic alteration. Mineralization at both Tuwu and Yandong deposits (which are 8 km apart and poorly exposed) is represented by linear (elongated) stockwork zones [247]. Disseminated and veinlet-hosted ore minerals include chalcopyrite, pyrite, bornite, chalcocite, magnetite, sphalerite and hessite (Table 10). The ore-bearing porphyry intrusions contain quartz, plagioclase and biotite phenocrysts in the groundmass composed of quartz, plagioclase and hornblende [246,247]. The Tuwu-Yandong plagiogranite–tonalite porphyry displays low Y (3.4–6 ppm) and Yb (0.4–1.4 ppm) and high Sr (347–920 ppm; average of 563 ppm) contents along with high Sr/Y ratios (Figure 19) and MORB-like, low $^{87}Sr/^{86}Sr$ isotope values typical of slab-derived adakites [247]. Several detailed petrological studies suggest that the Tuwu-Yandong adakites were produced via partial melting of subducted oceanic lithosphere and that "the ore-bearing plagiogranite porphyries formed during fast and oblique convergence between the paleo-Tianshan ocean and the Dananhu arc" [247].

The northeastern China region (easternmost portion of the CAOB; Figure 18) is composed of various microcontinental blocks (Erguna, Xing'an, Songliao and Jiamusi) separated by several ophiolite-bearing suture zones (Xinlin, Heilongjiang, Hegenshan-Heihe). The region is richly endowed with large to giant Cu–Mo–Au porphyry deposits, some of which are associated with adakitic rocks [260–262]. The giant Duobaoshan Cu–Au–Mo ore field is associated with Ordovician, Silurian, Devonian and minor Carboniferous, Permian, Triassic and Cretaceous volcanic (intermediate to felsic lavas and volcaniclastic rocks) and sedimentary (tuffaceous sandstone and siltstone with subordinate marble and slate) rock formations [258]. The volcano-sedimentary formations are intruded by granodiorite, granite, tonalite, monzogranite and quartz diorite porphyries. The copper–gold mineralization within the Duobaoshan ore field is mostly hosted in granodiorite and is primarily associated with phyllic (serecite-dominated) alteration [258]. Other alteration styles include a potassic core (centered on porphyry intrusion) and an outer propylitic envelope. Ore bodies generally have banded or lenticular shape and dip in northwestern directions (220–240°) with angles around 70–80° [258]. The ore minerals are chalcopyrite, bornite, pyrite and molybdenite (Table 10), while the gangue minerals include quartz, serecite, chlorite and carbonate with minor epidote, biotite and K-feldspar. The potassic (K-feldspar, quartz, biotite) core is usually rich in molybdenite, the phyllic (quartz, serecite, chlorite) alteration hosts high-grade quartz–molybdenite and the quartz–chalcopyrite–pyrite veins' propylitic (chlorite, carbonate, epidote, amorphous silica) envelope is associated with low-grade disseminated quartz–calcite–pyrite mineralization [258]. Cambrian-Ordovician and Triassic adakite-like rocks (Figure 16) within the Duobaoshan ore field originated in the amphibole to garnet stability field (>40 km), but fractionated and re-equilibrated at shallower (35–40 km) crustal levels [258]. Trace element geochemistry and Sr–Nd–Hf isotopic signature of the oldest (Cambrian-Ordovician) adakites at Duobaoshan suggest their derivation from the subducted Paleo-Asian oceanic crust. The ore-causative Early-Middle Ordovician and Middle-Late Triassic adakites were formed, as proposed in [258], through partial melting of the thickened lower crust. In our opinion, trace element composition of the "older" (Ordovician, pre-mineralization) and "younger" (Triassic, syn-mineralization) adakites is almost identical (Figure 19), suggesting that both felsic igneous suites are related to oceanic slab melting. We suggest that Ordovician (485–473 Ma) adakites were produced through the northwestward subduction of the Paleo-Asian ocean crust beneath the Erguna and Xing'an continental terranes, while Triassic (246–229 Ma) adakite is related to the southeastward subduction of the Mongol-Okhotsk oceanic floor beneath the Erguna and Xing'an crustal blocks (cf. Figure 17 in [258]). We also suggest that the formation of

the Duobaoshan ore field on the southern margin of the Late Triassic Mongol-Okhotsk Ocean mirrors the tectono-magmatic processes that led to the emplacement of metal-rich adakites in the Stanovoy Suture Zone on the northern side of the same oceanic basin. In both cases, adakites played an instrumental role in the formation of large-scale Cu–Au porphyry mineralization on the "outskirts" of the Mongol-Okhotsk Ocean.

### 4.4. The Tethyan Belt

The Tethyan orogenic belt stretches from northwestern Africa and Western Europe to the mountain ranges of the Balkans, Asia Minor (Anatolia), Zagros, Caucasus, Hindukush, Pamir and Tibet. This is an extremely fertile metallogenic environment, which includes various mineralization styles formed in diverse tectonic settings [263]. Adakites are found at different age levels in the Tethyan belt and are frequently (Iran, Tibet) associated with high-Nb basalts [65,115,137,264–266] and Cu–Au deposits [110,267–275]. Petrologic models for Tethyan adakites and high-Nb basalts include ridge subduction and slab melting followed by mantle hybridization and formation of HFSE-enriched sources, as well as the melting of thickened mafic lower crust (especially in Tibet), crustal delamination and subduction modification (basification) of continental crust.

#### 4.4.1. Tibet

Adakites are widespread in crustal terranes of all ages within the western, northern, southern, central and easterns parts of the Tibetan plateau, where they were formed through a range of petrogenetic processes including slab melting, the partial melting of underplated and rifted continental crust and high-pressure (amphibole $\pm$ garnet) fractionation of hydrous, mantle-derived magmas [65,68,69,71,72,110,112,114,125]. At several locations in Tibet, adakites are closely associated with magnesian andesites and high-Nb basalts suggesting intense adakite–peridotite interactions in the sub-Tibetan mantle [65,266,268]. Most adakite manifestations are closely linked with Cu–Au porphyry-style mineralization (Figure 20; [271–274]). The Tibetan orogen, created by protracted Cenozoic collision between the Eurasian super-plate and Indian continental block, is characterized by four principal tectonic epochs associated with large-scale ore formation [276]: (1) pre-collision stage related to the northward subduction of the Bannu Ocean (Early Jurassic to Cretaceous); (2) the main collisional stage (65–41 Ma); (3) the late-collisional transform structural setting (40–26 Ma); and (4) the post-collisional crustal extension (25 Ma—recent). The collisional belt of Tibet includes several world-class Cu–Mo–Au belts [268,272,276]: (1) the Yulong porphyry Cu belt in eastern Tibet; (2) the Gangdese porphyry Cu belt in south Tibet; and (3) the Duolong mineral district in western Tibet (Figure 20). The Yulong porphyry belt was developed along the eastern margin of the Tibetan Plateau almost perpendicular to the general longitudinal arrangement of the Tibetan crustal terranes (Figure 20). The porphyry mineralization is contained within a 300 by 15–30 km structural corridor and appears to be controlled (bounded on both western and eastern side) by a major regional strike-slip fault system [277]. The richest copper resource of 8 million tons of copper is associated with 5 major Late Eocene (41.2–36.9 Ma) porphyry deposits (Figure 20; Yulong—630 Mt @ 0.99% Cu, 0.028% Mo, 0.35 g/t Au including 60 Mt @ 4.74% Cu, 4.5 g/t Au; Malasongduo—230 Mt @ 0.44% Cu, 0.014% Mo, 0.06 g/t Au; Duoxiasongduo—248 Mt @ 0.38% Cu, 0.04% Mo, 0.06 g/t Au; Mangzong—70 Mt @ 0.34% Cu, 0.03% Mo, 0.02 g/t Au; Zhanaga—80 Mt @ 0.36% Cu, 0.03% Mo, 0.03 g/t Au) associated with adakites intruding the Triassic sandstone and limestone [277]. Trace element characteristics of Yulong adakite (Y < 10 ppm; Sr/Y > 50; Yb < 1.2) along with high $K_2O$ contents, high initial $^{87}Sr/^{86}Sr$ ratios and low $\varepsilon Nd(t)$ suggest their derivation from the thickened juvenile lower crust of the Tibetan Plateau [278].

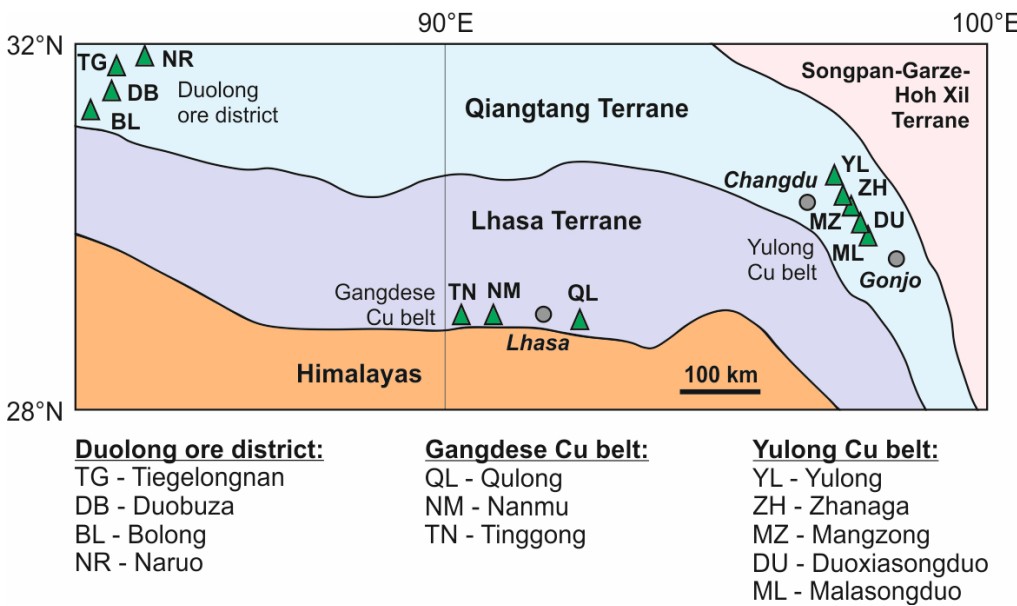

**Figure 20.** Distribution of major adakite-hosted Cu (Au, Mo) porphyry deposits (triangles) in Tibet. (Eastern Tethyan Belt). Crustal terrane nomenclature and geometry is after [263,268].

The Miocene Gangdese porphyry copper belt occurs within the Lhasa terrane of southern Tibet (Figure 20), which straddles the Indo-Asian collisional boundary [272]. The K-adakitic, shoshonitic and ultra-potassic magmatism associated with mineralization is dated at 11.5 to 19.7 Ma, while molybdenites yielded somewhat younger Re-Os ages of 13.8 to 14.8 Ma [279]. The mineralized Miocene porphyry stocks intrude the Gangdese granitic batholiths and Triassic to Tertiary sediments and form a 400-km long and 50-km wide, EW-striking mineral trend (Figure 20) parallel to the central thrust zone of the Lhasa terrane [272]. Most porphyry stocks are small (<5 km in diameter), coalesced together and emplaced at shallow depths of 1 to 3 km below the paleo-surface. Mineralized explosive breccias surround some granodiorite stocks at Qulong and Nanmu deposits [279]. The Gangdese mineral belt can be further sub-divided into two mineral trends [272]: (1) a southern trend dominated by porphyry systems almost exclusively intruded into the Gangdese batholiths and (2) a northern trend composed of skarn polymetallic deposits and mineralized Miocene adakites. Adakite stocks intrude the Permian-Tertiary volcano-sedimentary sequences and are controlled by the central Lhasa thrust zone (Figure 20). Mineralization in the northern sub-trend is quite diverse and includes skarn-type, lenticular Pb–Zn–Cu ore bodies hosted in Triassic limestone [272]. The largest Qulong porphyry deposit (Figure 20) in the southern trend contains around 7 million tons of copper at an average grade of 0.5% Cu. Mineralization is localized at the intersection of EW-trending, north-dipping thrust faults and NNE-striking normal faults and is centered over the Early Miocene (17.58 ± 0.74 Ma) monzogranite intruded into Middle Jurassic felsic tuff, limestone, sandstone and slate and Paleocene granodiorite [272]. Alteration includes a potassic (biotite, K-feldspar, anhydrite) core surrounded by phyllic (quartz, serecite, pyrite), patchy argillic (quartz and kaolinite) and external propylitic (chlorite, epidote, calcite) types. The Cu–Mo mineralization in the potassic core includes pyrite–chalcopyrite–molybdenite–bornite and quartz–molybdenite veinlets as well as chalcopyrite–bornite–sphalerite–galena stockwork. The supergene blanket is composed of malachite, covellite and minor chalcocite [272]. The Cu–Mo mineralization is also structurally controlled at the Tinggong deposit (Figure 20), where porphyry stocks intrude Paleocene-Oligocene andesite–dacite volcanics, welded felsic tuff and tuffaceous sandstone [272]. The Tinggong mineralized system contains over 1 million tons of copper at 0.5% average Cu centered over monzogranite stock enveloped in potassic (K-feldspar, biotite, magnetite, quartz), phyllic (quartz, serecite) and argillic (kaolinite) roughly concentric alteration zones. Principal ore minerals include chalcopyrite,

pyrite, chalcocite and molybdenite with minor bornite, galena, malachite, covellite and limonite [272]. All Miocene porphyry stocks within the Gangdese Cu belt possess general adakitic geochemical characteristics but display some systematic temporal variations in $(La/Yb)_N$, Zr/Sm and Sm/Yb ratios [280]. It was earlier proposed on the basis of the Andean data, that adakites produced by the dehydration (amphibole breakdown) melting of a mafic source display Sm/Yb ratios of 5 to 7 [217]. The pre-mineralized porphyries in the Gangdese belt are characterized by high Zr/Sm and low Sm/Yb and Y, suggesting a garnet-rich, amphibole eclogite (garnet > amphibole) source [280]. The depletion of this source in hydrous phase (amphibole) limits fluid supply to the upper crustal environments. The syn-mineralization adakites display Sm/Y of 4.6–7.9 and Zr/Sm of 14–50, which is characteristic of dehydration melting of garnet-bearing amphibolite (amphibole > garnet) in the lower crust accompanied by the release of metal-rich, hydrous fluids [280]. The temporal evolution of the lower crustal source from mostly eclogitic to mostly amphibolitic plays an important role in the formation of fertile adakitic magmas and associated Cu–Au(Mo) porphyry systems. Older (Cretaceous, ca. 90 Ma) adakites in the Gangdese belt are associated with smaller (compared to Miocene mineralization) skarn and vein-type Cu–Au deposits (Kelu, Shuangbujiere, Liebu, Chenba and Chongmuda) composed of bornite, chalcopyrite, chalcocite, cuprite, malachite, galena, native gold, Au–Ag-tellurides, wittichenite ($Cu_3BiS_3$) and bismutite [267]. Older adakites are believed to be products of oceanic slab melting in a Cretaceous subduction zone environment [267].

The Duolong mineral district is located in the Southern Qiangtang terrane of West Tibet (Figure 20) and hosts some of the largest Cu–Au porphyry deposits in China associated with Early Cretaceous (122.5 Ma) adakite and high-Nb basalt [268]. The district geology is dominated by Jurassic clastic sediments of the Mugagangri Group, Lower to Middle Jurassic sandstone and siltstone slope deposits and the Upper Cretaceous post-collisional molasses. Mesozoic sediments are intruded by numerous Early Cretaceous (128–106 Ma) granitoids and east–west-trending mafic dikes hosted exclusively within the Jurassic Mugagangri group [268]. Preliminary estimates of proven resources in the Duolong district stand at >20 million tons of copper accompanied by >500 tons of associated Au resources [268]. The Duolong mineral district is approximately 50 km long and 20 km wide (Figure 20) and includes one giant (2 Bt of ore at 0.53% Cu, 37 t Au at 0.13 g/t and over 2600 t Ag at 1.78 g/t) Tiegelongnan Cu–Au porphyry-epithermal system and several smaller porphyry deposits at Duobuza (648 Mt of ore at 0.46% Cu), Bolong (640 Mt of ore at 0.43% Cu) and Naruo (660 Mt of ore at 0.38% Cu; >2.5 Mt Cu; >80 t Au and >800 t Ag) along with several smaller porphyry copper deposits and mineralized prospects (Figure 20). The Naruo porphyry system (Figure 20) is centered on an Early Cretaceous (122–120 Ma) granodiorite porphyry stock, which intruded the Jurassic sandstones [281]. The granodiorite is associated with potassic and propylitic alteration at depth and patchy argillic alteration closer to the surface. Fine-grained phyllic alteration follows some hydrothermal veins and locally overprints potassic or propylitic alteration [281]. Hydrothermal magnetite, quartz–magnetite and quartz–K-feldspar veins contain variable amounts of pyrite, chalcopyrite, molybdenite and bornite. Breccia-hosted mineralization occurs within the western portion of the Naruo deposit (Figure 20). Sedimentary breccia surrounds the granodiorite porphyry and carries mostly pyrite with minor chalcopyrite, galena and sphalerite. Magmatic breccias are localized in the bottom granodiorite and diorite roof of the porphyry intrusion and are associated with pervasive silicic and propylitic alteration [281]. Ore minerals include pyrite, chalcopyrite, molybdenite with minor galena, sphalerite, bornite and magnetite. The porphyry-epithermal mineralization at Tiegelongnan is hosted in 120–121 Ma old granodiorite intrusions emplaced into the Lower Jurassic arcose sandstone [281]. The alteration is represented by the inner potassic core surrounded by the intermediate phyllic and outermost propylitic zones. The earlier porphyry-style, veinlet-disseminated mineralization is composed of pyrite, chalcopyrite, molybdenite and bornite. Later-stage, disseminated epithermal-style pyrite, enargite, digenite, covellite and chalcopyrite frequently replace earlier pyrite, chalcopyrite and molybdenite in advanced argillic

metasomatites [281]. Porphyry stocks hosting both Naruo porphyry-breccia and Tiegelong-nan porphyry-epithermal mineralization display adakitic chemistry and are interpreted as slab melts [280]. Adakite porphyries are closely associated with Nadun Nb-enriched arc basalts (NEABs; $TiO_2$ = 1.16–1.69 wt.%; $Na_2O+K_2O$ = 3.44–6.01 wt.%; Nb = 15.4–18.6 ppm), which form dikes in the vicinity of porphyry intrusions. Sr–Nd–Hf isotopic systematics indicate that Duolong adakites and Narun NEABs are not only contemporaneous, but also most probably produced from an isotopically similar MORB-like source (subducted basaltic slab and adakite-hybridized depleted mantle) contaminated by small amount of oceanic sediment melt [280]. Interestingly, Nadun NEABs display systematically higher Cu contents (82–151 ppm) in comparison with contemporaneous island-arc basalts (IABs) in West Tibet (37–67 ppm) as well as global IABs (~50 ppm) [280]. The partial melting of the oceanic crust with high copper contents of 60 to 120 ppm produces Cu-enriched slab melts [282], which, upon interaction with Cu-poor mantle wedge peridotites, are capable of forming re-fertilized, Cu–Au-bearing, hybrid mantle sources for global HNB- and NEAB-type magmas. In summary, Cretaceous adakites in Tibet are slab melts locally associated with Nb-enriched arc basalts and Cu–Au–Ag porphyry-epithermal ore systems, while Miocene Cu–Mo(Au) skarn/porphyry deposits are genetically related to partial melts from the thickened juvenile (basalt underplated) crust of the Tibetan Plateau.

### 4.4.2. Iran and Pakistan

Adakite-hosted Cu–Au porphyry deposits are present in the Chagai magmatic arc of Western Pakistan, which occupies the southern part of the Afghan block accreted along the southern margin of the Eurasian plate in Eocene-Oligocene. The largest Saindak porphyry deposit contains 1.7 million tons of copper at 0.4% Cu and 0.30–0.48 g/t Au. Copper–gold mineralization is closely associated with the Early Miocene (22–24 Ma) monzodiorite and granodiorite porphyry intrusions emplaced into the Eocene shale, limestone and felsic to intermediate calc-alkaline lava and volcaniclastic conglomerate [275]. The granodiorite porphyry carries pyrite-, chalcopyrite- and molybdenite-bearing, veinlets and disseminated mineralizations. Trace element characteristics and Sr–Nd–Pb isotopes suggest a slab-melting origin for Chagai arc adakites with some possible additions of subducted sediment component [275]. Adakites are found in northwestern Iran, in the central part of the Zagros Ranges, in the eastern Iran region (around Bibi-Maryam) and in the southeastern portion of the Urumieh-Dokhtar magmatic arc [269,283–285]. Adakites in NW Iran are frequently associated with contemporaneous high-Nb basalts [137,264,265] and Cu–Au porphyry and epithermal mineral deposits [269,270,283]. The Miocene Sungun Cu–Mo porphyry systems on the NW terminus of the Urumieh-Dokhtar magmatic arc (Figure 21) are centered on high-K granodiorite stocks [286] with adakite geo chemical characteristics ($K_2O$ = 2.4–3.5 wt.%; Sr = 363–1984 ppm; Y = 8–17 ppm; Figure 22). The common presence of anhydrite, high zircon $Ce^{4+}/Ce^{3+}$ ratios (33–728), relatively low zircon saturation temperatures (620–804 °C) and abundant plagioclase phenocrysts with excess Al content suggests that these fertile adakite magmas were formed via water-fluxed melting of the subduction-modified juvenile lower crust beneath the Urumieh-Dokhtar magmatic arc of NW Iran [283]. Late Miocene (11.7–11.0 Ma) adakites host the large Sari Gunay epithermal gold deposit (52 Mt of ore grading 1.77 g/t Au) in the Hamedan-Tabriz volcanic belt of the Sanandaj-Sirjan tectonic zone [270]. The geochemical features of these high-silica adakites (Figure 22) are consistent with their formation through the partial melting of the delaminated continental lithosphere and/or the lower crustal amphibolite following the collision between the Arabian and Iranian lithospheric plates [270].

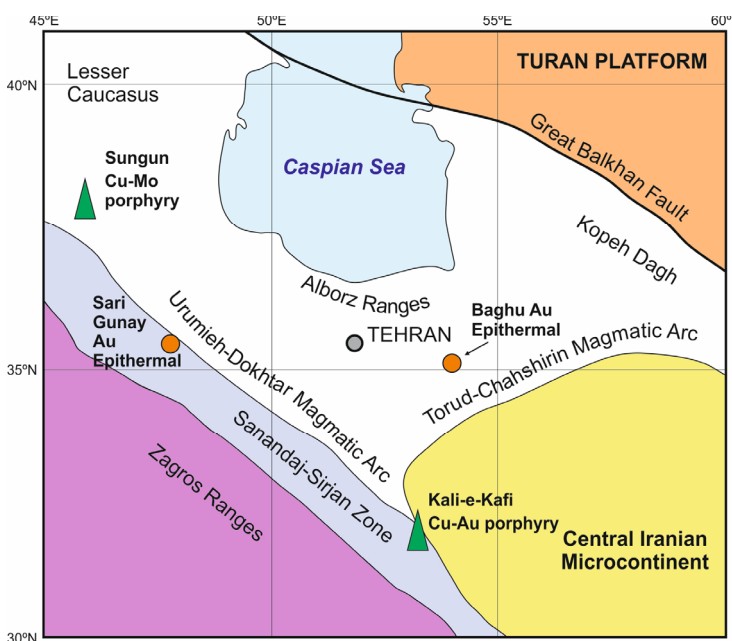

**Figure 21.** Distribution of major adakite-hosted Cu (Au, Mo) porphyry (triangles) and Au epithermal (circles) deposits in Iran (Western Tethyan Belt). Crustal terranes (Zagros Ranges, Sanandaj-Sirjan Zone, Alborz Ranges), cratonic cores (Turan Platform and Central Iranian Microcontinent) and magmatic arcs (Lesser Caucasus, Kopeh Dagh, Urumieh-Dokhtar and Torud-Chahshirin) are after [283,284,287,288].

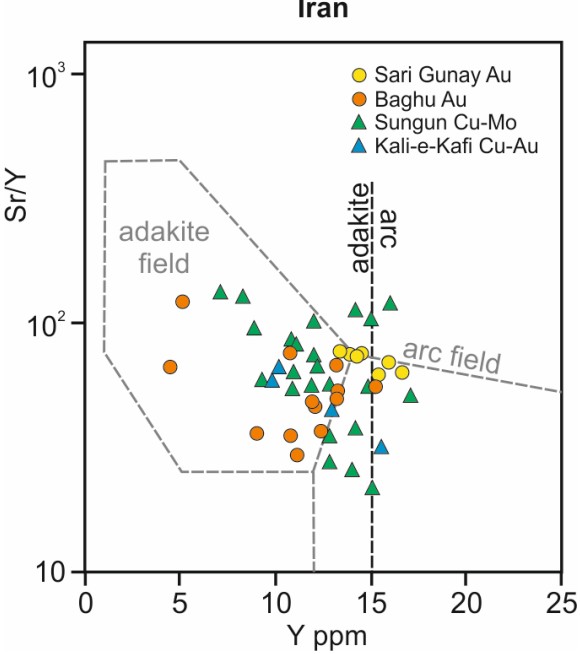

**Figure 22.** Variations of Sr/Y versus Y (ppm) in igneous rocks associated with porphyry Cu–Au–Mo (triangles) and epithermal Au–Ag (circles) deposits in the Western Tethyan province of Iran. Data sources: Sari Gunay [270]; Baghu [288]; Sungun [286]; Kali-e-Kafi [287].

The Eocene (50–52 Ma) Kali-e-Kafi gabbro-monzonite–quartz monzonite–granite complex intrudes on the Upper Proterozoic and Paleozoic metamorphic rocks and Eocene K-rich calc-alkaline volcanics in Central Iran [287]. Kali-e-Kafi intrusives host large-scale Cu–Mo–Au mineralization containing 245 million tons of ore at an average grade of 0.26% Cu and 0.026% Mo. The main ore stockwork (chalcopyrite, molybdenite, pyrite, magnetite, hematite, bornite, galena, sphalerite, electrum) is centered on quartz monzonite and granite

porphyry associated with potassic, phyllic, argillic and silicic alteration. Felsic rocks from the Kali-e-Kafi complex have low Y (average of 13 ppm) and Yb (average of 1.28 ppm) and high Sr (average of 1227 ppm) contents coupled with high Sr/Y (average of 108.4) and La/Yb (average of 22.2), indicating adakite affinity [286]. Geochemical modeling suggests that Kali-e-Kafi adakites were derived through partial melting of delaminated lower crust at pressures equivalent to crustal thicknesses in excess of 40 km [287]. Eocene (43–44 Ma) adakitic intrusions also host gold mineralization within the Torud-Chahshirin magmatic arc of the Semnan province in north-central Iran (Figure 21). The principal ore minerals in the Baghu deposit (3.5 million tonnes at 1.17 g/t Au) include native gold, pyrite, pyrrhotite, chalcopyrite, bornite, sphalerite, covellite, digenite, chalcocite and malachite [288]. Host adakitic granodiorites and granites (Figure 22) were most probably sourced from the thickened lower crust of the Torud-Chahshirin magmatic arc [288]. In general, Iranian adakites represent fractionates of fertile, highly oxidized, hydrous felsic magmas compositionally similar to metal-rich adakitic melts in Tibet, the CAOB, Andean magmatic arc, Kamchatka and other major Cu–Au porphyry and epithermal provinces of the world.

## 5. Concluding Remarks

Adakites are commonly defined as intermediate to felsic (54–70 wt.% $SiO_2$), Al-rich (typically > 15 wt.% $Al_2O_3$), predominantly sodic volcanic and plutonic rocks with abundant amphibole phenocrysts, pronounced Y and Yb depletions and characteristically high Sr/Y (>40) and La/Yb (>20) ratios. Adakite magmas typically erupt in convergent plate environments, where they occur in oceanic and continental arc settings, as well as collision zones and accretionary belts. Some adakites are emplaced in post-collisional and continental rifts in relation to the thermal erosion and lithospheric thinning of cratonic crust (e.g., North China Craton). Adakites are occasionally associated with Nb-enriched (Nb—10–20 ppm) and high-Nb (Nb > 20 ppm) arc basalts in some modern and ancient subduction settings, where they may or may not be linked genetically. Principal models for adakite petrogenesis include the partial melting of young and hot oceanic crust, the melting of thickened or basalt-underplated continental crust and the high-pressure fractionation (amphibole ± garnet) of mantle-derived, hydrous mafic magmas. In our opinion, it is rather difficult to discriminate between adakites of different origins using exclusively their chemical compositions. Consequently, the tectonic, geologic and structural settings of adakite series along with reliable geochronological and geochemical data should be used in geodynamic and petrologic reconstructions. High-Nb basalts in arcs are believed to be either (1) partial melts of adakite-hybridized mantle wedge peridotites (type I HNB—N-MORB-like Sr–Nd–Pb isotopes and OIB-like trace elements) or (2) mafic magmas derived from HFSE-enriched asthenospheric mantle sources incorporated into mantle wedge environments (type II—both isotopic and trace element signature is OIB-like).

Data from Paleozoic to Cenozoic magmatic arcs and collisional orogens presented above suggest strong spatial, structural, geochronological and, most probably, genetic links between adakites and high-Nb basalts and Cu–Au(Mo) mineralization. Adakites are water-rich (common presence of hornblende and biotite phenocrysts), highly oxidized magmas and also appear to be enriched in sulfur (anhydrite in Pinatubo and El Chichon adakites, magmatic barite in Stanovoy adakites) and chlorine (Cl-bearing mineral phases in Stanovoy adakites). Base (copper and molybdenum) and precious (gold, silver, platinum) metals are highly soluble and mobile in $H_2O$–S–Cl-rich melts. Consequently, hydrothermal fluids exsolved from such metal-rich adakite melts will be anomalously enriched in copper and noble metals. This is consistent with the widespread occurrence of Cu–Au–Ag intermetallic compounds in barite-bearing adakites from the Stanovoy Suture Zone. In addition, adakites in compressional (collisional) regimes tend to form crustal magmatic conduits, which are plausible environments for the exsolution of metal-rich hydrous fluids to form the high-level Cu–Au porphyry and Au–Ag epithermal systems. Slab-derived adakites possibly inherit their metal endowments from subducted metabasaltic crust and oceanic metalliferous sediments (including seafloor massive sulfides and mineralized hydrothermal vents).

Lower crustal adakites may have scavenged their metal inventory from the lower crustal sources re-fertilized by metal-rich asthenospheric melts. Some young volcanic adakites in the Southwest and Central Japan are not associated with any Cu–Au mineralization, although their slab melting origins appear to be unequivocal. The association of high-Nb basalts, especially in the absence of contemporaneous adakites, with Cu–Au mineralization in magmatic arcs appears to be more enigmatic. Type I HNBs may inherit high metal contents from the metal-rich, slab-derived adakites that hybridize depleted mantle wedge materials, while type II HNBs may possibly represent differentiates of metal-bearing mafic melts derived from fertile asthenospheric or metasomatized lithospheric mantle sources. We propose that the spatial, structural, temporal and possible genetic link between adakites and Cu–Au deposits at convergent plate edges is not an exclusive consequence of oxidized nature of adakitic magmas or specific conditions of associated tectonic regime (e.g., oblique, slow or flat subduction, subduction initiation, arc–transform interaction, etc.) but a direct reflection of initial metal endowment of primary adakite and high-Nb basalt melts derived from fertile crustal and mantle sources.

**Author Contributions:** Conceptualization, P.K., N.B. and N.K. (Nikita Kepezhinskas); methodology, P.K. and N.B.; validation, P.K., N.B. and N.K. (Nikita Kepezhinskas); investigation, P.K., N.B., N.K. (Nikita Kepezhinskas) and N.K. (Natalia Konovalova); resources, P.K., N.B. and N.K. (Natalia Konovalova); data curation, P.K. and N.B.; writing—original draft preparation, P.K.; writing—review and editing, P.K., N.B. and N.K. (Nikita Kepezhinskas); visualization, N.B., N.K. (Nikita Kepezhinskas) and N.K. (Natalia Konovalova); supervision, P.K. and N.B.; project administration, P.K. and N.B. All authors have read and agreed to the published version of the manuscript.

**Funding:** This research received no external funding.

**Institutional Review Board Statement:** Not applicable.

**Informed Consent Statement:** Not applicable.

**Data Availability Statement:** Not applicable.

**Acknowledgments:** P.K. thanks his long-time collaborators, Marc Defant, Mark Drummond, Elizabeth (Liz) Widom, Alfred Hochstaedter and many other Russian and international colleagues for numerous fruitful discussions on the nature of adakite and high-Nb basalt magmas and their relatioinships with base and precious metal mineralization in subduction zones. We acknowledge continuous support and encouragement for our work in the Stanovoy Range of the Russian Far East from Khingan Minerals AS and its Chairman, Tore Birkeland. This paper is dedicated to the bright memory of our colleagues Nikita Bogdanov, Jon Davidson, Igor Kravchenko-Berezhnoy and Alexey Osipenko, whose companionship, stewardship and friendship the senior author enjoyed for so many years during his wonderings in the misty mountains of the Kamchatka peninsula.

**Conflicts of Interest:** The authors declare no conflict of interest.

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
