# Peer review of "Adakites, High-Nb Basalts and Copper–Gold Deposits in Magmatic Arcs and Collisional Orogens: An Overview"

_geosciences, doi:10.3390/geosciences12010029_

Round 1

Reviewer 1 Report

The present manuscript presents an overview on adakites, high-Nb basalts and associated Cu-Au mineralization all over the world. The authors synthesize the available genetic models and major case studies for these related rocks around the world. The manuscript content is therefore thorough, comprehensive and fruitful, not only informative for beginners but also beneficial for experts. Obviously, the authors have spent a lot of effort for this in-depth review. I do not have any further comments on the manuscript, but only a few minor suggestions. First, the title shows that the manuscript deals with adakites, high-Nb basalts and associated Cu-Au mineralization in magmatic arcs. However, the manuscript also includes this rock association that may have formed in post-collisional and intracontinental (cratonic) extension settings. The title could thus be revised, e.g., deleting the words “in magmatic arcs”, to properly show the whole coverage. Secondly, the subdivision of text may also be modified a little bit in a simpler manner with the same logic, for example, as:

  1. Introduction
  2. Adakites
  3. High-Nb Basalts
  4. Examples of Adakites and High-Nb basalts hosting Cu-Au Mineralization

         4.1. Russian Far East

         4.2. Circum-Pacific Magmatic Arcs

         4.3. Central Asian Orogenic Belt

         4.4. the Tethyan Belt

  1. Concluding Remarks

Thirdly, it would be reader friendly to include, if possible, regional (tectonic) maps for the following sections/regions: 4.2. (original text subdivision scheme) Mesozoic Stanovoy Suture Zone (Russian Far East), 5.4. South-Eastern Alaska, 6. Adakite- and HNB-hosted Cu-Au mineralization in the Central Asian Orogenic Belt, and 7. Adakite-Hosted Cu-Au Mineralization in the Tethyan Belt, to cope with the various geographic locality names in text. In addition, although not necessary, it might still be better to use the same color symbols for porphyry and epithermal deposits, respectively, in all regional maps. And lastly, Dabie orogeny and Dexing (South China), as well as SW and Central Japan, are listed in Table 2, but are not discussed/mentioned in text examples. Are adakites in these regions not associated with mineralization? If so, a few comments in this respect are worthwhile in Concluding Remarks.

Finally, some typos in text need be revised:

  1. P.5. Correction (in bold-italic) in the following sentence: “Adakites associated with lower crustal melting appear to have systematically lower Al2O3 and higher?? K2O contents in comparison with slab melts and mafic differentiates, while MgO contents and Sr/Y ratios display almost complete overlap between the three types of adakitic compositions (Figure 1).”
  2. P.6. Correction (in bold-italic) in the following sentence: “High-Nb basalts can be aphyric or contain phenocrysts and micro-phenocrysts (Figure 4) of olivine, clinopyroxene (augite to Ti-augite), plagioclase, amphibole (pargasite or kaersutite), Ti-magnetite and, in some cases, ilmenite [52, 58, 59, 131, 132].”
  3. P.9. Corrections (in bold-italic) in the following sentences: “Recently Kepezhinskas and Kepezhinskas [153] proposed existence of two principal types of high-Nb basaltic magmas in volcanic arcs, which can be best distinguished by their Sr-Nd isotope variations (Figure 3D). Type 1 HNBs (exemplified by Kamchatka and Honduras; Table 3) are characterized by highest Nb contents (30-70 ppm) and Nb/LREE (e.g. Nb/La) and Nb/LILE (e.g. Nb/U) ratios coupled with most depleted (broadly similar to MORB and frequently more depleted than spatially and temporally associated “normal” arc lavas [52]) Sr and Nd isotopic signature similar to the Pacific MORB (Figure 3D).”

Reviewer 2 Report

  1. The some keywords could be modified (eg., slab melting, lower crustal melting, Cu-Ag-Au alloys).
  2. For the Stanovoy suture geological map is missing.
  3. Table 6. Could be better table of rocks modal compositions. In table 6 it is not clear composition of different hornblendes (Hbl 1-3)
  4. Description of Cu-Au deposits in Circum-Pacific magmatic a partly highly chaotic. For the South-Eastern Alaska the geological map is missing.
  5. For Cu-Au mineralisation in the Central Asian orogenic belt and the Tethyan belt the geological maps are also missing. And description of these mineralisation is also partly highly chaotic. 
  6. From the whole paper it is not clear, which data are original data and which data were published in papers of other authors.

Round 2

Reviewer 2 Report

The new version of manuscript is very interested and useful for researches, which are interested about ore mineralisation connected with adakites.